# Transcriptomics and trans-organellar complementation reveal limited signaling of 12-*cis*-oxo-phytodienoic acid during early wound response in *Arabidopsis*

Khansa Mekkaoui [1], Ranjit Baral[1], Fiona Smith[1], Moritz Klein [2], Ivo Feussner [2] & Bettina Hause [1]✉

12-*cis*-oxo-phytodienoic acid (OPDA), a precursor of jasmonoyl-isoleucine (JA-Ile), is known to have distinct signaling roles in Arabidopsis, as shown in studies using the *opr3* mutant, which lacks OPDA REDUCTASE3 (OPR3). This mutant, however, accumulates low levels of JA-Ile through an OPR2-mediated bypass. To investigate OPDA signaling, the wound-induced transcriptome of the *opr2opr3* mutant is compared to that of wild-type and *allene oxide synthase* mutant. Endogenous OPDA shows no unique transcriptional signature under control or wounding conditions, and previously identified OPDA-responsive genes are wound-induced independently of OPDA. Applying OPDA to *opr2opr3* triggers a distinct response suggesting compartmentalization of endogenously formed OPDA. Trans-organellar complementation reveals that expression of *OPR3* or *OPR2* in *opr2opr3* restores JA-Ile production regardless of localization, whereas mitochondrial targeted OPR1 exhibiting low OPDA/4,5-ddh-JA conversion activity does not. Our findings show that OPDA primarily serves as a JA precursor with limited independent signaling functions in the early wound response.

Jasmonic acid (JA) and its bioactive form jasmonoyl-isoleucine (JA-Ile) function primarily as signaling molecules coordinating plant growth and reproduction and responses to biotic and abiotic stresses[1,2]. They play a central role in the defense of plants against mechanical damage and herbivory, as JA-Ile rapidly accumulates at the site of injury and acts as a signaling molecule by triggering a cascade of defense responses[3]. JA-Ile signaling involves a series of molecular events mediated by its perception by the F-box protein CORONATINE INSENSITIVE 1 (COI1) leading to the degradation of the JASMONATE ZIM-DOMAIN (JAZ) repressors, hence the release of the expression of JA-responsive genes[4–6].

One major intermediate in the formation of the bioactive JA-Ile, 12-*cis*-oxo-phytodienoic acid (OPDA), has been known to be a signaling molecule with independent functions from JA-Ile in Arabidopsis, despite its signaling mechanism being uncharacterized[7,8]. OPDA formation takes place in the chloroplast via the octadecanoid pathway, by oxygenation of α-linolenic acid by 13-lipoxygenases (LOXs) and the coupled actions of ALLENE OXIDE SYNTHASE (AOS) and ALLENE OXIDE CYCLASE (AOC)[2] (Fig. 1a). Metabolization of OPDA into JA occurs after its transfer to peroxisomes, where the reduction of its cyclopentenone ring by OPDA REDUCTASE3 (OPR3) and three rounds of β-oxidation take place[2]. A parallel hexadecanoid pathway gives rise

[1]Department of Cell and Metabolic Biology, Leibniz Institute of Plant Biochemistry, Weinberg 3, Halle/Saale, Germany. [2]Department of Plant Biochemistry, Albrecht-von-Haller-Institute for Plant Sciences and Goettingen Center for Molecular Biosciences (GZMB), University of Goettingen, Justus-von-Liebig-Weg 11, Göttingen, Germany. ✉e-mail: Bettina.Hause@ipb-halle.de

to *dinor*-12-oxo-phytodienoic acid (*dn*-OPDA), which in turn is converted to JA following the same pathway as OPDA, but with two rounds of β-oxidation only[9].

The putative distinct function of OPDA in Angiosperms partially derives from its function in non-vascular plants by inducing defense- and growth inhibition-related transcriptional reprogramming in *Marchantia polymorpha*[4,10,11]. Such JA-independent functions of OPDA

were also characterized in vascular plants for different vegetative processes including initiation of tendril coiling in *Bryonia dioica*[12], regulation of embryo development in *Solanum lycopersicum*[13], and inhibition of seed germination and induction of stomatal closure in *Arabidopsis thaliana*[14,15]. Despite its perception being not identified in Arabidopsis, OPDA was shown to act as signaling molecule by transcriptionally inducing a set of genes, which were responsive to a lesser

**Fig. 1 | MeJA pretreatment restitutes AOC protein levels in the mutants and OPDA content in the *opr2opr3* mutant. a** Synthesis of jasmonic acid (JA)/JA-Ile from α-linolenic acid (α-LeA) starting with the release from chloroplast membrane galactolipids and its conversion to (13S)-hydroperoxyoctadecatrienoic acid (13-HPOT) by 13-LIPOXYGENASE (LOX). *cis*-(+)-12- oxophytodienoic acid (OPDA) is formed in the chloroplast (green background) from 13-HPOT through sequential actions of ALLENE OXIDE SYNTHASE (AOS) and ALLENE OXIDE CYCLASE (AOC). In the peroxisomes (yellow background) reduction of cyclopentenone ring of OPDA is catalyzed by OPDA REDUCTASE3 (OPR3) resulting in the formation of 3-oxo-2-(20(Z)-pentenyl)-cyclopentane-1-octanoic (OPC-8:0) followed by three rounds of β-oxidation to yield JA. The cytosol-released JA is conjugated to Isoleucine (Ile) by JASMONATE RESISTANT 1 (JAR1), resulting in the major bulk formation of the biologically active JA-Ile. OPDA can directly undergo β-oxidation, yielding 4,5-didehydro-JA (4,5-ddh-JA) in a parallel OPR3-bypass pathway. 4,5-ddh-JA released to the cytosol is then reduced to JA by OPR2 resulting in JA-Ile formation. Enzymes shaded in green, yellow and blue are located in the plastids, peroxisomes and cytosol, respectively. **b** AOC protein content in seedlings pretreated with 1 μM MeJA or water (Mock) visualized by immunoblot, equal loading is depicted by staining with No-Stain™ Protein Labeling Reagent. **c** AOC band intensity quantified relative to total protein loading. **d** RT-qPCR of JA-responsive genes *LOX2*, *AOS*, *AOC2* and *OPR3* showing transcript accumulation at control condition (-wounding) and at 1 h after wounding. Seedlings were pretreated with MeJA or water (Mock) during development. **e** Levels of *cis*-OPDA, *dn*-OPDA, JA and JA-Ile in seedlings of wild-type (Col-0), *opr2opr3*, and *aos* at control condition (-wounding) and 1 h after wounding. Seedlings were pretreated with MeJA or water (Mock) during development (see Supplementary Fig. 2a for experimental set-up). Bars represent means of three biological replicates with 120 seedlings each (single dots; ±SEM). Statistically significant differences among genotypes and condition were calculated using One-Way-ANOVA in (**c**) and Two-Way ANOVA in (**d**–**e**) followed by Tukey HSD post-hoc test and are indicated by different letters ($p < 0.05$). Source data is provided as a Source Data file.

extent to JA[16]. These OPDA-specific response genes were characterized as wound- and OPDA-responsive in a COI1-independent manner and encompassed mainly stress-responsive genes, e.g., *DRE-BINDING PROTEIN 2 A* (*DREB2A*), *FAD-LINKED OXIDOREDUCTASE* (*FAD-OXR*), *SALT TOLERANCE ZINC FINGER10* (*ZAT10*), *ETHYLENE RESPONIVE ELEMENT BINDING FACTOR5* (*ERF5*), *GLUTATHION TRANSFERASE6* (*GST6*), and the glutaredoxin member *GRX480*. OPDA contains an α, β-unsaturated carbonyl group, which is thought to react with nucleophilic molecules by Michael addition[17]. The electrophilic nature of oxylipins could convey biological activities, as deduced from the accumulation of oxylipins adducts, including OPDA-GSH conjugates, during the hypersensitive response in pathogen-elicited tobacco leaves[18]. OPDA binding to the CYCLOPHILIN20-3 (CYP20-3) was suggested to trigger the formation of the cysteine synthase complex activating sulfur assimilation to regulate cell redox homeostasis through mediation of *TGACG-BINDING* (TGA) transcription factors[19]. Here, CYP20-3-dependent OPDA signaling induces high levels of GSH leading to a retrograde signaling, which coordinates expression of OPDA-induced genes[20]. These transcriptional changes in Arabidopsis induced by OPDA through CYP20-3 module and/or additional uncharacterized mechanism consolidated the hypothesis of its function not limited to being only a JA precursor[8]. The detection of amino acid conjugates of OPDA in Arabidopsis further hinted towards its possible bioactivity[21] in line with the JA-Ile conjugate being the main bioactive form of JAs. Recent findings, however, indicate that the formation of OPDA amino acid conjugates in Arabidopsis contributes to the regulation of OPDA homeostasis rather than to conferring bioactivity to OPDA[22]. Similarly, and most importantly, the conjugation of the bioactive dn-*iso*-OPDA to amino acids in *Marchantia polymorpha* was recently shown to be a deactivation route of dn-*iso*-OPDA[23].

The previous work characterizing OPDA signaling and JA-independent functions in Arabidopsis relied mainly on the use of the *opr3* mutant as the genetic background where OPDA and JA biosynthesis were uncoupled due to missing OPR3[24]. Another mutant in the pathway is the *aos* mutant, which is deficient not only in JA but also in OPDA[25]. Both mutants are male sterile and show defects in anther dehiscence, stamen elongation and pollen development, indicating that these defects were due to impairments in JA production[24,25]. Persistent resistance of the *opr3* mutant against necrotrophic pathogens initially hinted towards a defense function of OPDA, but further studies uncovered a conditional removal of the T-DNA containing intron upon fungal infection of *opr3-1* on the one hand[26], and an OPR3-independent pathway leading to JA/JA-Ile formation through the cytosolic OPR2 in the *opr3-3* mutant on the other hand[27]. Low JA-Ile formation in both *opr3* mutants rather than OPDA was shown to be responsible for their resistance, hence challenging the suggested JA-independent OPDA signaling function[26,27]. Mainly the recently identified formation of JA-Ile via bypassing the OPR3 (Fig. 1a) questioned previously identified

functions of OPDA. Consequently, OPDA-specific signaling exclusively analyzed with OPDA-producing but JA/JA-Ile deficient mutants requires further studies.

In this work, we re-address the question whether OPDA can function as an independent signaling molecule in Arabidopsis besides being a JA precursor. Using the *opr2-1 opr3-3* mutant (further named *opr2opr3*) where formation of JA/JA-Ile does not occur[27], we show the effects of basal and wound-induced OPDA on early gene expression in seedlings and rosette leaves of *A. thaliana*. The results suggest that the endogenously formed OPDA is unlikely to be responsible for mediating early gene expression, and that the known OPDA-responsive genes were wound-inducible in an OPDA-independent manner. Exogenous supply of OPDA, however, results in a distinct signaling, including the activation of the sulfur assimilation pathway. This hints towards a predominant detoxification of the supplied OPDA and leads to the hypothesis that endogenously formed OPDA is compartmentalized within the cell. Therefore, we address the question of where OPDA occurs in the cell. A trans-organellar complementation approach suggests that OPDA is likely to be confined to the cell compartments where it is synthesized (plastids), transported through (cytosol) and metabolized (peroxisomes). These complementation results support the assumption that OPDA has no signaling function in the early wound response of Arabidopsis plants.

## Results

### Rescue of OPDA levels in *opr2opr3* using the JA feedback loop

The *opr2opr3* double mutant was characterized as JA-deficient, with JA formation through the peroxisomal OPR3 and alternatively through the cytosolic OPR2 being both interrupted[27]. To check this for 10-day-old seedlings grown in liquid medium, JA contents were measured in the *opr2opr3* and *aos* seedlings under both control and wounding conditions (Supplementary Fig. 1a). In the mutants, JA levels remained at the limit of quantification, with ~0.002 nmol/g of fresh weight JA measured in the wounded *opr2opr3*. This represents one hundredth of the reported 0.2 nmol JA per g fresh weight in wounded *opr3* mutant plants[24], and confirms the stronger JA deficiency of the *opr2opr3* mutant[27]. JA deficiency is associated with a decrease in the accumulation of JA-biosynthesis enzymes due to the disruption of the JA positive feedback loop during development, and among these enzymes specifically the AOC proteins which are encoded by four genes in Arabidopsis[28]. Consequently, it was expected that *opr2opr3* would produce lower amounts of OPDA due to reduced availability of biosynthetic enzymes in their tissues. Here, both the background and wound-induced levels of OPDA were observed to be lower compared to the wild type (Supplementary Fig. 1b). This reduction correlated with a diminished content of AOC proteins in *opr2opr3* and *aos*

seedlings compared to wild-type (Col-0) seedlings, as detected using an antibody that recognizes all four AOC isoforms[28] (Supplementary Fig. 1c, d).

Due to the partial OPDA deficiency of opr2opr3 seedlings, it was crucial to normalize the levels with those of the wild type prior to investigating OPDA signaling within this mutant. To achieve this, we mimicked the missing JA positive feedback loop in the opr2opr3 and aos mutants by supplying seedlings with 1 µM JA methyl ester (MeJA) during development to increase the AOC protein levels (Supplementary Fig. 2a). In contrast to water (mock-)pretreated seedlings, seedlings pretreated with MeJA during development exhibited increased AOC protein levels in wild type and both mutants (Fig. 1b). Notably, this increase was more pronounced in the mutants resulting in AOC protein levels in opr2op3 and aos seedlings matching those of wild type seedlings (Fig. 1c). Transcript accumulation of the JA biosynthesis enzymes LOX2, AOS, AOC2 and OPR3 determined by RT-qPCR corroborated the function of the JA positive feedback loop: MeJA pretreatment during development resulted in either compensation or an enhancement of their basal transcript levels in the mutants compared to wild type (Fig. 1d). They all are, however, highly induced upon wounding in wild-type seedlings, but their induction is significantly lower in the mutants despite the MeJA pretreatment (Fig. 1d). Most importantly, however, the restoration of AOC protein content in the opr2op3 enhanced its basal and wound-induced OPDA and dn-OPDA levels to the levels of wild type, whereas the aos mutant remained OPDA deficient as expected (Fig. 1e). Despite the accumulation of low residual levels of JA in opr2opr3 and aos seedlings upon MeJA pretreatment, wound-induced accumulation of JA and JA-Ile was only observed in wild-type seedlings and correlated with the transcriptional response of the JA-responsive genes (Fig. 1e, d). This validates the JA deficiency of both mutant lines within this experimental design and the exclusive OPDA enhancing role of the developmental JA-feedback loop for the opr2opr3 mutant (Fig. 1e).

## Transcriptomes from wounded wild type and mutant plants

To clarify OPDA-related signaling function, a comparative transcriptomics approach was set between wild type and opr2opr3 seedlings. To uncouple putative OPDA signaling from common wound-induced signaling processes, aos seedlings were included as a negative control, since both JA and OPDA productions are abolished in this mutant (Supplementary Fig. 2b). mRNA sequencing of MeJA-pretreated seedlings from Col-0, opr2opr3 and aos was performed using unwounded seedlings (control) and seedlings harvested 1 h after wounding. The overall wound-induced response showed a significant transcriptional change in all three genotypes (Supplementary Fig. 3). Here, a strong similarity between the transcriptomes of opr2opr3 and aos seedlings was evident in the expression heatmap, with a clear overlap observed in the principal component analysis (PCA), irrespective of the condition (Supplementary Fig. 3). The wound-induced transcriptional response showed most difference in the number of differentially expressed genes (DEGs) between Col-0 and the JA-deficient mutants, with the number of DEGs being 10.84% and 7.48% bigger in Col-0 than opr2opr3 and aos, respectively (Fig. 2a). A common wound response was observed in all the genotypes which was enriched in genes involved in stress, oxygen, and abiotic stimulus responses and was therefore rather wound-inducible but independent from the JA pathway (Fig. 2a, b, Supplementary Data 1). Interestingly, this group contained known stress-responsive genes with several genes being previously reported as OPDA-responsive, such as DREB2A and FAD-OXR, which were also induced in the aos mutant by wounding despite its OPDA deficiency[16] (Fig. 2c, d, and Supplementary Table 1, Supplementary Data 1). Additionally, other major OPDA-marker genes like ZAT10, ERF5, TCH4 and GST6 also showed a wound-responsive induction in aos seedlings indicating that they are associated with an OPDA-independent wound-

induced stress response (Fig. 2d). To further exclude a JA/JA-Ile dependence of their induction by wounding, transcript levels of these genes were monitored in rosette leaves of two coi1-mutant lines, coi1-16 and coi1-30 (Supplementary Fig. 4a, b). The data show that all tested genes are inducible by wounding to nearly the same level as in wild type, irrespective of the lower levels of OPDA in the coi1-mutants reaching in coi1-16 only 25 % of that of wounded wild-type leaves, whereas there was no increase in wounded leaves of coi1-30 caused probably by the higher age of the plants used (Supplementary Fig. 4c). Hence, the wound-induced transcription of these OPDA-markers in both coi1-alleles did not correlate with OPDA levels despite its COI1-independence, further showing that they are rather governed by other signaling processes in the wound response. JA and JA-Ile levels appeared to be significantly lower in wounded coi1 plants in comparison to wild type pointing again to the missing positive feedback in JA signaling and biosynthesis during plant development[28].

In wild-type seedlings, the signaling mediated by JA/JA-Ile was evident as it distinctly shaped the transcriptome, sustaining statistically differential basal levels of 88 JA-regulated genes when directly comparing the wild-type transcriptome to those of the mutants at control condition (Supplementary Fig. 5a, b). Wounding of wild-type seedlings resulted in a more drastic transcriptional change highlighted by 432 DEGs specifically induced in Col-0 and enriched in genes related to JA, wounding, and immune responses (Supplementary Fig. 5a, c, Supplementary Data 1). These transcriptional responses, notably of genes encoding JAZ transcription factors and other JA-responsive genes, such as CHLOROPHYLLASE 1 (CLH1) and N-ACETYLTRANSFERASE ACTIVITY 1 (NATA1), were dampened in the opr2opr3 and aos mutants confirming their JA dependence upon wounding (Supplementary Fig. 5d, and Supplementary Data 2). These data suggest that next to a common wound response, JA/JA-Ile are main mediators of early wound-induced transcriptional changes in wild-type seedlings.

To identify OPDA-regulated genes, transcriptional differences between opr2opr3 and aos were dissected and indicated a restricted number of DEGs at control and wounding conditions (Fig. 3a). Except for the mutated genes, which showed diminished expression in the respective mutant backgrounds (AOS, OPR2, and OPR3), only nine specific genes exhibited differential expression in opr2opr3 seedlings compared to aos seedlings, regardless of the experimental condition (Fig. 3b, c). These genes encode five RNA species and four proteins, among them one cyclic nucleotide-gated ion channel (CNGC11), one cytochrome P450 (CYP81D11), and two hypothetical proteins. Among them, CYP81D11 has been shown to be highly induced by lipophilic xenobiotics such as phytoprostanes, OPDA, and cis-jasmone[29,30]. Nevertheless, this is a very narrow or almost no difference between the transcriptomic profiles of the mutants, despite the significant induction of OPDA levels in opr2opr3 which was absent in aos (Fig. 1e). These results make an OPDA-mediated gene expression in wounded or control seedlings of the opr2opr3 mutant unlikely.

To investigate the impact of developmental stage on our findings, the wounding experiment was done using mature four-week-old rosettes (Supplementary Fig. 2c). Pretreatment with 1 µM MeJA enhanced OPDA levels in wounded leaves of opr2opr3 compared to the mock pretreated ones (Supplementary Fig. 6a). Regardless of whether the MeJA pretreatment was administrated, wounding led to a high number of DEGs in the wild type and a lower, but similar number of DEGs in the opr2opr3 and aos mutants (Supplementary Fig. 6b, c). Here, the bigger difference in the number of DEGs between wild type and mutants in comparison to seedlings is likely caused by the developmental stage. Most importantly, consistent with the findings for seedlings, the transcriptomes of the aos and opr2opr3 mutants were highly similar despite significant OPDA accumulation in the latter and differed only with the same DEGs found in the transcriptomes of seedlings (Supplementary Fig. 6d–f).

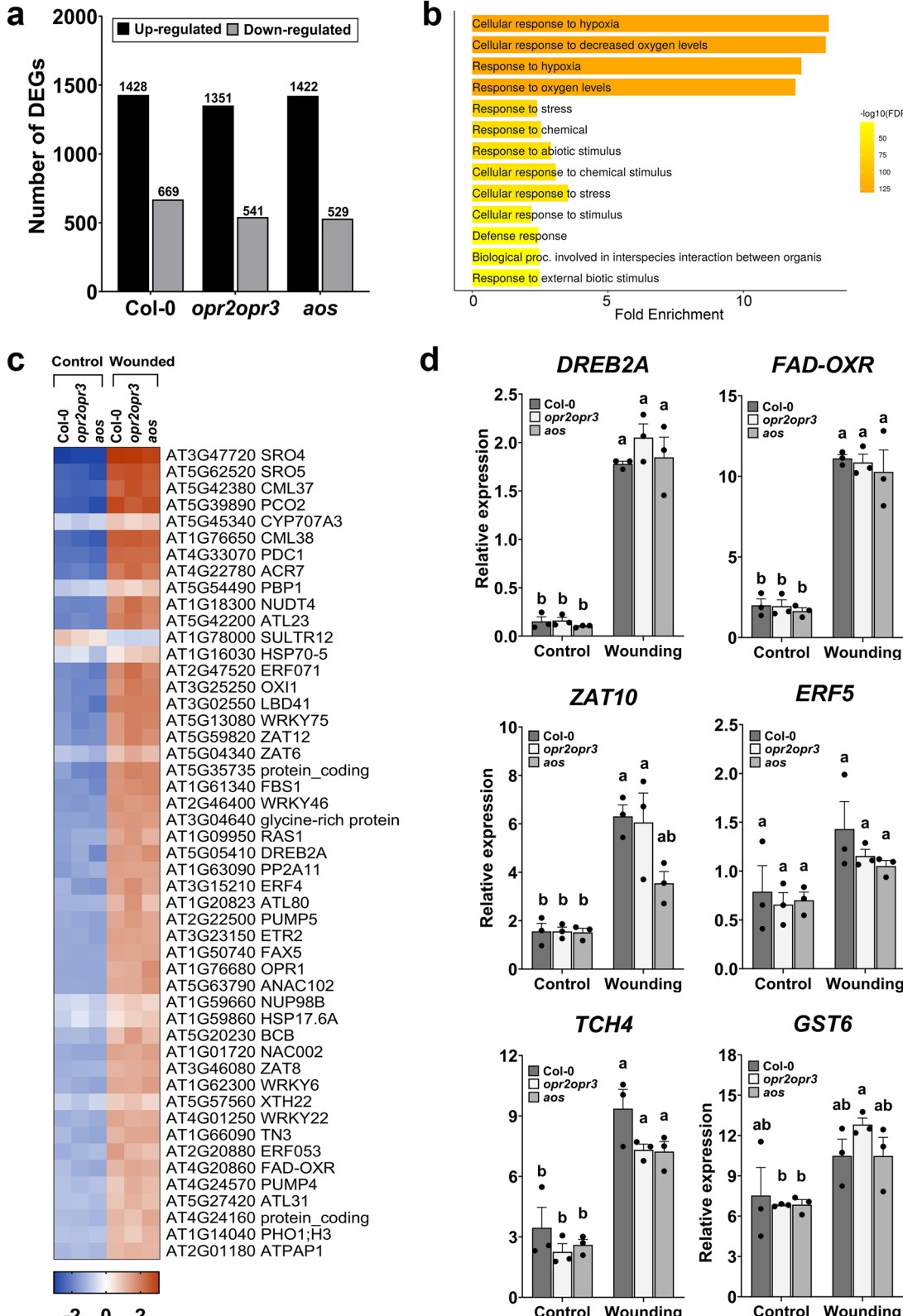

**Transcriptome of *opr2opr3* seedlings after OPDA application**

Previous work demonstrated a signaling role of OPDA in Arabidopsis by its exogenous supply to wild-type plants or to the *opr3* mutant[16,19,31]. To determine whether the origin of OPDA might cause the outcome, RNAseq data resulting from wounding of *opr2opr3* seedlings (= endogenous rise of OPDA) were compared to transcriptomic data generated from *opr2opr3* seedlings, which were treated with 25 μM OPDA for 30 min (= exogenous supply of OPDA). This time was chosen since already after 30 min the maximum of OPDA accumulation is reached inside the tissues[32]. OPDA treatment resulted in transcriptional changes that were highly different from those induced by wounding (Supplementary Fig. 7a, and Supplementary Data 3). The clustering of DEGs induced by OPDA treatment in comparison to DEGs induced by wounding showed a significant induction of a sub-cluster

**Fig. 2 | The JA/OPDA-independent wound response of seedlings.** Ten-day old seedlings from Col-0, *opr2opr3* and *aos* were pretreated with MeJA during development and were either unwounded (control) or harvested 1 h post wounding with forceps (wounding). **a** Number of differentially regulated genes (DEGs) after wounding compared to non-wounded controls in seedlings of Col-0, *opr2opr3* and *aos*, all pretreated with 1 μM MeJA during development (see Fig. S2 for experimental set-up). DEGs were identified using FDR and FC cutoffs of 0.05 and 2, respectively. **b** Bar plot showing Gene Ontology enrichment analysis summarizing the significant biological processes (FDR = 0.05) enriched commonly in all genotypes in response to wounding, with bars indicating gene fold enrichment and color scale indicating FDR values. **c** Heatmap illustrating 50 out of 211 DEGs enriched in the GO term 'response to stress' from (**b**). Average FPKM values of three independent biological replicates were transformed into row normalized Z-score. **d** RT-qPCR validation of DEGs previously described as OPDA-responsive genes. Transcript accumulation of *DREB2A, FAD-OXR, ZAT10, ERF5, TCH4,* and *GST6* in Col-0, *opr2opr3* and *aos* at control and wounding conditions. Transcript levels were normalized to those of *PP2A3*. Bars represent means of three biological replicates (single dots; ±SEM). Statistically significant differences among genotypes within each condition were calculated using Two-Way ANOVA followed by Tukey-HSD and are indicated by different letters. Source data are provided as a Source Data file.

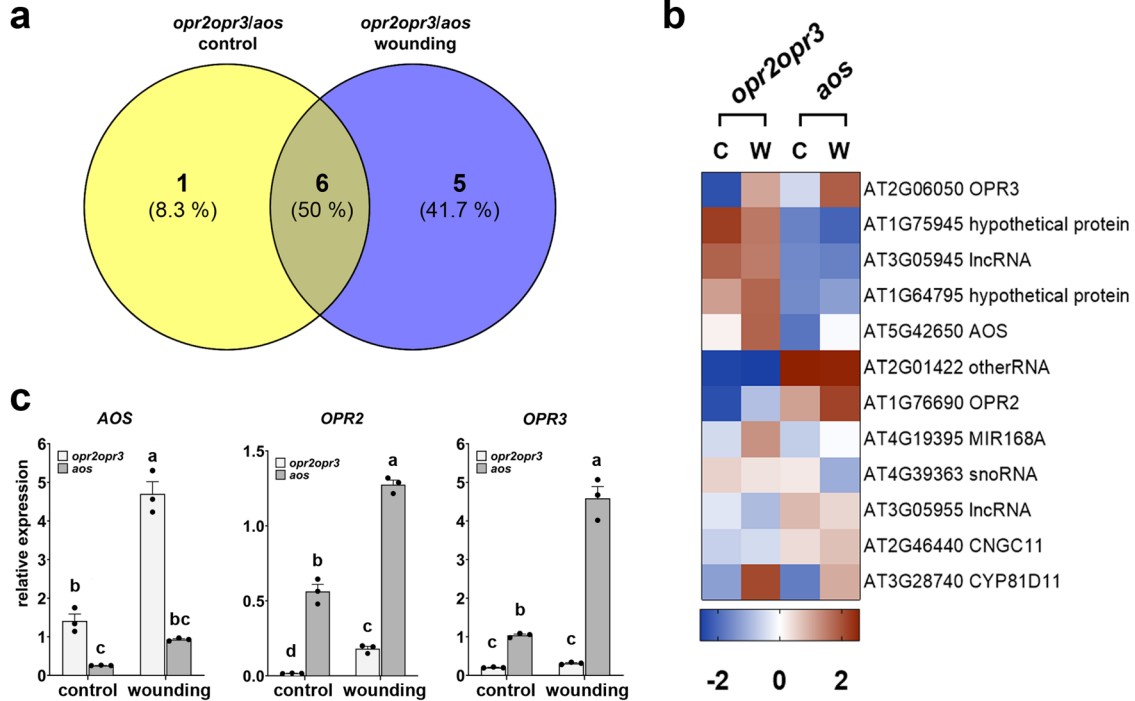

**Fig. 3 | OPDA-specific transcriptional change deduced from transcriptomic comparison between seedlings of *opr2opr3* and *aos* at control and wounding conditions.** Ten-day old seedlings from *opr2opr3* and *aos* were pretreated with MeJA during development and were either unwounded (control, C) or harvested 1 h post wounding with forceps (wounding, W). **a** Venn diagram representing genes showing differential expression when *opr2opr3* is compared to *aos* at control and wounding conditions (FDR cutoff = 0.05 and FC cutoff = 2). **b** Heatmap illustrating the expression levels of the eleven wounding-related DEGs from (A). The average FPKM values of three independent biological replicates were transformed into row-normalized Z-score. **c** Transcript accumulation of *AOS, OPR2,* and *OPR3* selected from (**b**) and determined by RT-qPCR. Transcript levels were normalized to those of *PP2A3*. Bars represent means of three biological replicates (single dots; ±SEM). Statistically significant differences among genotypes within each condition were calculated using Two-Way ANOVA followed by Tukey-HSD and are indicated by different letters. Source data is provided as a Source Data file.

of genes belonging to sulfate reduction and assimilation pathway, the phosphorelay signal transduction pathway as well as the hormone-mediated signaling pathway (Supplementary Fig. 7b, and Fig. 4a, Supplementary Data 3). These genes were not induced by the endogenous rise of OPDA and included genes encoding sulfate reductases, such as SULFATE-DEFICIENCY INDUCED (SDI1), APS REDUCTASE3 (APR3), and RESPONSE TO LOW SULFUR3 (LSU31), and ORA59, an APETALA2/ETHYLENE RESPONSE FACTOR (AP2/ERF) domain transcription factor which is involved in JA and ethylene signaling and in defense[33] (Fig. 4b). This is in line with the previously reported induction of sulfur metabolism by OPDA binding to the CYP20-3 module[19]. Here, the activation of the sulfur metabolism pathway was a specific effect of exogenously supplied OPDA that did not correspond to that of the endogenous compound. Formerly described OPDA-induced genes were induced by both OPDA treatment and wounding of the *opr2opr3* mutant, however, their wound induction in *aos* indicated their involvement in general stress-response pathways (Fig. 2d, and Fig. 4c). In addition, OPDA application to *opr2opr3* seedlings resulted

in a slight, but significant induction of JA signaling genes (Supplementary Fig. 7a, b, and Supplementary Data 3). Comparing OPDA-treated *opr2opr3* and wounded wild-type seedlings showed that they shared the up-regulation of several JA marker genes, such as those encoding JAZ transcription factors and JA biosynthesis enzymes (Supplementary Fig. 7c). Validation of transcript accumulation of *JAZ2, JAZ7, JAZ13* and *CHL1* by RT-qPCR confirmed their induction in *opr2opr3* seedlings upon OPDA application (Supplemental Fig. 7d). While the induction of *JAZ13* and *CHL1* was exclusive to OPDA treatment, *JAZ2* and *JAZ7* showed a weak induction by wounding in the *opr2opr3* seedlings, however, to a lower extent compared to OPDA treatment. This hints towards the conversion of the exogenously applied but not the endogenously formed OPDA to JA/JA-Ile despite the loss of OPR3 and OPR2 in the *opr2opr3* background. OPDA and JA levels in *opr2opr3* seedlings determined after wounding and OPDA treatment showed unequivocally that *opr2opr3* seedlings accumulated slowly higher levels of JA-Ile following OPDA application in comparison to wounding (Supplemental Fig. 7e). Despite these levels being lower

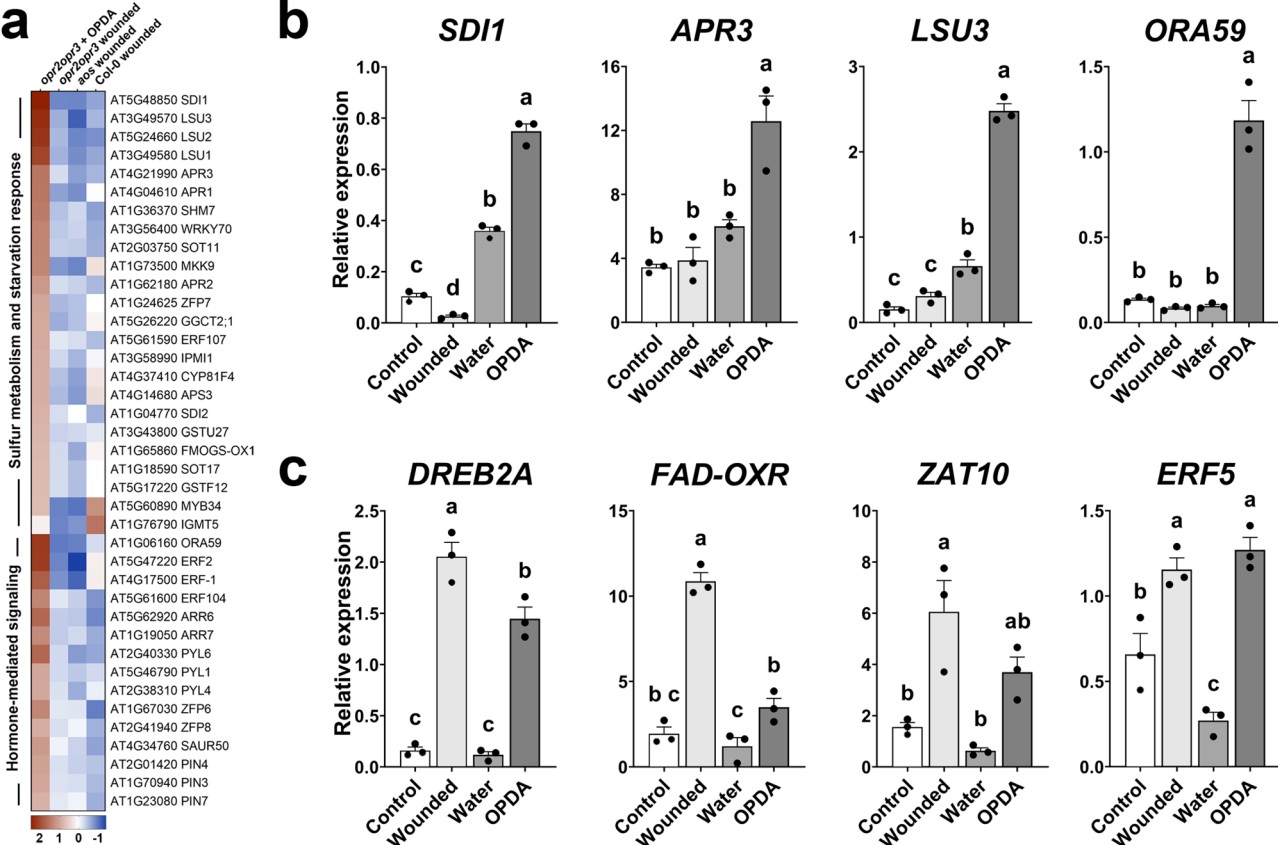

**Fig. 4 | Application of OPDA results in the induction of sulfur metabolism.**
**a** Heatmap of DEGs showing enrichment in genes involved in sulfur assimilation and common hormone signaling in seedlings of *opr2opr3* following application of 25 μM OPDA for 30 min. DEGs were plotted and compared to those from wounded seedlings of Col-0, *opr2opr3* and *aos* using average FPKM values transformed into row-normalized Z-score of three biological replicates. **b** RT-qPCR validation of the up-regulation of the sulfur metabolism genes *SDI1*, *APR3* and *LSU3* as well as the regulator of JA- and ethylene-responsive gene expression *ORA59* in *opr2opr3* seedlings treated with OPDA in comparison to wounded *opr2opr3* seedlings. OPDA-treatment and wounding of seedlings were compared to seedlings which were non-wounded (control) and water-treated (water), respectively. **c** RT-qPCR validation of the up-regulation of the OPDA-responsive genes *DREB2A*, *FAD-OXR*, *ZAT10* and *ERF5* in *opr2opr3* seedlings treated with OPDA in comparison to wounded *opr2opr3* seedlings. OPDA-treated and wounded *opr2opr3* seedlings were compared to their respective controls, non-wounded (control) and water-treated seedlings, respectively. Bars (**b**, **c** represent means of three biological replicates with 120 seedlings each (single dots; ±SEM). Statistically significant differences among treatments were calculated using Two-Way ANOVA followed by Tukey-HSD and are indicated by different letters. Source data is provided as a Source Data file.

than the JA-Ile levels in wounded wild-type seedlings, they correlated with the activated JA signaling in *opr2opr3* seedlings after OPDA application. Interestingly, the OPDA-fed *opr2opr3* seedlings accumulated only slightly higher OPDA levels than the wounded seedlings, whereas *dn*-OPDA accumulated significantly only following endogenous OPDA formation after wounding (Supplemental Fig. 7e). This suggests that the exogenously applied OPDA is either detoxified by conjugation to GSH[34] or rapidly converted to 4,5-ddh-JA as shown by ref. 35. In contrast, *dn*-OPDA accumulating after wounding in *opr2opr3* seedlings might be produced from OPDA entering the β-oxidation cycle[27] or from the parallel hexadecanoid pathway[9]. These results raise the question whether the different patterns of gene expression detected here are due to a compartmentalization of OPDA synthesized following wounding of seedlings, thereby not inducing the responses that the exogenously supplied OPDA does.

**Trans-organellar complementation of *opr2opr3* with OPR3**
The data obtained from wounded seedlings of *opr2opr3* and described above suggest that endogenously produced OPDA does not exhibit signaling capacity in the early wound response. This leads to the question, whether OPDA is restricted to the cell compartments of its synthesis, transfer and metabolization, namely plastids, cytosol and peroxisomes, respectively. To test its putative translocation to other

cell compartments, a trans-organellar complementation approach according to ref. 36 using the *opr2opr3* mutant was performed. OPR3 was targeted to different organelles in the *opr2opr3* mutant, and the fertility of these plants was checked. A rescue of the JA-deficiency phenotype of the *opr2opr3* mutant would indicate possible transloca-tion of its substrate, OPDA, to the tested organelles (Supplementary Fig. 8).

OPR3 is a peroxisome-localized protein and is imported into the matrix of peroxisomes (Fig. 5a, c) through the peroxisomal targeting signal type 1 (PTS1)[37]. OPR3 contains a C-terminal SRL, which is a var-iation of the prototypic PTS1 signal SKL. Removal of this signal resulted in cytosolic and nuclear localization of a fusion of OPR3ΔSRL with YFP when transiently expressed in *Nicotiana benthamiana* leaves, indicat-ing that import into peroxisomes was abolished (Fig. 5b). Exclusive targeting of OPR3 either to the nucleus, cytosol, plastids, mitochon-dria, or endoplasmic reticulum (ER) was done by fusing OPR3ΔSRL with the respective targeting signals or target peptides (Supplemen-tary Table 2). Validation of the subcellular localization of the different organelle variants of OPR3 with co-localization studies in *N. ben-thamiana* protoplasts was performed using established organelle markers[38] and indicated correct subcellular localizations of OPR3 compared to the markers independently of the position of the YFP fluorescent tag (Supplementary Fig. 9). Stable transgenic *opr2opr3*

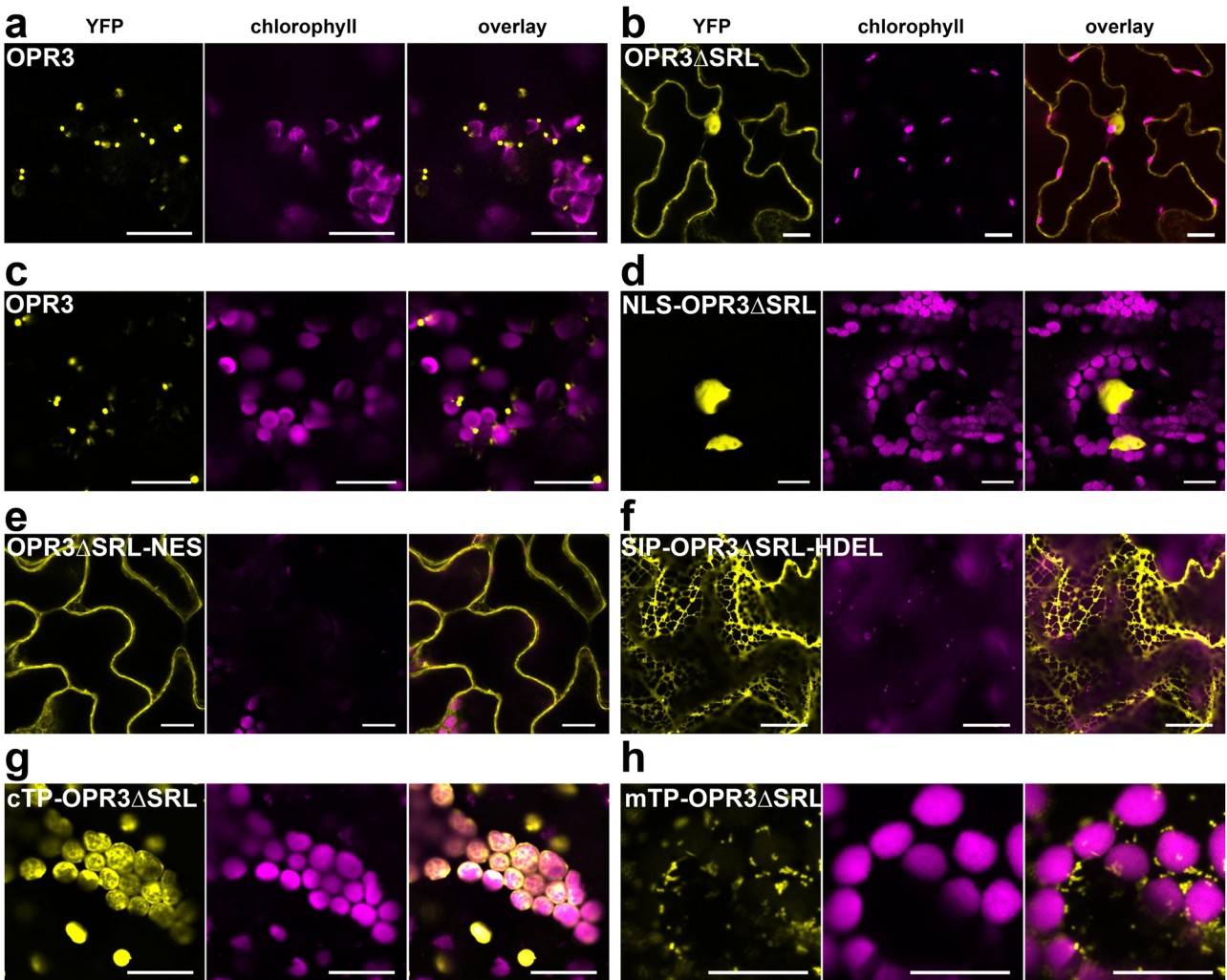

**Fig. 5 | The removal of the SRL peroxisomal signal results in the loss of peroxisomal import of OPR3 in *N. benthamiana* leaves and targeting of OPR3ΔSRL to the nucleus, cytosol, ER, plastid stroma and mitochondria in stable transformed *opr2opr3* lines. a, b** OPR3 (**a**) and OPR3ΔSRL (**b**) were C-terminally fused to YFP and transiently expressed in *N. benthamiana* leaves under control of the CaMV *35S* promoter. Imaging was performed at 48–72 h post infiltration. **c–g** Stable *opr2opr3* complementation lines expressing OPR3, which occurs in the peroxisomes (**c**), OPR3ΔSRL targeted to the nucleus (**d**), the cytosol (**e**), ER (**f**), chloroplast stroma (**g**) and mitochondria (**h**), all under control of the CaMV *35S* promoter. All pictures are taken from homozygous T2 single insertion lines. The yellow signal corresponds to YFP, whereas the magenta corresponds to chlorophyll autofluorescence. Scale bars represent 20 μm. NLS, nuclear localization signal; NES, nuclear export signal; SIP, signal peptide; cTP, chloroplast targeting peptide; mTP, mitochondria targeting peptide. Representative pictures from at least three independent experiments revealing similar results are shown.

lines complemented with the organelle variants of OPR3ΔSRL fused to YFP were generated and resulted in clear subcellular localization of OPR3-YFP to the nucleus, cytosol, ER, chloroplast stroma, and mitochondria (Fig. 5d–h). As tested by RT-qPCR, all transgenic lines exhibited a similar level of transgene expression (Supplementary Fig. 10).

The main phenotypic feature of JA-deficient and JA-insensitive Arabidopsis mutants is the male sterility, which is also characteristic for *opr2opr3* plants and is visible by defects in filament elongation, pollen release and seed set[27]. Plants of *opr2opr3* transformed with an empty vector (EV) showed short stamens with non-dehiscing anthers at the open flower stage 14 (Fig. 6a, b). As a positive control, the complementation of *opr2opr3* with the peroxisome targeted OPR3 showed full rescue of the flower phenotype (Fig. 6c). Interestingly, all other tested organelle-variants of OPR3 also rescued the flower fertility and showed open flowers like the wild type with elongated stamens and anthers releasing pollen resulting in silique formation and seed production (Fig. 6d–h). Plants complemented with the ER-targeted OPR3 having partial rescue of flower fertility were the

only exception. These plants formed fewer and smaller siliques compared to the wild type (Fig. 6f). This suggests that OPR3 localized to the ER may have restricted access to its substrate, potentially limiting JA/JA-Ile biosynthesis compared to other subcellular compartments.

Reconstitution of the JA pathway in vegetative tissue of the *opr2opr3* complementation lines was analyzed by determination of wound-induced levels of OPDA, JA and JA-Ile in two-week-old seedlings. Here, we selected lines complemented with OPR3 targeted to either peroxisomes, cytosol or mitochondria to compare the "native" site of OPR3 with its artificial location in the cytosol, where OPDA has to be transferred through, and one compartment, which is an organelle not involved in JA biosynthesis. Hormone levels of these lines were compared to those of wild-type and *opr2opr3* transformed with an empty vector. Data from two independent transformed lines each revealed that complementation of *opr2opr3* with OPR3 restored the JA/JA-Ile levels to wild-type levels (Fig. 6i). These results show the reconstitution of JA biosynthesis by OPR3 localized to the three tested compartments.

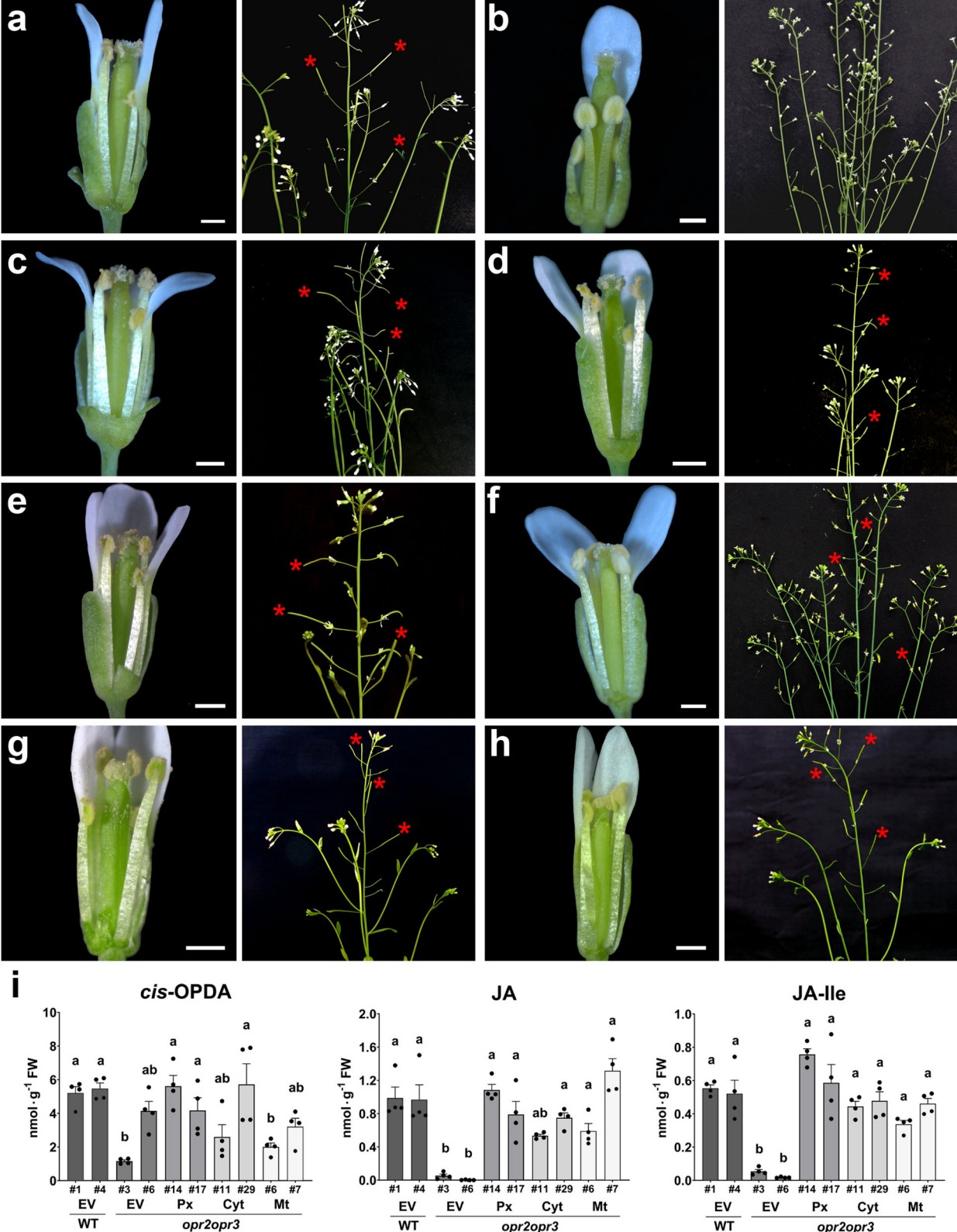

## Trans-organellar complementation of *opr2opr3* with OPR2 or OPR1

JA biosynthesis in *opr2opr3* complemented with OPR3 located in different organelles would occur only if its substrate, OPDA, is able to translocate to these compartments. However, as an alternative pathway, OPDA could enter directly the β-oxidation as reported recently[27] and in ref. 35. This would result in the formation of 4,5-ddh-JA, which might be metabolized by OPR3 irrespective of the organellar location. Conversion of 4,5-ddh-JA to JA by OPR3 was tested by in vitro enzymatic assay and showed that OPR3 reduces 4,5-ddh-JA in vitro, however to a lesser extent than OPDA (Supplementary Fig. 11a). Based on substrate consumption after 10 min, only half of the given 4,5-ddh-JA was reduced while OPDA was completely converted to 3-oxo-2-(20(*Z*)-pentenyl)-cyclopentane-1-octanoic (OPC-8). The control reactions with

**Fig. 6 | Restitution of the fertility and JA/JA-Ile levels of *opr2opr3* plants by OPR3-targeted to different organelles. a–h** Open flower buds showing stamen filament elongation and pollen release by anthers (scale bars represent 0.5 mm) and plant inflorescences showing silique formation at days 50-60 post germination with asterisks marking fully developed siliques. Col-0 and *opr2opr3* transformed with an empty vector (**a**, **b**) show fertility and sterility, respectively. *opr2opr3* T2 lines complemented with peroxisome- (**c**), nucleus- (**d**), cytosol- (**e**), ER- (**f**), plastid stroma- (**g**) and mitochondria- (**h**) targeted OPR3 fused to YFP show the rescue of the *opr2opr3* sterility phenotype. Red asterisk show exemplary siliques containing

seeds. **i** OPDA, JA and JA-Ile accumulation at 1 h after wounding of seedlings of wild-type (WT) or *opr2opr3* transformed with either empty vector (EV) or *35S::OPR3* targeted to peroxisomes (Px), cytosol (Cyt) or mitochondria (Mt). Numbers refer to independent transgenic lines, which were selected for single insertion and homozygosity. Bars represent means of four biological replicates with 120 seedlings each (single dots; ±SEM). Statistically significant differences between *opr2opr3* transformed with empty vector and transformed with the OPR3 constructs were calculated using One-Way ANOVA followed by Tukey HSD (p < 0.05) and are denoted by different letters. Source data are provided as a Source Data file.

inactivated enzyme showed no reduction activity. The results indicate that OPR3 reduces both cyclopentenones, 4,5-ddh-JA and OPDA, in vitro. This raised the question whether indeed OPDA is the substrate able to translocate to the tested cell compartments or whether 4,5-ddh-JA is the mobile compound. To discriminate between both possibilities, peroxisomal, cytosolic and mitochondrial variants of OPR1 and OPR2, both fused to YFP, were used to complement the *opr2opr3* mutant. OPR2 was shown to convert 4,5-ddh-JA with an apparent $K_M$ in the same range as OPR3 exhibiting for OPDA[27], but is less active on OPDA[39], whereas OPR1 reduces OPDA at very low rate but not 4,5-ddh-JA[27,39]. To confirm this, both enzymes were produced recombinantly in *E. coli* and were tested for their activity on OPDA and 4,5-ddh-JA (Supplemental Fig. 11b, c). As expected, OPR2 consumed 4,5-ddh-JA completely within the reaction time, but converted only one third of the given OPDA to OPC-8. Under our conditions, the reduction of OPDA by OPR1 was also very low, whereas 4,5-ddh-JA was reduced to JA by OPR1 at detectable rates.

OPR2 and OPR1 are located within the cytosol, but the addition of either a C-terminal SKL or an N-terminal mitochondrial target sequence led to localization in peroxisomes and mitochondria, respectively (Fig. 7a). All three variants expressing OPR2 rescued the flower fertility and showed open flowers like the wild type with elongated stamens and anthers releasing pollen (Fig. 7b) as well as formation of seed-containing, fully developed siliques (Fig. 7c). By contrast, expression of OPR1 located to peroxisomes was the only OPR1 variant leading to a rescue of the flower fertility, whereas targeting OPR1 to the cytosol led to partial rescue only and targeting OPR1 into the mitochondria did not restore the fertility (Fig. 7b, c). These data were supported by the fact that only seedlings harboring the OPR2 variants and peroxisomal-located OPR1 were able to accumulate JA and JA-Ile upon wounding (Fig. 7d), although all lines accumulated similar levels of OPDA (Supplementary Fig. 12). The JA/JA-Ile levels were lower than those of wounded wild-type seedlings, but the significantly increased levels in comparison to *opr2opr3* transformed with the empty vector showed the capability of these plants to produce JA/JA-Ile. Most importantly, however, expression of OPR1 located to cytosol or mitochondria did not restore wound-induced JA/JA-Ile levels. Considering that the specific activities of OPR1 towards OPDA and 4,5-ddh-JA are very low, we conclude that the metabolite levels in cytosol and mitochondria might not be sufficient for OPR1 enzymatic activity and therefore did not lead to the biosynthesis of JA/JA-Ile.

## Discussion

JA plays a major role in growth and defense against pathogens, herbivory attacks and mechanical wounding. Its signaling through the bioactive JA-Ile is well established in Angiosperms, while in non-vascular plants, such as *M. polymorpha*, OPDA, *dn-iso*-OPDA and $\Delta^4$-*dn-iso*-OPDA are bioactive[10,40]. Given that *A. thaliana* plants produce OPDA and *dn*-OPDA, possible function of OPDA as a signaling molecule independently from JA was proposed. OPDA was shown to induce a distinct transcriptional change from JA when exogenously applied to Arabidopsis plants[16,31]. Additionally, OPDA´s involvement in processes like seed germination and stomatal closure was demonstrated[14,15]. The mechanism by which OPDA exerts these functions in vascular plants is, however, still elusive[7]. Here, we addressed the question of whether

OPDA functions independently from JA as a signaling molecule in Arabidopsis by utilizing the *opr2opr3* mutant, which does not produce JA/JA-Ile upon wounding[27].

To inspect the separate roles of OPDA and JA upon wounding, a comparative analysis of the transcriptomes of seedlings of wild type, *opr2opr3* and *aos* was carried out. The use of the *opr2opr3* mutant is, however, limited by its lower production of OPDA in comparison to the wild type due to the disrupted JA positive feedback loop as indicated by reduced protein levels of the biosynthesis enzymes, such as AOC[2]. The enhancement of the AOC levels and subsequent OPDA production in the JA-deficient mutants was achieved by restituting the JA feedback loop through a repetitive supply of MeJA to seedlings during development (Fig. 1). The transcriptomic change induced by wounding in wild-type seedlings involved an up-regulation of genes previously identified as JA-dependent or associated with immune responses, validating the specific signaling role of JA-Ile in the wound response of seedlings as it has been described for mature leaves of Arabidopsis and other plant species[3]. Unexpectedly, seedlings of the *aos* mutant being deficient in synthesis of JA and OPDA, also showed a major transcriptional change upon wounding that was common to the *opr2opr3* mutant and the wild type (Fig. 2). The transcriptional response was significantly enriched in general stress response pathways, with an induction of oxidative stress- and transcription-related genes. This implies that JA-independent signaling processes occur in the early response to mechanical wounding leading to transcriptional reprogramming. Among such processes, water stress might contribute to the regulation of wound-responsive genes in a JA-independent manner[41]. Ethylene and abscisic acid (ABA) also contribute to the wound response by regulating photosynthesis and drought responsive genes, respectively[42]. An ABA-dependent and JA-independent transcriptional regulation of wax biosynthesis to seal the wounded sites of Arabidopsis leaves has been recently characterized, but appears to be controlled post-translationally by JA[43]. As plants defective in JA/JA-Ile biosynthesis or perception are severely diminished in their defense response[24,44], the JA-independent wound response has rather limited contribution to plant defense and might predominantly mitigate the adverse effects of wounding itself by promoting wound healing and tissue regeneration[45].

The general stress response that was both JA and OPDA-independent, comprised genes that were previously identified as OPDA-specific response genes, such as *DREB2A*, *FAD-OXR*, *ZAT10*, *ERF5*, *TCH4*, and *GST6*[16] all of them induced by exogenous application of OPDA (see Fig. 3c and[35]). These genes, however, did not show differential expression when the transcriptomes of *opr2opr3* and *aos* were compared, indicating that their induction by wounding is OPDA-independent (Fig. 2). Moreover, these genes were induced by wounding also in leaves of two *coi1*-mutant lines excluding any induction by JA/JA-Ile possibly caused by an insufficient JA-deficiency of the mutants. Several abiotic stresses were reported to up-regulate the same genes[46,47], further indicating their independence from OPDA. Moreover, direct comparison of transcriptomes from wounded *opr2opr3* and *aos* seedlings with standard cut-off parameters did not yield a significant list of putative OPDA-regulated genes further confirming the absence of an OPDA signaling (Fig. 3). This could be due to several facts: (i) A putative receptor for OPDA might be missing in *A.*

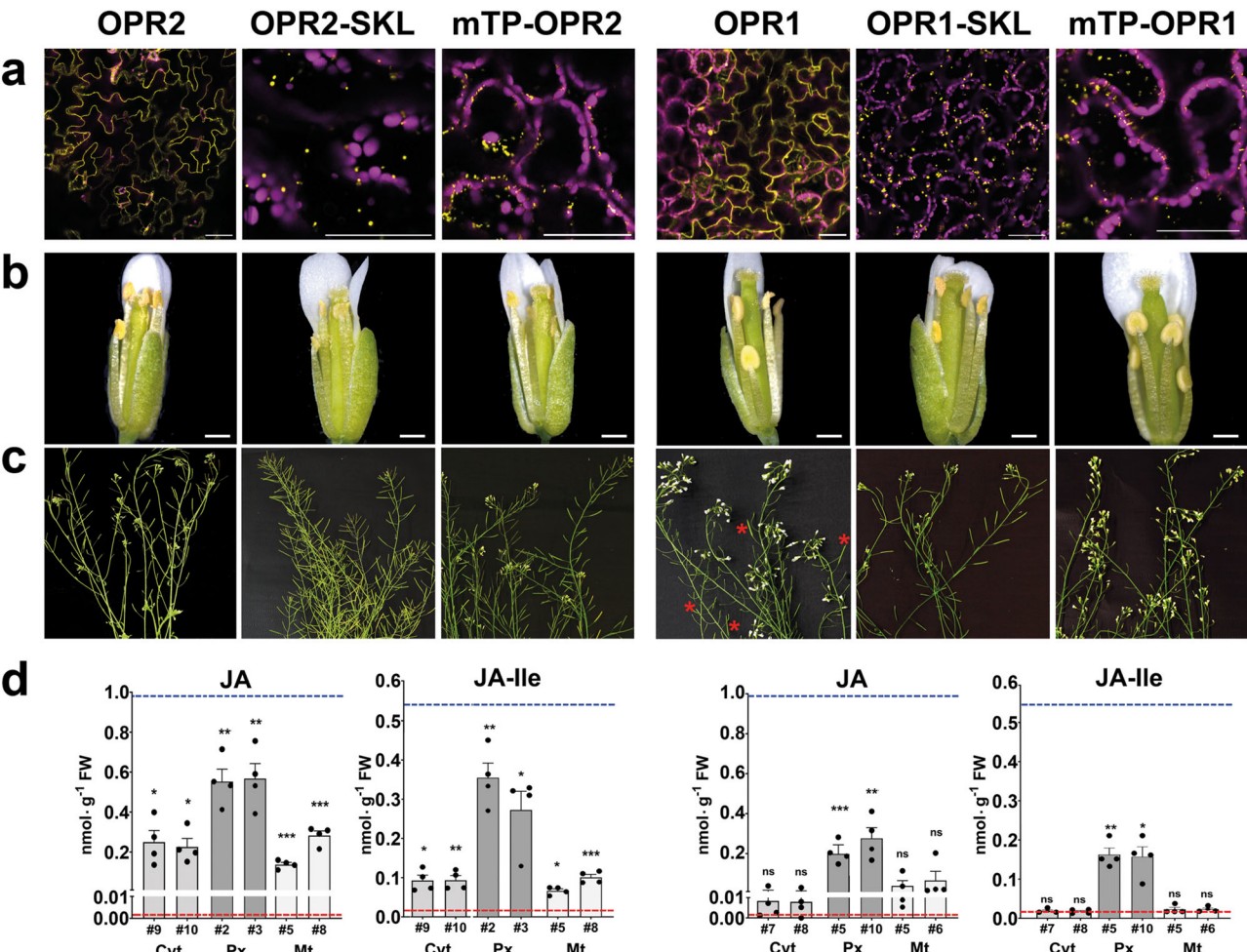

**Fig. 7 | Restitution of the fertility and JA/JA-Ile levels of *opr2opr3* plants by OPR2, but not by OPR1, both targeted to different organelles.** Stable *opr2opr3* complementation lines expressing OPR2 (left) or OPR1 (right) targeted to the cytosol (without any signal peptide), to the peroxisomes (OPR-SKL) or to mitochondria (mTP-OPR), all under control of the CaMV *35S* promoter. All transgenic lines showed a similar level of transgene expression (Supplementary Fig. 10). **a** Micrographs showing the correct localization of the respective OPR-YFP-fusion. The yellow signal corresponds to OPR-YFP, whereas the magenta signal corresponds to chlorophyll autofluorescence. Scale bars represent 50 μm. **b**, **c** Open flower buds to show stamen filament elongation and pollen release by anthers (**b**) and plant inflorescences showing silique formation at days 50-60 post germination (**c**). Note that transformation of *opr2opr3* with all variants of OPR2 and with OPR1 targeted to peroxisomes resulted in a rescue of fertility visible by elongated filaments and fully developed siliques. Targeting of OPR1 to cytosol led to partial

rescue shown by few fully developed siliques only (red asterisks), whereas targeting of OPR1 to mitochondria did not rescue fertility. Scale bars in (**b**) represent 0.5 mm. **d** JA and JA-Ile accumulation at 1 h after wounding of seedlings of *opr2opr3* transformed with either *35S::OPR2* or *35S::OPR1*, either targeted to cytosol (Cyt), peroxisomes (Px), or mitochondria (Mt). Numbers refer to independent transgenic lines, which were selected for single insertion and homozygosity. The mean levels of JA/JA-Ile of wild type and *opr2opr3* transformed with the empty vector are taken from Fig. 6 and are represented for comparison by blue and red dashed lines, respectively. Bars represent mean of four biological replicates with 120 seedlings each (single dots; ±SEM). Statistically significant differences between *opr2opr3* transformed with the different *OPR2* and *OPR1* constructs and *opr2opr3* transformed with the empty vector were calculated using Student's t-test and are indicated by asterisks with *$p < 0.05$, **$p < 0.01$, and ***$p < 0.001$; *ns* non-significant. Source data and exact *P* values are provided as a Source Data file.

*thaliana*, (ii) products from the α-linolenic acid pathway upstream of the AOS branch and present in the *aos* mutant convey similar transcriptional responses as OPDA therefore masking its effect, (iii) OPDA is not genuinely a signaling molecule and is converted to 4,5-ddh-JA[27], which does not accumulate to levels having signaling capacity[35], and (iv) OPDA remains largely sequestered in the plastids by its biosynthetic enzymes[48]. Regarding perception of OPDA, evolutionary studies of the JA-Ile co-receptor COI1 showed that the Arabidopsis protein does not perceive OPDA or *dn*-OPDA, dismissing possible signaling of OPDA through COI1 in this plant[4]. It has been shown that *dn-iso*-OPDA rather than *dn*-OPDA is the bioactive molecule in mosses, and that this compound does not occur in *A. thaliana*[10]. Moreover, the transcriptomic data obtained here from *opr2opr3*, *aos*, *coi1-16* and *coi1-30* support the conclusion that OPDA or its homolog *dn*-OPDA do not mediate a transcriptional change through a receptor or their

electrophilic properties in Arabidopsis seedlings when they occur basally at control condition or accumulate in the early response to wounding.

A transcriptional signature of OPDA was predominantly observed in experiments relying on its exogenous application[16,19,31]. Therefore, we compared the transcriptional changes following application of OPDA to *opr2opr3* seedlings with those by wound-induced, endogenously produced OPDA. This comparison revealed an up-regulation of sulfur assimilation and GSH production by exogenous OPDA, processes that were previously attributed as specific signaling functions of OPDA[19]. Here, the transcriptional activation of sulfur metabolism pathway was exclusive to the exogenous OPDA supply, contradicting an intrinsic signaling function of the endogenous OPDA (Fig. 4). As an electrophilic species, OPDA was shown to disturb redox homeostasis[49] and to inhibit photosynthesis by affecting the photosystem II

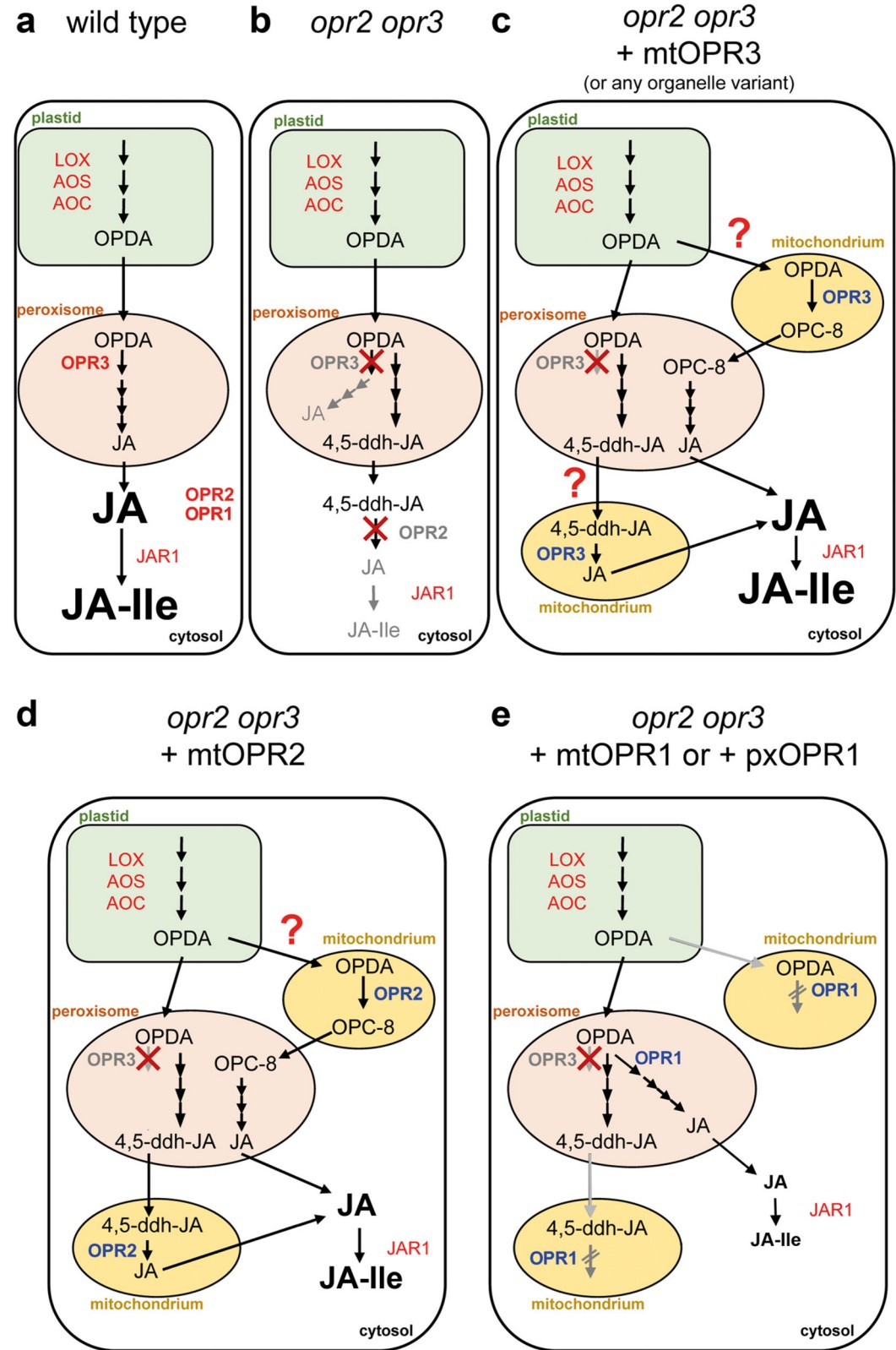

fluorescence[17,30]. In parallel, the *in planta* conjugation of OPDA to GSH, which is an important antioxidant substance, and its degradation in the vacuole were also demonstrated[34]. This implies that OPDA may not directly regulate the sulfur assimilation pathway but instead triggers this response through its potential detoxification when exogenously applied to plants, suggesting that the observed transcriptional response to exogenously applied OPDA could be a side effect rather than a direct effect of OPDA itself. In this line, Ueda et al.[35] showed that OPDA applied to Arabidopsis seedlings does not serve as genuine bioactive form responsible for the expression of genes. Instead, applied OPDA seems to be predominantly converted to 4,5-ddh-JA, which is inducing the expression of specific response genes in a dose-dependent manner, possibly due to its electrophilic properties. After application of OPDA, 4,5-ddh-JA accumulates up to 4000 pmol per g

**Fig. 8 | Schematic picture on the approaches leading to insights into the putative distribution of OPDA and 4,5-ddh-JA within cells of a wounded Arabidopsis seedling. a** In wild-type plants, OPDA is converted to JA in the peroxisome through the OPR3 canonical route. High amounts of JA/JA-Ile are produced upon wounding as visualized by large letters. The contribution of OPR1 and OPR2 located in the cytosol appears to be neglectable but the activity of OPR2 comes into place if OPR3 is not functional (e.g., *opr3* single mutant, see ref. [42]). **b** The interruption of both routes in the *opr2opr3* mutant results in conversion of OPDA to 4,5-ddh-JA, which is not metabolized to JA/JA-Ile. **c** Targeting OPR3 to any organelle in *opr2opr3* plants (depicted for mitochondrial localization) restitutes JA biosynthesis to almost wild-type levels (see Fig. 6). If this is a result of either an ability of OPDA or 4,5-ddh-JA to translocate to other cell compartments cannot be decided by this approach, since OPR3 is active to convert OPDA and 4,5-ddh-JA, although showing less activity towards 4,5-ddh-JA (red question marks). **d** Mitochondrial targeting of OPR2 with

its higher specificity to 4,5-ddh-JA complemented JA biosynthesis favoring rather an ability of 4,5-ddh-JA to be mobile, although the conversion of OPDA can not be excluded due to some activity of OPR2 towards OPDA (red question mark). The resulting JA/JA-Ile levels upon wounding are lower than in wild type (depicted by small letters), but still able to rescue the *opr2opr3* phenotype. **e** As last proof, OPR1 was targeted to cytosol, peroxisomes or mitochondria, because OPR1 shows only low activity in conversion of OPDA and 4,5-ddh-JA. mtOPR1 in *opr2opr3* did not restitute JA/JA-Ile production. Even the very limited JA formation by cytosolic OPR1 suggests restricted occurrence of OPDA or 4,5-ddh-JA in this compartment (not shown in this scheme, data see Fig. 7). Only OPR1 targeted to peroxisomes resulted in some JA/JA-Ile production restoring the phenotype (see Fig. 7, visualized by small letters). The lack of JA formation by mitochondrial OPR1, however, suggests that OPDA and/or 4,5-ddh-JA may occur only in neglectable levels within this compartment.

FW, whereas upon wounding of *opr2opr3* it accumulates up to 500 pmol per g FW only[27], a level that does not seem to be sufficient to induce a specific gene expression[35]. Therefore, the general stress response might be the main factor for transcriptional changes including the previously identified OPDA-marker genes as depicted from the wound response of *aos* mutant plants. These genes, being responsive to several abiotic stresses[46,47], including wounding in a JA-independent manner (as shown in this manuscript), further suggest their unspecific response to OPDA. This rather indicates that applying electrophilic compounds, such as 4,5-ddh-JA or OPDA, causes oxidative stress, resulting in the induction of general stress-responsive genes. Next to jasmonates, there are other compounds known that mediate at least a part of the wound response, such as reactive oxygen species (ROS)[50] and γ-aminobutyric acid (GABA)[51]. These JA-independent responses might be necessary for wound-healing, which seems to be a process independent on the JA-mediated defense response[45].

To address the question whether wound-induced OPDA remains largely sequestered in the organelles of its biosynthesis and metabolization, a trans-organellar complementation approach was performed by directing OPR3 into various organelles (Figs. 5, 6, and Supplementary Fig. 9). The occurrence of OPR3 in any of the tested organelles resulted in a complementation of the phenotype and JA/JA-Ile production upon wounding in *opr2opr3* mutants. This led to the assumption that OPDA might travel throughout the cell. However, the possibility still exists that OPDA is translocated rapidly via the cytosol to the peroxisomes, where it is converted to 4,5-ddh-JA by β-oxidation as shown previously for *opr3* and *opr2opr3*[27]. Given that OPR3 converts 4,5-ddh-JA to JA and OPDA to OPC-8 in vitro (Supplementary Fig. 11a), the complementation approach with OPR3 could not discriminate between the presence of either OPDA or 4,5-ddh-JA outside of the plastids or the peroxisomes (Fig. 8). To go into more detail, organelle variants of the isoenzymes OPR2 and OPR1 were used to complement the *opr2opr3* mutant (Fig. 7). With this approach we took into account that OPR2 and OPR1 are not so effective in conversion of OPDA, since the in vitro specific activities are in the range of 50 pkat (mg protein)$^{-1}$ and 117 pkat (mg protein)$^{-1}$, respectively[39]. This differs greatly from OPR3, which has an in vitro specific activity of 17.8 nkat (mg protein)$^{-1}$[39]. These activities were reflected also by the respective enzymes expressed *in planta*[39]. As expected, reconstitution of JA biosynthesis was obtained using either OPR1 or OPR2 targeted to peroxisomes (Fig. 7). Targeting OPR1 or OPR2 to cytosol and mitochondria, however, differentiated whether the substrate(s) are occurring within other organelles in amounts allowing enzymatic conversion (Fig. 8). Here, complementation of *opr2opr3* with cytosolic or mitochondrial located OPR2 rescued fertility and JA/JA-Ile contents, although the latter to a lower level than occurring in wild type or in OPR3-complemented lines. This might be due to the fact that OPR2 shows high activity towards 4,5-ddh-JA, but the amount of 4,5-ddh-JA produced upon wounding is relatively low (Supplementary

Fig. 11b and ref. 27). In contrast, an overexpression of cytosolic-located OPR1–showing low activity in conversion of OPDA and 4,5-ddh-JA (Supplemental Fig. S11c)–led to partial rescue of fertility only, but not to an increase in wound-induced JA/JA-Ile. Most importantly, mitochondrial-located OPR1 did not rescue fertility and JA/JA-Ile production at all (Fig. 7). This leads to the assumption that the amount of OPDA or 4,5-ddh-JA able to access the mitochondria is not sufficient to result in a detectable conversion to OPC-8 or JA, respectively. Taking together, these results suggest a limited occurrence of endogenously formed OPDA in the cytosol and mitochondria, whereas it might be preferentially compartmentalized in plastids and peroxisomes. In addition, it is tempting to speculate that 4,5-ddh-JA might be the substrate able to translocate between cell organelles, since OPR3 and OPR2 show reasonable activity on 4,5-ddh-JA but OPR1 does not. However, this cannot be completely evidenced by this approach. Nevertheless, the possible translocation of 4,5-ddh-JA correlates with its potential role in signaling or its effect as an electrophilic compound, as demonstrated by Ueda et al.[35].

In conclusion, despite its perception and signaling in non-vascular plants, OPDA itself does not mediate transcriptional changes in the early wound response of Arabidopsis. The induction of a specific transcriptional change occurred exclusively upon exogenous supply but not upon endogenous rise of OPDA, although in both cases OPDA is (partially) converted to 4,5-ddh-JA[27,35]. Here, the endogenous levels of 4,5-ddh-JA – being low upon wounding, but high upon exogenously applied OPDA – might be determining, whether it acts as an electrophilic (toxic) compound inducing expression of stress-related genes. In turn, the data obtained after wounding suggest that the endogenous OPDA is compartmentalized and thereby its level in the cytosol highly regulated. The discrepancies between the effects of endogenously produced and exogenously supplied OPDA suggest an absence of signaling per se and/or a tightly regulated compartmentalization to prevent its effects as an electrophilic species. Both scenarios imply that OPDA does not inherently function as a signaling compound in the early wound response of Arabidopsis or when occurring in basal levels at control condition, unless such functionality manifests under specific, yet uncharacterized stress conditions or developmental cues. It is plausible that OPDA-oxidation by JASMONTE-INDUCED DIOXYGENASE1 (JID1)[52], or conjugation to amino acids[21,22] or GSH[20] contribute to the regulation of the cytosolic OPDA level.

## Methods

### Plant material, growth conditions, and treatment
*Arabidopsis thaliana* wild-type (ecotype Col-0) and the mutant lines *opr2opr3*[27], *aos* (*dde2-2*[25], *coi1-16*[53] and *coi1-30*[27] were genotyped using the primers listed (Supplementary Table 3). After surface sterilization with 4% bleach, ten to twelve seeds were sown into 2 mL liquid Murashige and Skoog (MS) medium (pH 5.7, 1% sucrose) per well in a 24-well tissue culture plate (TPP, 92424), stratified at 4 °C for three days and grown for ten days. Seedlings from several wells were pooled to

form one biological replicate. Adult plants were grown individually in pots containing steam-sterilized clay, coir fiber, and vermiculite for four weeks. Three rosettes were pooled to form one biological replicate. Plant growth was conducted in Phytocabinets (Percival Scientific, www.percival-scientific.com/) at a light intensity of 120 µE m$^{-2}$ s$^{-1}$ under short day conditions (10/14 h light/dark cycle), at 21/19 °C and 65% relative humidity. For wounding treatments, clustered seedlings were squeezed eight times with moderate pressure using forceps with serrated teeth, while a single wound was inflicted in the midrib of all leaves of the adult rosettes. Phytohormone treatments were done by adding either 25 µM OPDA (≥ 95%, Cayman Chemical, www.caymanchem.com) or 1 µM MeJA to the medium for seedlings, while adult plants were sprayed with the compounds.

### RNA isolation, RNA-seq and quantitative RT-PCR analysis

Total RNA was isolated from homogenized frozen material using the RNeasy Plant Mini Kit (Qiagen) and treated with the DNA-free™ DNA Removal Kit (Invitrogen, #AM1906). First strand cDNA synthesis from 1 µg DNA-free RNA was carried out using the RevertAid H Minus reverse transcriptase with oligo(dT)18 primers (ThermoFisher Scientific™, www.thermofisher.com). Quantitative PCR was carried-out in Hard-Shell® 96-Well PCR Plates (Bio-Rad Laboratories, www.bio-rad.com, #HSP9601) supplied with 10 µL reaction mix of 1.5 ng/µL cDNA, 1x EvaGreen QPCR Mix II (Bio&Sell, www.bio-sell.de, #BS76.580.0200) and 0.2 µM of forward and reverse primers (Supplementary Table 3). The reactions were run on a CFX Connect Real-Time PCR Detection System (Bio-Rad Laboratories) with denaturation (95 °C for 15 min), amplification (40 cycles of 95 °C for 15 s and 60 °C for 30 s) and melt curve analysis (95 °C for 10 s, 65 °C heating up to 95 °C with a heating rate of 0.05 °C s$^{-1}$). Gene expression was normalized to the housekeeping gene *PROTEIN PHOSPHATASE 2 A SUBUNIT A3* (AT1G13320)[54,55] using the $2^{-\Delta CT}$ method[56] and included biological triplicates.

### Transcriptome analysis

RNA quality and integrity were assessed on Agilent 2100 Bioanalyzer system (Agilent Technologies, www.agilent.com) using the RNA 6000 Nano Kit for standard RNA sensitivity (Agilent, #5067-1511). Three biological replicates were submitted to Novogene (www.novogene.com) for mRNA paired-end short-read sequencing (150 bp length) on an Illumina NovaSeq 6000 Sequencing System and bioinformatics analysis according to their pipeline. Reads were mapped to the *A. thaliana* TAIR10 reference genome. Gene expression levels were determined using the FPKM (Fragments per kilobase per million) method. Gene expression heatmaps, principal component analysis (PCA), gene clustering, and Gene Ontology (GO) enrichment analysis were generated using the iDEP.96 (http://bioinformatics.sdstate.edu/idep96/), iDEP 1.1 (http://bioinformatics.sdstate.edu/idep11/) and ShinyGO 0.80 (http://bioinformatics.sdstate.edu/go/) software tools[57,58].

### Protein extraction and immunoblotting

Proteins were extracted by incubation of 50 mg frozen material in 200 µl of extraction buffer (25 mM Tris-Cl pH 6.8, 1% SDS, 1% [v/v] β-mercaptoethanol) at 95 °C for ten minutes. Extracts were treated with 1% (v/v) Halt™ Protease and Phosphatase Inhibitor Cocktail (ThermoFisher Scientific™, #78440) and quantified at 595 nm on the SPARK® multimode microplate reader (TECAN) using the Pierce™ Bradford Plus Protein Assay Kit (ThermoFisher Scientific™, #23236). Ten micrograms of total protein were incubated at 96 °C for 10 min in Laemmli Buffer (1:1)[59] to dissolve AOC trimers[60]. Proteins were separated on SDS–PAGE (4% and 12% acrylamide for stacking and resolving gels, respectively), transferred to a PVDF membrane and detected using Ponceau S staining or No-Stain™ Protein Labeling Reagent (ThermoFisher Scientific™, #A44449). Membrane blocking in 5% (w/v) BSA in TBST (20 mM Tris–Cl pH 7.8, 150 mM NaCl, 0.05% [v/v] Tween) was followed by immuno-staining using anti-AtAOC primary antibody

(1:5000[28],) and a goat anti-rabbit IgG secondary antibody conjugated with alkaline phosphatase (1:4000, Chemicon®, Sigma-Aldrich, www.sigmaaldrich.com, #AP307P). Detection by chemiluminescence with Immun-Star AP Substrate (ThermoFisher Scientific, #1705018) was visualized using a Fusion FX Imaging system (Vilber, www.vilber.com). AOC protein bands were quantified relative to total protein loadings using densitometry analysis with ImageJ (https://imagej.nih.gov/ij/index.html).

### Cloning of the OPRs and plant transformation

The OPR3 (1176 bp), OPR2 (1125 bp), and OPR1 (1194 bp) coding sequences (CDS) without stop codons were PCR-amplified from Col-0 cDNA using the Q5® High-Fidelity DNA Polymerase (Bio Labs, www.biolabs.io, #M0491) and the listed primer pairs (Supplementary Table 4). For OPR3ΔSRL, the last nine nucleotides were removed, and silent mutations C205G and G1109C were introduced to eliminate *BsaI* and *BpiI* sites, respectively. The CDS were cloned into the Golden Gate level 0 vectors pAGM1287 and pAGM1299[61]. OPR fusions to YFP were generated by in-frame assembly of the level 0 with YFP at the C-terminus connected by a Gly-Ser linker. The final cloning cassettes containing the CaMV *35S* promoter and *tOcs* terminator in addition to the Oleosin-RFP plant selection marker were assembled in the pAGM55171 vector.

Plant stable transformation was carried-out by floral dip with *Agrobacterium tumefaciens* strain *GV3101* according to [62]. To circumvent the *opr2opr3* mutant sterility, MeJA was applied to flower buds from 48 h post-dipping until the first silique formation. Transformant selection relied on the seed coat RFP marker and single-insertion lines were identified by segregation analysis determining the ratio of fluorescent to non-fluorescent seeds.

### Subcellular targeting of the OPRs

Organelle variants of OPR3, OPR2, and OPR1 were generated by adding organelle targeting signals to the N- and/or C-terminal regions of the OPRs and/or YFP sequences (Supplementary Table 2). In case of OPR3, all organelle-targeted variants were created using OPR3ΔSRL, whereas for OPR2 and OPR3 the full-length proteins were used. To direct the OPR enzymes exclusively into the cytosol, a nuclear export signal was added (see Supplementary Table 2).

### Transient expression in *N. benthamiana* leaves and protoplasts

*A. tumefaciens* strain *GV3101*, harboring OPR constructs, was cultured for 48 h in LB medium with the appropriate antibiotics. The bacteria were syringe-infiltrated into *N. benthamiana* leaves according to ref. 63. Leaf disc imaging was performed 48–72 h post-infiltration.

Protoplasts were isolated from leaves of 4-week-old *N. benthamiana* plants, as described[64]. Isolated protoplasts were co-transfected with level 1 plasmids of OPR3 constructs and organelle markers[38] using a mixture of 200 µl of protoplast suspension and 10 µg of DNA. Protoplasts were incubated at room temperature overnight after PEG-transformation and used for imaging.

### Confocal microscopy

Confocal microscopy utilized LSM880 and LSM900 laser scanning microscopes (Zeiss Germany, www.zeiss.de). Fluorophores were excited with 514 nm (YFP) and 561 nm (mCherry) laser lines and detected at 510–560 nm and 570-650 nm, respectively. Image acquisition and processing were performed using ZEN software (version 3.4, 2021, Zeiss).

### Phytohormone measurements

Measurements of OPDA, *dn*-OPDA, JA, and JA-Ile were performed using a standardized Ultra-performance liquid chromatography–tandem Mass Spectrometry (UPLC– MS/MS)-based method[65]. 50 mg of powdered frozen tissue were extracted with 500 µl 100% LC-MS methanol

supplemented with 50 ng of [²H₅]OPDA, [²H₆]JA, and [²H₂]JA-Ile each as internal standards. After solid-phase extraction on HR-XC (Chromabond, Machery-Nagel, www.mn-net.com), 10 µl of the eluate were analyzed via UPLC–MS/MS, and analyte content was determined relative to the internal standard peak heights. *dn*-OPDA content was determined using [²H₅]OPDA as the internal standard.

## Determination of enzymatic activities of OPRs

OPR CDS inserted in pET28a(+) with *Eco*RI and *Not*I cloning sites was ordered from BioCat GmbH (www.biocat.com) and transformed into *Escherichia coli* BL21 Star (DE3) cells (ThermoFisher Scientific) by heat shock. Protein expression was performed in auto-induction medium at 16 °C for three days[66]. Harvested cells were resolved in buffer HisA (50 mM Tris/HCl pH 7.8, 100 mM NaCl, 10% (v/v) glycerol, 2 mM MgCl₂ and 5 mM imidazole) supplemented with lysozyme, DNAse I and 0.2 mM PMSF. After a 30-minute incubation on ice, cells were sonicated and cleared lysate was applied to 1 mL HisTrap FastFlow column (Cytiva, www.cytivalifesciences.com). Protein was eluted with buffer HisB (50 mM Tris/HCl pH 7.8, 100 mM NaCl, 10% (v/v) glycerol, 2 mM MgCl₂ and 500 mM imidazole), which was changed to storage buffer (50 mM Tris/HCl pH 7.8, 100 mM NaCl, 10% (v/v) glycerol, 2 mM MgCl₂) by dialysis (MWCO 12-14 kDa) at 4 °C overnight. Following SDS-PAGE, protein fractions were used for in vitro assays at 0.1 mg/mL together with 0.05 mM substrate, 1 mM NADPH and 10 mM NaCl in 20 mM Tris/HCl pH 7.8. Reactions were incubated at 30 °C for 10 minutes and stopped by one volume acetonitrile. Enzyme deactivated by one volume acetonitrile was used as control. Assays were analyzed via UHPLC-high resolution (HR)-MS according to[67]. Data was analyzed using the MassHunter Qualitative Analysis 10.0 software (Agilent Technologies).

## Statistics and Reproducibility

For all transcript/transcriptome analyses, three independent biological samples have been analyzed, three to five samples were used for hormone analyses. Except for RNA-seq experiments, all experiments were repeated at least two times. No statistical method was used to predetermine sample size. No data were excluded from the analyses. Seedlings/plants were grown in a randomized design, all extractions (RNA, jasmonates) and follow-up analyses of extracts as well as microscopical analyses of transient or stable transformed plants were done blinded. Micrographs and photographs were processed through PHOTOSHOP 12.0.4 (Adobe Systems, http://www.adobe.com). The statistical analyses applied to the different datasets as indicated in the figures were performed using GraphPad Prism (www.graphpad.com).

## Reporting summary

Further information on research design is available in the Nature Portfolio Reporting Summary linked to this article.

## Data availability

The transcriptomic raw data generated in this study have been deposited in Sequence Read Archive (SRA) of NCBI (PRJNA1088739 [https://www.ncbi.nlm.nih.gov/bioproject/PRJNA1088739] and https://www.ncbi.nlm.nih.gov/bioproject/PRJNA1105938). Source data for Figs. 1–4 and 6–7, and Supplementary Figs. 1, 4, 5d, 6a, 7d-e, 10 and 12 are provided as a Source Data file. All other data that support the findings of this study are available from the corresponding author upon request. Source data are provided with this paper.

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

## Acknowledgements

We thank Hagen Stellmach (IPB Halle, Germany) for help in jasmonate quantification, Andreas Schaller (University of Hohenheim, Germany)

and Mats Hamberg (Karolinska Institute, Sweden) for providing seeds of *opr2opr3* and 4,5-ddh-JA, respectively. We thank Sylvestre Marillonnet (IPB Halle) for providing the Golden Gate modules and vectors used in this study. Khabat Vahabi (IPB Halle) is acknowledged for assistance in RNAseq preparation and data analysis. Claus Wasternack (IPB Halle) and Alain Tissier (IPB Halle) are acknowledged for critical reading of the manuscript. We thank Prof. Minoru Ueda (Tokhoku University, Japan) for sharing his data and helpful discussions on our manuscript. K.M. and B.H. were supported by the Deutsche Forschungsgemeinschaft (DFG, German Research Foundation) grant No 400681449/GRK2498. M.K. was supported by the DFG grant No 273134146/GRK 2172. I.F. acknowledges funding from the DFG (GRK 2172-PRoTECT, INST 186/1434-1, and ZUK 45/2010).

## Author contributions

Conceptualization, B.H. and K.M.; Methodology, K.M., R.B., and F.S.; Investigation, K.M., R.B., F.S., and M.K.; Analyzing Data, K.M., I.F. and B.H.; Writing–Original Draft, K.M.; Writing–Review & Editing, B.H., K.M., and I.F.; Funding Acquisition, B.H. and I.F.

## Funding

## Competing interests

The authors declare no competing interests.
