## [Transparent Peer Review file · Nature Communications]

Transcriptomics and trans-organelle complementation reveal limited signaling of 12-*cis*-oxo-phytodienoic acid during early wound response in Arabidopsis

Corresponding Author: Dr Bettina Hause

A version of this paper was originally rejected for publication by Nature Communications, however that decision was reconsidered after appeal by the authors.

Version 0:

Reviewer comments:

Reviewer #1

(Remarks to the Author)

12-*cis*-oxo-phytodienoic acid (OPDA), a major precursor of the plant growth and defense hormone JA-Ile, is thought to have unique signaling functions in stress responses. In this study, the authors used the Arabidopsis mutant lines *opr2opr3* and *aos* to analyze the wound-induced transcriptome compared to the wild type to determine the signaling function of OPDA. *opr2opr3* and *aos* mutants showed different gene expression profiles compared to the wild type, indicating that OPDA has a limited effect on gene expression. The authors showed that OPDA does not have its own transcriptional signature. Known OPDA-responsive genes were induced by wounding independently of OPDA; OPR3 complemented the fertility and wound-induced JA-Ile production of the *opr2opr3* mutant, regardless of localization. The results also suggest that OPDA is strictly compartmentalized within the peroxisome and has no signaling function. The authors show that OPDA has no unique signaling function in Arabidopsis beyond its role as a JA precursor, highlighting the central role of JA in the plant injury response. While the content of this paper is very interesting and has the potential to be an important achievement in a related field. However, the current data in this manuscript are still insufficient to warrant the conclusion shown by the authors. I hope you will find the following comments helpful.

Major concerns

- 1) The authors concluded that JA-Ile, biosynthesized from OPDA, is the principal active signal. To clarify this, it is essential to conduct experiments to verify the effect of OPDA on the *coi1-1* mutant.
- 2) In Taki et al. Plant Physiol 2005, many genes are shown as OPDA-markers, however, the authors discuss a limited number of the major markers and some of the minor markers. In the authors' experiments, the expression of ZAT10 (Fig. 2d), used as an OPDA-marker in many papers, shows apparent differences between *opr2opr3/aos* and is clearly JA-independent and OPDA-dependent. Based on these results, it is unlikely that OPDA-marker gene expression is JA-dependent based on the current data.
- 3) The authors have ruled out the possibility that OPDA is released outside the peroxisome using a well-designed OPR enzyme localization system. However, the recently reported presence of OPDA-amino acid conjugates (Flokova et al. Phytochemistry, 2016 and Mik et al. Phytochemistry, 2023) indicates the possibility that OPDA is released outside the peroxisome. The authors' views in this regard should be described.
- 4) Although many experiments have been conducted to verify the effects of externally administered OPDA using various mutants, this is not an accurate analysis because it is impossible to distinguish from the endogenous background unless accurate conversion experiments using deuterium-labeled compounds are conducted. Exogenously applied OPDA also induces gene expression, but according to the authors, OPDA has a clear compartmentation and does not leak externally. How much of OPDA is converted to JA?
- 5) Line 182-183 The authors described that the difference in gene expression between *opr2opr3* and *aos* is slight, but based on FigS4b, it is likely that there are more differently expressed genes (DEGs). Rather than processing data based on *opr2opr3/aos*, I would like to propose an accurate quantification based on RT-qPCR analysis of each gene expression.
- 6) Line 230-233 It is impossible to derive this statement from the current data. It is only a hypothesis unless we do experiments with *opr1* mutant lines, thus, this should be moved to Discussion.

7) Line 236-240 Could these be side effects of exogenously applied OPDA? The possibility that there are other main effects cannot be ruled out.

8) Lines 288-291 After conversion to OPC-8, does OPC-8 undergo β -oxidation at the peroxisome? And is there evidence for the existence of an OPC-8 transporter? An experiment in which deuterated α -linolenic acid is externally administered (Chini et al. Nat. Chem. Biol. 2018) may be useful to show that endogenous OPCDA is converted to JA-Ile in this experiment.

9) Lines 320-326 In order to draw this conclusion, the use of opr2opr3 double mutant is essential, and the current data are insufficient to draw this conclusion.

Minor concern

1) lines 138-144 It is difficult to follow what the authors are trying to say. Since aos mutants cannot biosynthesize allene oxide, is it evident that they cannot synthesize OPDA even in the presence of AOS protein?

2) Fig 1d It is strange that when wounding is not applied, AOC protein is upregulated, but aos gene expression is not.

Reviewer #2

(Remarks to the Author)

The manuscript addresses an important debated question in the field related to a putative role of OPDA in the activation of wound responses independent of its conversion in JA/JA-Ile.

Firstly, the authors undertook a transcriptomic approach to study the response to wounding of mutants deficient in OPDA and JA/JA-Ile (aos) or deficient in JA/JA-Ile but accumulating OPDA (opr2opr3). The transcriptomic results show very little or no differences at all between these mutants in their response to wounding.

Secondly, the authors show that exogenous treatment with OPDA, rather than the endogenous molecule is responsible for the OPDA responses described previously.

Finally, to clarify if OPDA or 4,5-ddh-JA is the mobile signal between cellular compartments, they took a trans-organelle complementation approach of OPR isoenzymes in opr2opr3 mutant background. This part is very interesting and elegant, and authors conclude that the mobile molecule is 4,5-ddh-JA.

Despite addressing an important question and the elegance of the approach, the conclusions are not fully supported by the results and there are a few issues that need further clarification.

Major issues:

1- Despite authors' assumption, the double opr2opr3 mutant is not a complete depletion of JA since OPR1 is still active in the cytosolic pathway. Similarly, aos can also produce some OPDA by the non-enzymatic pathway and JA. I agree with the authors that in their measurements the levels of JA in the mutants are very low. However, if they want to test a putative role of OPDA independent of JA, the use of "leaky" mutants is not the cleanest option. This is particularly relevant, as the authors introduce, when the precedent of using the "leaky" opr3 has caused a wealth of wrong conclusions in the field. Instead of leaky mutants, if authors want to test a putative JA-independent (=COI1-independent) role of OPDA, they should use the coi1 mutant. Does coi1 accumulate OPDA after wounding? The use of coi1 would also avoid over-manipulation of the plants with the MeJA treatment. At least, the coi1 results would complement/confirm those presented here.

2- Transcriptomic analyses: Although pairwise comparisons show scarce differences between opr3opr2 and aos mutants, the Venn diagrams in Figs 2A and Fig S5B show an important difference (108+173 and 211+406 genes). Since pairwise comparisons do not take into account basal levels, a better analysis such as WGCNA could arise significant differences. Additionally, patterns shown in Fig S3a would suggest that JA/JA-Ile have a very little role in wound-induced gene expression, since only a few genes differ between WT and mutants. However, the likely answer to this issue is that the mutants are leaky and, therefore, the use of coi1 is important. This explanation also applies to Fig S4 for the induction of JA-dependent genes in the mutants.

3- The major issue in this work, however, is the assumption that "in vivo" the specificity of OPR1 and OPR2 for 4,5-ddh-JA and OPDA would be similar to their specificity "in vitro". It is clear that the double opr2opr3 still accumulates some JA, which supports that "in vivo" OPR1 either reduce OPDA or 4,5-ddh-JA. The fact that OPDA can be in the cytoplasm is obvious by its "travel" from the chloroplasts to the peroxisomes, and also by previous works demonstrating that OPDA is conjugated with aminoacids, or oxidized by the recently identified JID1.

Minor points:

-Intro: "Lower plants" is not a correct term. It is better to use "non-vascular plants" or other more appropriate term

-Intro: The "leakiness of the opr3" was first demonstrated by Chehab et al., 2011.

-Results: Page 6: Have the authors tested the levels of AOC after wounding?

-Supp Fig 5a: Differences are not statistically significant in non-wounded plants. This argues against the MeJA treatment.

-Supp Fig 6: The result is not so surprising since, as shown in Fig 4B, such a high concentration of OPDA induces the accumulation of JA-Ile in the mutants. Again, the use of coi1 would prevent this type of results.

-Fig 4c: It is surprising the low amount of cis-OPDA after OPDA treatment. Is this result correct? Why is the plant not accumulating the OPDA from the treatment?

-Fig 5c-g: I couldn't find these panels

-Author's data suggest that the JA-independent response after wounding is related to general stress. Have authors considered that other stresses different than wounding may induce higher levels of OPDA, which may induce an OPDA-specific response?

-Are expression levels of transgenics in the complementation assays comparable?

- In figures 6a-h and 7c, is there any possibility to quantify the phenotype/recovery of the fertility?
- In the discussion, dn-OPDA is described as "a beta-oxidation product of OPDA". Actually, the major source of dn-OPDA is 16:3, and the beta-oxidation of OPDA contributes very little.

Version 1:

Reviewer comments:

Reviewer #2

(Remarks to the Author)

In this revised version authors have addressed some of my concerns, but still important issues remain unsolved. The most important one is again the assumption of OPRs substrate specificity based on previous published results. The authors refer to a previous work (Chini et al., 2018) to support that:

- 1- 4,5-ddh-JA is not a substrate of OPR1
- 2- Wound induced JA accumulation was only slightly reduced in opr3opr1 compared with opr3
- 3- Chini et al., conclude that OPR1 is not involved in 4,5-ddh-JA reduction.

I have carefully revised this work (Chini et al., 2018) and found that these conclusions cannot be drawn from this paper for the following reasons:

- 1- Fig 6D in Chini et al., only shows that OPR1 does not work in this particular assay. However, there is no proof that this enzyme works at all. Since this is a purified enzyme in vitro, it may well be that the protein is just not functional. Therefore, if the authors want to support their conclusion they should demonstrate that OPR1 is functional and can reduce OPDA but not 4,5-ddh-JA
- 2- If "wound induced JA accumulation was slightly reduced in opr3opr1" this mean that OPR1 uses 4,5-ddh-JA as a substrate. The small (statistically significant) reduction is likely a consequence of OPR2 being more active
- 3- "Chini et al., conclude that OPR1..." This sentence is not true. I have looked very carefully through Chini et al. and they never concluded that OPR1 is not involved in 4,5-ddh-JA. Actually, they conclude the opposite; that both OPR1 and OPR2 use 4,5-ddh-JA as a substrate, although OPR2 is much more active than OPR1. There are several places in the manuscript where Chini et al., explain that OPR1 and OPR2 are both involved in this reduction, with OPR2 being the main player, but a sentence concluding that OPR1 has no role on 4,5-ddh-JA reduction cannot be found in this paper. Therefore, the experiments of transorganellar complementation are based on wrong premises and would require a formal demonstration of the enzymatic specificity of OPR1.

Reviewer #3

(Remarks to the Author)

The manuscript by Mekkaoui et al. addressed the long-debated and important question on whether OPDA functions as an independent signaling molecule. Comparative transcriptomics studies on rapid (1h) wound response of mutants deficient in OPDA and JA/JA-Ile(aos) or deficient in JA/JA-Ile but accumulating OPDA (opr2opr3) show that very few differential expressed genes between these mutants. Furthermore, the authors show that exogenously applied, but not endogenous OPDA is responsible for the expression of OPDA-marker genes described previously. Interesting and elegant transorganellar complementation experiments of OPR isoenzymes in opr2opr3 mutant background were carried out to test if OPDA or 4,5-ddh-JA is mobile between cellular compartments. The revised manuscript has largely addressed the issues raised by the reviewers. The accompanying manuscript by Ueda et al demonstrated that OPDA signaling is independent of the canonical COI1-JAZ-MYC signaling pathway using coi-1, myc2myc3myc4 and opr2-1 opr3-3 Arabidopsis mutant, and that OPDA is not genuine bioactive forms using cts mutant. The common mutants such as opr2-1 opr3-3 and coi-1, and very similar figure (Fig. 1 vs Fig 1A) were used in both manuscript, incorporation of some of their results into this manuscript would consolidate these conclusions.

Major issues:

1. The studies are focused on rapid (1 hour) wound responses. Is there the possibility that OPDA signaling activated during late phases of wounding?
2. I wonder whether expression of OPRs in the nucleus would rescue the male sterility of opr21opr3 mutant.

Minor points:

1. L246, there is no Fig. 4d.
2. Fig. 7d, WT and opr2opr3 control should be added.
3. In Fig. S6, the average content of cis-OPDA and dn-OPDA is significantly different between opr2opr3 and aos mutant, but statistical analyses showed no significant (ns), please check.
4. Fig. S9, why the subcellular localization patterns at the ER (a, b) and cytosol (c, d) are very similar?

Reviewer #4

(Remarks to the Author)

The manuscript "Transcriptomics and trans-organellar complementation reveal a limited signaling capacity of 12-cis-oxo-

phytyldienoic acid in wounded *Arabidopsis*" by Khansa Mekkaoui et al., presents an interesting hypothesis on OPDA compartmentalization and limited signaling properties, contradicting previous reports on (exogenous) OPDA ability to induce gene expression independently on the COI1 receptor.

The authors used an elegant strategy to assess the role of endogenous OPDA using knock-out mutants treated with MeJA and wounding to induce comparable jasmonate levels, as well as a trans-organellar complementation to determine OPDA subcellular localization. Although I appreciate the authors' efforts to investigate the role of OPDA endogenously produced after wounding, I think that including additional controls in their transcriptomic analyses would strengthen their conclusions and clarify potential contradictions with previously published datasets. Unless these controls are included, it is difficult to evaluate whether the experimental conditions are comparable to previous reports, which hinders the interpretation of the results. The presented results do not support the authors' claim that OPDA does not have signaling functions.

Similar to the previous reviewers' comments, I strongly recommend the use of *coi1* mutants and OPDA exogenous treatment to get a comprehensive picture. The newly added results on the expression of some marker genes in *coi1-16* and *coi1-30* mutants is not enough to assess the contribution of JA/JA-Ile and OPDA genes to the massive transcriptional changes observed in wounded Col-0, *opr2opr3* and *aos* plants. I agree with the other reviewers' comments that the number of genes classified as JA/JA-Ile-dependent are strikingly low, even though the JA/JA-Ile-dependent genes constitute a fraction of the wounding response. I found surprising that the JA-dependent wound response (Figure S5) seems to be already activated in Col-0 control plants and only a few of these genes are up-regulated upon wounding in Col-0. The inclusion of *coi1-30* in the RNAseq experiment would be instrumental to differentiate the JA/JA-Ile/COI1-dependent genes during wounding and differentiate them from the OPDA- or wound-induced genes.

Along this line, I also found surprising that exogenous OPDA induced JA genes in *opr2opr3* (Figure S7), which might indicate that *opr2opr3* is not the right tool to assess the role of OPDA independent on JA/JA-Ile/COI1. Is this induction due to OPR1 reducing OPDA in the cytosol in planta? Was wounding not strong enough to induce OPDA? Is it a matter of timing? Why were the wounded plants collected after 1 h and the seedlings treated with OPDA only for 30 min? Again, including *coi1-30* in this experiment (wounded and OPDA-treated plants) would be helpful to clarify which genes are regulated by JA/JA-Ile/COI1 and which genes are regulated by OPDA and/or wounding.

Regarding the OPDA levels, in Figure S4d a wound-induced OPDA increase in *coi1-30* could be expected. Comparison between the wound-induced OPDA accumulation in Col-0 plants as well as the control in Figure S4c and S4d showed remarkable differences between the experiments. I am aware that inflicting mechanical wounding to *Arabidopsis* leaves might vary depending on the person and on the lab. I was wondering whether more intense wounding would result in a more pronounced accumulation of OPDA that would result in regulation of the previously reported OPDA-induced genes. As one of the reviewers pointed out, exogenous treatment with OPDA usually induces a hyperaccumulation of OPDA, whereas in Figure S7d the OPDA accumulation looks relatively similar after wounding or exogenous OPDA treatment. For the sake of comparison, this is a very good point (similar accumulation using two different triggers), but I am not convinced of these conditions being optimal for the proposed analysis. Again, an appropriate control would be OPDA-treated and wounded Col-0 and *coi1-30*.

In Figure S6, the induction with MeJA did not seem to work as in seedlings based on the hormone measurements. Moreover, the PCA plot shows that the samples do not cluster so well. I agree that PC1 shows the variance depending on the treatment, but PC2 does not indicate that the samples group depending on the genotype. This variability might affect the interpretation of the results.

Regarding the complementation of *opr2opr3* by any OPR3 variant (localized in different subcellular compartments), I would not necessarily interpret such results as compartmentalization of OPDA in the peroxisome. Alternatively, is it because during the translation of the OPR3 variants they could potentially encounter OPDA in the cytosol? In such case, the trans-organellar complementation might not be feasible with the current approach. I am very intrigued by the complementation of *opr2opr3* by mitochondrial OPRs. What do you think that the OPDA might be doing in the mitochondria? Does OPDA regulate mitochondrial genes? I really like the idea of the trans-organellar complementation using OPRs with different substrate specificity, but there are a few aspects that should be discussed in more depth, such as the differences between enzymatic *in vitro* assays and the activity in planta, or the potential role of OPDA in mitochondria.

Reviewer #5

(Remarks to the Author)

Studying OPDA-specific functions in Angiosperm plants is difficult and full of pitfalls. The revised (v2) manuscript of Mekkaoui et al is an advanced study that reads very well. While many relevant comments have been made previously by two reviewers and consistent changes have been introduced, I will not come back to all solved questions but rather complement with a few additional points.

This is a nice piece of work where a long standing and ever-evolving question is shined with new light, owing to the combination of a smart set of genetic tools along with an elegant complementation strategy. By doing so, the authors delineate precisely the sectors of the wound response that are under the control of JA/JA-Ile, and spectacularly find out that there is no OPDA signature upon wounding. A main message is that some established 'OPDA markers' are not regulated by endogenous OPDA upon wounding. In a second part, they investigate convincingly some compartmentation aspects of the

substrates of the different OPR isoforms and finally propose possible reasons for distinct responses to external OPDA supply.

The zig-zag historic route for knowledge acquisition on the topic of OPDA signaling is well introduced in the ms, taking into account most of the past studies and highlighting their limitations.

Authors treated properly queries on v1 by reviewer#1

#1 : OK with the arguments provided, but maybe reviewer #1 meant analyzing transcriptome in coi1. The coi1 transcriptome would be expectedly similar to those of opr2opr3 and of aos, but to my opinion the interpretation would be blurred by the fact that part of the OPDA produced would be converted to JA/JA-Ile, thus the homeostasis of OPDA is different from opr2opr3. Also, as argued, coi1 has less OPDA and cannot be complemented.

#2: OK

#3: OK

#4: the amounts of OPDA before and after OPDA feeding do not correspond to the cited Fig. 4c. Should be Fig. S7e ?

#5: OK

#7: OK

#8: OK

My own comments:

Abstract

L26 : slightly accumulate JA-Ile ?

L28: endogenous OPDA

L29: were 'found' wound-induced

Results

1. L116: The developmental stage/experimental setup used should be precised at start of results (Fig. S1), as wounding 10 d-old seedlings is unusual. The in vitro-grown material is therefore composed of whole seedlings (including roots), versus leaf-only for adult plants. Methods should also precise i) number of seeds placed in which type of wells to achieve 120 seedlings; ii) how standardized wounding is applied to 120 liquid-grown seedlings. What about metabolite leaking/dilution from crushed material to medium ?

2. There is an issue with statistical analysis in some instances (eg Fig. 1d, 1e, Sup Fig 6a.): authors discuss in text effects of treatment (for exemple gene inducibility), but statistical significance is shown only between genotypes, not treatments. A two-way ANOVA would be better suited to pinpoint what is significant.

3. L212: It is noteworthy that CYP81D11 is the top upregulated gene by phytoprostane and OPDA in Muller et al (<https://doi.org/10.1105/tpc.107.054809>). Therefore this would be a candidate OPDA marker.

4. L124: AOC is encoded in Arabidopsis by 4 genes (AOC1-4). The authors should precise if all 4 AOC are impacted by this positive feedback loop, and if the antibody used recognizes all 4 isoforms.

5. L170: I am not sure how these values are obtained considering the numbers in Fig. 2a. The number of JA-Ile-dependent DEGs appears substantially lower in 10d seedlings (barely visible in Fig. 2a, compared to a reduction by half in mutant rosettes Fig S6b) than in adult rosettes. Therefore, this sector of the wound-response seems to be age-dependent. Also, can the JA-dependent sector be limited compared to other studies because an early time point (1 h) is analyzed, and early-responding (signaling) transcripts are less JA-dependent, whereas JA-dependent, (defense) related transcripts are regulated at later timepoints ?

6. Fig. 5b shows a nuclear export signal picture to the cytoplasm (NES), not a nuclear localization signal (NLS). Please explain/correct.

7. Fig. S12: how is it explained that OPDA levels are quite fluctuating across the two equivalent lines whereas JA/JA-Ile levels are strikingly stable in the same sibling lines ?

Minor points:

L142: this statement is not fully valid in absence of data for wounding without MJ treatment.

All Figures with bar charts : individual datapoints are barely visible in black on dark grey histograms. As a general comment, colors could be used for bar filling, or lighter tones of background used.

L120: It is not clear if authors mean detection or quantification. As discrete values are given for opr2opr3 wounded, 0.002 nmol/g FW seems like LOQ.

L275: this conclusion applies to a wounded seedling context, but cannot be generalized yet, as signaling properties have

been associated with thermotolerance.

L281: the phenotypic rescue can happen only if the OPR3 product, OPC:8 is produced in a location from where it can access β -oxidation in peroxisomes, possibly by comatose-facilitated import. One could imagine a scenario where a re-located version of OPR3 catalyzes the reduction of occurring OPDA, but that OPC:8 cannot reach peroxisomes and prevent subsequent formation of JA-Ile (for ex can OPDA freely enter and 4,5-ddh-JA exit mitochondria ?). In other words, the restoration of fertility requires more than just OPDA metabolism.

L287: cytosol and nucleus ?

L295: ER-OPR3 and cytosol-OPR3 also label the nucleus

L316: why only these 3 compartments ?

Discussion

L411: At first sight, this contradicts markers used by Ueda et al. This means that they can be induced by exogenous OPDA, but are wound-induced independently of OPDA.

L427: 'conclusion' rather than 'fact'. Here I suggest to be more cautious: OPDA could mediate responses by chemical reactivity, not requiring a unique, specific receptor. Also, the conclusion should be restricted to wounding as other biological situations may lead to different interpretations.

L449: it would be worth adjusting concentration of exogenous OPDA such as 4,5-ddh-JA also reaches 500 pmol/g FW and compared gene inducing activity.

L476: maybe a mistake on OPR2 and OPDA. This contradicts line L336 and ref 21.

L491: in wounded Arabidopsis

What about treating aos with OPDA ? This would show the responses to exogenous OPDA without the context of endogenous OPDA.

opr2opr3 also produces arabidopsides, aos not..

It is unexpected to see that all plant lines expressing an OPR active on 4,5-ddh-JA do complement regardless of the compartment. Does this mean that there is some 4,5-ddh-JA in any cellular compartment ?

Version 2:

Reviewer comments:

Reviewer #3

(Remarks to the Author)

The revised manuscript by Mekkaoui et al. used elegant experiments to reveal the role of endogenous OPDA using opr2opr3 knock-out mutants treated with MeJA and trans-organelle complementation. Their findings on distinction in transcription regulation between endogenous and exogenous OPDA provide new insights into the role of OPDA and highlight the significance of the subcellular compartmentation in its signaling. The revised manuscript has largely addressed the issues raised by the reviewers, thus I recommend to publish the manuscript.

Reviewer #4

(Remarks to the Author)

In the revised version of the manuscript the authors have convincingly addressed the reviewers' concerns. Regarding the induction of JA-regulated genes by wounding in opr2opr3, the authors should correct the following statement:

Line 253-255: "Validation of transcript accumulation of JAZ2, JAZ7, JAZ13 and CHL1 by RT-qPCR confirmed their exclusive induction in opr2opr3 seedlings upon OPDA application but not upon endogenous OPDA accumulation by wounding (Supplemental Fig. 7d)."

This is not correct. The expression of JAZ2 and JAZ7 is induced by wounding in opr2opr3 albeit to a lesser extent than in opr2opr3 treated with OPDA. How do you explain the induction of JA-regulated genes in opr2opr3 after wounding if you consider that the OPDA conversion to JA-Ile is negligible?

Minor comments:

Suppl. Fig. 1B: the y axis lacks part of the labels

Figure 7: The figure will be easier to read if the graphs in panel d showed the same order as the images in a-c: cytosol, peroxisome, mitochondria.

Reviewer #5

(Remarks to the Author)

The authors responded convincingly to my comments.

L32 abstract : 'located' may be replaced with 'targeted'

I have no further comments.

Congratulations for this meticulous work.

Rebuttal letter to the reviews of the manuscript “Transcriptomics and trans-organelle complementation reveal a limited signaling capacity of 12-*cis*-oxo-phytodienoic acid in wounded *Arabidopsis*” (NCOMMS-24-18845)

First, we want to thank the reviewers for the competent throughout evaluation of our manuscript. We are very grateful for their positive overall assessment of our work and appreciate the critical points they have raised. In the following text, we address all the concerns both reviewers had – either by including new data, changing the text or directly replying to their points. All reviewer comments are underlined, our answers appear in normal font.

Reviewer #1:

12-*cis*-oxo-phytodienoic acid (OPDA), a major precursor of the plant growth and defense hormone JA-Ile, is thought to have unique signaling functions in stress responses. In this study, the authors used the *Arabidopsis* mutant lines *opr2opr3* and *aos* to analyze the wound-induced transcriptome compared to the wild type to determine the signaling function of OPDA. *opr2opr3* and *aos* mutants showed different gene expression profiles compared to the wild type, indicating that OPDA has a limited effect on gene expression. The authors showed that OPDA does not have its own transcriptional signature. Known OPDA-responsive genes were induced by wounding independently of OPDA; OPR3 complemented the fertility and wound-induced JA-Ile production of the *opr2opr3* mutant, regardless of localization. The results also suggest that OPDA is strictly compartmentalized within the peroxisome and has no signaling function. The authors show that OPDA has no unique signaling function in *Arabidopsis* beyond its role as a JA precursor, highlighting the central role of JA in the plant injury response. While the content of this paper is very interesting and has the potential to be an important achievement in a related field. However, the current data in this manuscript are still insufficient to warrant the conclusion shown by the authors. I hope you will find the following comments helpful.

Major concerns

1) The authors concluded that JA-Ile, biosynthesized from OPDA, is the principal active

signal. To clarify this, it is essential to conduct experiments to verify the effect of OPDA on the *coil-1* mutant.

It is well-known that JA-Ile is the best characterized jasmonate compound mediating alterations in gene expression. This we did not want to question. Instead, we asked the question, whether OPDA, the main intermediate of JA biosynthesis, has signaling capacity in Arabidopsis seedlings and rosette leaves upon wounding. To do so, we used wild type (Col-0), which is able to synthesize OPDA, JA and JA-Ile, the double mutant *opr2opr3*, which shows almost undetectable JA levels upon wounding (see Chini et al., 2018, Figure 6a), and *aos*, a mutant lacking ALLENE OXIDE SYNTHASE (single copy gene in *A. thaliana*) and therefore deficient in OPDA and JA/JA-Ile. To our opinion, the use of these three genotypes should allow to distinguish between the role of the different metabolites in wound-induced gene expression. Nevertheless, we followed the advice of this and the second reviewer and analyzed levels of OPDA, JA and JA-Ile in *coil*-mutants upon wounding. Plants of two different *coil*-alleles showed an increase of OPDA and JA-levels upon wounding, but these levels were around 25 % of that of the WT possibly due to the missing positive feedback in JA-biosynthesis (Scholz et al., 2015; Wasternack and Song, 2017). This was similar to *opr2opr3*, which also showed lower levels (Fig. S1). While it was possible for *opr2opr3* to perform a kind of rescue to get wild type OPDA levels, *coil* mutants are insensitive and cannot be rescued. This limits the use of *coil*-mutants to get a clear (putative) response to OPDA in terms of altered gene expression.

Nonetheless, we tested several of the genes shown as OPDA-markers by (Taki et al., 2005) upon wounding of *coil-16* and *coil-30* (see below). Their induction in *coil-1* after application of OPDA was checked by Ueda et al. in a complementary manuscript, which will be submitted in parallel. They showed that these genes are induced by application of OPDA in a COI1-independent manner.

2) In Taki et al. Plant Physiol 2005, many genes are shown as OPDA-markers, however, the authors discuss a limited number of the major markers and some of the minor markers. In the authors' experiments, the expression of ZAT10 (Fig. 2d), used as an OPDA-marker in many papers, shows apparent differences between *opr2opr3/aos* and is clearly JA-independent and OPDA-dependent. Based on these results, it is unlikely that OPDA-marker gene expression is JA-dependent based on the current data.

The major OPDA-responsive genes (ORGs) identified by (Taki et al., 2005) consist of six genes: *ZAT10*, *ERF5*, *DREB2A*, *GST6*, *FAD-OXR*, and *TCH4*. In our study, we selected primary three out of these six genes for qRT-PCR validation, along with another major OPDA-marker, *GRX480* (Park et al., 2013; Arnold et al., 2016). Now, we included the three missing ones (*GST6*, *ERF5* and *TCH4*). While *ZAT10* exhibited slightly higher expression levels in Col-0 and *opr2opr3* compared to *aos* in the RNAseq data, it did not surpass the commonly used fold-change (FC) cutoff of 2 to be classified as a differentially expressed gene (DEG). It is important to note that while our interpretation of the data differs from that of (Taki et al., 2005), our results do not fully contradict theirs. They tested a few of the OPDA-markers in the *aos* mutant, and these markers were induced by wounding despite the absence of OPDA in that mutant (see Figure 4 in Taki et al., 2005). Therefore, based on our data and that of Taki et al., (2005), this suggests that these markers may not be reliable OPDA markers as we aim to clarify OPDA function in Arabidopsis.

To search for (more) OPDA-induced genes, we performed the transcriptomic approach that should identify genes, which will be induced/repressed in WT and *opr2opr3*, but not in *aos*. We think, the comparison between *opr2opr3* and *aos* is the most important one, because these both genotypes differ mainly in the occurrence of OPDA. Therefore, genes induced by wounding in all three genotypes represent generally wound-induced, but JA- and OPDA-independent genes. Even the known OPDA-markers belong to this group (as shown in Fig. 2c, d). Therefore, we agree completely with the reviewer that the wound-induced expression of these selected known OPDA-markers is unlikely to be JA-dependent. Their expression is neither JA nor OPDA dependent as evidenced by the respective transcript accumulation in the *aos* mutant. In addition, our RNAseq data revealed 21 other ORGs from Taki et al. (2005) as wound-responsive, but independently from JA and OPDA signaling (see table below, now included as Supplementary Tab. 4 into the manuscript). In contrast, DEGs only detectable in wild type represent JA-dependent genes, such as shown for *CLH1* and *NATA1*, which are not induced by wounding in *opr2opr3* and *aos* (Fig. S5d).

Regarding the whole set of ORGs identified by Taki et al. (2005), the selection of the ORGs was based on the treatment of plants with OPDA and JA. Additionally, several ORGs selected were chosen with fold-change (FC) cut-offs of less than 1 and 0.5 FC under wounding condition, as shown in Table I and Table II of their study. In our dataset, we identified DEGs in response to wounding at the 1-hour time point using an FC cut-off of 2 (logFC=1, p-value

0.05). Due to these differences in treatment and FC cutoffs, it is to be expected that our dataset does not entirely overlap with that of (Taki et al., 2005).

Table S4: Genes selected from Taki et al. 2005 showing a similar wound-induced induction or repression in Col-0, *opr2opr3* and *aos*.

AGI	Gene name	LogFC			Adjusted p-value			Description
		Col-0	opr2opr3	aos	Col-0	opr2opr3	aos	
Up regulated genes								
AT5G42380	CML37	5.09	4.94	4.76	2.2E-13	5.3E-13	6.3E-13	Calcium-binding protein CML37
AT3G25250	OXI1	3.76	3.84	3.53	1.9E-11	2.1E-11	4.9E-11	Serine/threonine-protein kinase OXI1
AT5G59820	ZAT12	3.54	3.50	3.29	4.3E-10	6.3E-10	1.2E-09	Zinc finger protein ZAT12
AT5G35735	-	3.42	3.73	3.62	3.2E-11	1.7E-11	2.1E-11	Cytochrome B561
AT1G61340	FBS1	3.40	3.44	3.52	1.2E-10	1.4E-10	1.0E-10	F-BOX stress induced 1
AT2G46400	WRKY46	3.14	3.31	3.25	2.6E-09	1.9E-09	2.1E-09	Probable WRKY transcription factor 46
AT3G04640	-	3.11	3.01	3.11	9.7E-10	1.8E-09	1.2E-09	Glycine-rich protein
AT2G22500	PUMP5	2.89	2.74	2.82	1.4E-11	3.3E-11	2.2E-11	Mitochondrial uncoupling protein 5
AT3G46080	ZAT8	2.53	2.55	2.34	1.3E-08	1.5E-08	3.6E-08	Zinc finger protein ZAT8
AT1G66090	-	2.49	2.71	2.56	1.4E-08	6.4E-09	1.1E-08	Disease resistance protein (TIR-NBS class)
AT4G24570	DIC2	2.24	2.12	1.83	1.2E-09	2.9E-09	1.4E-08	DICARBOXYLATE CARRIER 2
AT5G27420	ATL31	2.18	1.93	2.00	9.0E-08	4.0E-07	2.6E-07	E3 ubiquitin-protein ligase ATL31
AT4G24160	-	2.01	2.43	2.30	1.0E-10	1.8E-11	3.0E-11	1-acylglycerol-3-phosphate O-acyltransferase
AT5G57560	XTH22	1.69	1.57	1.42	1.7E-08	4.6E-08	1.3E-07	Xyloglucan endotransglucosylase/hydrolase
AT1G15520	ABCG40	1.64	1.69	1.33	6.6E-07	5.4E-07	7.2E-06	ABC transporter G family member 40
AT5G54490	PBP1	1.51	1.70	1.48	3.8E-07	1.2E-07	5.2E-07	Pinoid-binding protein 1
AT1G59660	NUP98B	1.49	1.20	1.44	2.0E-08	2.6E-07	3.2E-08	Nuclear pore complex protein NUP98B
AT1G59860	HSP17.6A	1.48	1.25	1.20	6.4E-06	4.3E-05	6.0E-05	17.6 kDa class I heat shock protein 1
AT1G16030	HSP70-5	1.35	1.37	1.41	2.5E-04	2.6E-04	1.9E-04	Heat shock protein 70-5
AT5G45340	CYP707A3	1.26	1.64	1.56	6.0E-04	6.2E-05	9.8E-05	Abscisic acid 8'-hydroxylase 3
Down regulated genes								
AT1G78000	SULTR1;2	-1.70	-1.35	-1.01	7.4E-07	9.8E-06	1.7E-04	Sulfate transporter 1.2

3) The authors have ruled out the possibility that OPDA is released outside the peroxisome using a well-designed OPR enzyme localization system. However, the recently reported presence of OPDA-amino acid conjugates (Flokova et al. Phytochemistry, 2016 and Mik et al. Phytochemistry, 2023) indicates the possibility that OPDA is released outside the peroxisome. The authors' views in this regard should be described.

Using the transorganellar complementation approach, we showed that OPDA might be restricted to the organelles of its biosynthesis (plastids) and metabolism (peroxisomes). An occurrence of OPDA in the cytosol cannot, however, be excluded, since there has to be a transfer from the plastids to the peroxisomes via the cytosol. While the use of the term “OPDA sequestration” in our phrasing may have been interpreted as implying the absence of OPDA in the cytosol, we state that OPDA is present in the cytosol but in a limited amount (see lines 469-471, 484-487, 1075-1076). We argue that the amounts of OPDA within the cytosol are rather minor due to the following facts: (i) OPR1 located in the cytosol led to a partial rescue of the mutant, while its peroxisomal counterpart led to a full complementation (Fig. 7b-c). This partial complementation supports a limited presence of OPDA in the cytosol but does not imply its complete absence. (ii) OPDA-amino acid conjugates (Floková et al., 2016) were detected at very low levels. The highest levels of *cis*-OPDA-Ile upon wounding is 1.46 ± 0.22 pmol/g FW compared to $15,493 \pm 1339$ *cis*-OPDA (Floková et al., 2016) and 1-10 pmol/g FW of other amino acid conjugates were detected in comparison to 25 nmol/g FW of OPDA (Mik et al., 2023). Therefore, our results align with the very low detected levels of OPDA conjugates, both indicating limited amounts of cytosolic OPDA in comparison to the total detectable OPDA amounts in the tissue. (iii) As a precursor of the bioactive JA-Ile, the transport of OPDA through the cytosol is likely tightly regulated, as indicated by its export from chloroplasts via JASSY (Guan et al., 2019) and import to the peroxisome via the ABC transporter COMATOSE (CTS) (Theodoulou et al., 2005; Ueda et al., accompanying manuscript). Moreover, as a reactive electrophile species, OPDA is relatively cytotoxic (Alm eras et al., 2003; Mueller et al., 2008), and its detoxification through conjugation with GSH or its metabolism by a dioxygenase (JID) has been demonstrated in Arabidopsis and tobacco (Mueller et al., 2008; Ohkama-Ohtsu et al., 2011; Yi et al., 2023), indicating the necessity for limited/regulated presence in the cytosol. The conversion of OPDA to 4,5-ddh-JA within the peroxisomes is evidenced by Ueda et al. in their complementary manuscript. These facts have been now discussed in more detail in lines 490-504.

4) Although many experiments have been conducted to verify the effects of externally administered OPDA using various mutants, this is not an accurate analysis because it is impossible to distinguish from the endogenous background unless accurate conversion experiments using deuterium-labeled compounds are conducted. Exogenously applied OPDA also induces gene expression, but according to the authors, OPDA has a clear compartmentation and does not leak externally. How much of OPDA is converted to JA?

We thank the reviewer for this valuable comment and apologize that we did not describe and discuss these experiments in a way that our evaluation of the results became clear. Application of OPDA was done to show the published effects of OPDA (induction of OPDA-marker genes) when applied from outside. We used water-treated *opr2opr3* seedlings as a control for comparison with the OPDA-treated *opr2opr3* seedlings. The water-treated seedlings represent the background levels of OPDA and are therefore used as control to estimate the amount of OPDA taken up by the seedlings during OPDA treatment. Whereas the control seedlings contained 7.7 nmol/g FW, OPDA-treatment resulted in an accumulation of 45.2 nmol/g FW *cis*-OPDA. This level reflects the actual internal amounts of OPDA taken up by the seedlings (Fig. 4c) and represents an almost six-fold increase in internal OPDA levels in the *opr2opr3* mutant. Although we used for these experiments the *opr2opr3* mutant to prevent formation of JA, minor amounts of JA/JA-Ile were measurable, but almost at detection limit of the method (0.1 and 0.015 nmol/g FW, respectively, shown now in Fig. S7). According to this 2.6-fold increase in JA-Ile levels above the basal levels, we found a slight induction of JA-related genes. Since this induction appeared to be rather minor, we shifted these data to the supplemental data and focused more on the OPDA-related changes in gene expression (now Fig. 4). Here, application of OPDA to *opr2opr3* plants resulted in a significant change of gene expression, whereas endogenous rise did not. This difference might be due to the different localization/compartmentation of OPDA and is now supported by the data presented by Ueda et al.: Endogenously produced OPDA might be confined to plastids and peroxisomes, whereas applied OPDA enters the cytosol followed by transport into peroxisomes, where it is converted to 4,5-ddh-JA.

5) Line 182-183 The authors described that the difference in gene expression between *opr2opr3* and *aos* is slight, but based on FigS4b, it is likely that there are more differently expressed genes (DEGs). Rather than processing data based on *opr2opr3/aos*, I would like to propose an accurate quantification based on RT-qPCR analysis of each gene expression.

In the former Fig. S4b (now Fig. S5b), the DEGs shown are JA-dependent at control condition and after wounding. This heatmap represents the average FPKM values of three independent biological replicates, transformed into row-normalized Z-scores. While subtle differences in expression levels of specific genes like MBP1, QQS, and DAO2 are apparent between *opr2opr3* and *aos* in the heatmap, it's crucial to note that these differences do not meet the conventionally used fold change (FC) cutoff of 2. As a result, the observed variations in expression between *opr2opr3* and *aos* for these genes cannot be considered significant. Therefore, these genes are induced in wild type only and did not show altered transcript levels between *opr2opr3* and *aos*. Moreover, the heatmap, which displays the 2000 most DEGs across all genotypes, along with the PCA plot, demonstrates high similarity between *opr2opr3* and *aos* whole transcriptomes under control conditions and wounding (Fig. S3). This aligns clearly with the direct comparison of the *opr2opr3* and *aos* transcriptomes (Fig. 3). These comparisons were conducted in all datasets using a conventional fold change (FC) cutoff of 2, leading to similar conclusions and demonstrating consistency in the findings.

6) Line 230-233 It is impossible to derive this statement from the current data. It is only a hypothesis unless we do experiments with opr1 mutant lines, thus, this should be moved to Discussion.

Thank you for your comment. We removed this statement.

7) Line 236-240 Could these be side effects of exogenously applied OPDA? The possibility that there are other main effects cannot be ruled out.

Thanks to the reviewer rising this point. OPDA application to plants is necessarily stress-inducing due to OPDA being a reactive electrophilic species known to be cytotoxic (Alm eras et al., 2003; Mueller et al., 2008). Therefore, side effects arising from exogenous OPDA treatment cannot be ruled out and may be also mistakenly attributed as primary OPDA effects. To accurately characterize OPDA effects, we compared the effects of the exogenously applied OPDA with those caused by endogenously produced OPDA, and here again we used a holistic transcriptomic approach (RNA-seq) to detect (almost) all DEGs after application of OPDA to *opr2opr3* mutant seedlings instead of picking single genes for RT-qPCR which does not give an overview of the whole transcriptional response. This is shown by the heatmap in Fig. 4. Therefore, the selected genes tested by RT-qPCR belong to the most abundantly induced

cluster of DEGs upon OPDA treatment. The comparison between both data sets indicates that endogenously produced OPDA in Arabidopsis does not elicit similar effects as the OPDA treatment does. This could be because endogenous OPDA either does not inherently induce such effects or its effects are mitigated due to its regulated localization within the cell.

8) Lines288-291 After conversion to OPC-8, does OPC-8 undergo β -oxidation at the peroxisome? And is there evidence for the existence of an OPC-8 transporter? An experiment in which deuterated α -linolenic acid is externally administered (Chini et al. Nat. Chem. Biol. 2018) may be useful to show that endogenous OPCDA is converted to JA-Ile in this experiment.

The reviewer is right that we did not say it very conclusively, how OPR3 located in different organelles would be able to contribute to JA formation. At this point of the “story”, this was not possible with the data shown. Therefore, we speculated two possible biosynthesis pathways: (i) first OPDA reduction to OPC-8 followed by its β -oxidation to form JA, and/or (ii) OPDA shortened to 4,5-ddhJA by β -oxidation followed by conversion to JA by OPR3. To clarify this, we used then OPR2 and OPR3 for the transorganellar complementation. The final proof was done by locating OPR1 into mitochondria – OPR1 is not able to convert 4,5-ddh-JA. The missing rescue of the phenotype of the *opr2opr3* mutant was therefore taken as supporting the variant (ii) of the proposed pathways above. Taking all data together, it became obvious that not OPDA was converted by the respective OPR enzyme, but 4,5-ddh-JA. In addition, the manuscript submitted by Ueda et al. show unequivocally that deuterated OPDA applied to *opr2opr3* is also directly converted to 4,5-ddh-JA within the peroxisomes. We now refer to these data shown in the accompanying submission.

In the canonical pathway of JA formation, it is agreed that OPC-8 undergoes β -oxidation in the peroxisome, leading to JA formation (Vick and Zimmerman, 1984). As peroxisomal β -oxidation is the sole pathway for the metabolic breakdown of fatty acids in plants, β -oxidation of OPC-8 occurs necessarily in the peroxisomes. The ABC transporter COMATOSE (CTS) is a known importer of fatty acids into the peroxisomes and is also able to import OPDA into this organelle (Theodoulou et al., 2005, Ueda et al., accompanying manuscript) making also the import of OPC-8 into the peroxisomes plausible. Such a transport of OPC-8 has, however, not been described, and might be much less efficient than the transport of OPDA. This might be an additional reason that complementation with cytosolic OPR1 led to a partial rescue of *opr2opr3* mutant only.

9) Lines 320-326 In order to draw this conclusion, the use of opr2opr3 double mutant is essential, and the current data are insufficient to draw this conclusion.

For all the complementation studies the mutant *opr2opr3* has been used. Furthermore, the *opr2opr3* mutant transformed with an empty vector has been used as a control for all comparisons.

Minor concern

1) lines 138-144 It is difficult to follow what the authors are trying to say. Since aos mutants cannot biosynthesize allene oxide, is it evident that they cannot synthesize OPDA even in the presence of AOS protein?

We apologize for this confusing sentence. It has been corrected.

2) Fig 1d It is strange that when wounding is not applied, AOC protein is upregulated, but aos gene expression is not.

This effect of upregulated AOC protein levels is due to the long-lasting effect of JA-treatment on protein levels of biosynthetic enzymes. The MeJA pretreatment was done during the seedlings' development to restore the JA feedback loop characterized by (Stenzel et al., 2003). It was terminated four days before the seedlings' wounding and harvest (Fig. S2a).

Consequently, at such a long time point after MeJA treatment, we do not observe increased transcript levels as visible by short-term MeJA treatment or upon wounding as these effects are not typically sustained over time. Moreover, the discrepancy between AOC transcript and protein levels has been previously demonstrated by (Stenzel et al., 2003; Stenzel et al., 2003), as they showed that AOC mRNA levels show a transient increase (up to 4 hours) following MeJA treatment, but AOC protein levels reach their peak at 1-2 days post-treatment.

Reviewer #2:

The manuscript addresses an important debated question in the field related to a putative role of OPDA in the activation of wound responses independent of its conversion in JA/JA-Ile.

Firstly, the authors undertook a transcriptomic approach to study the response to wounding of mutants deficient in OPDA and JA/JA-Ile (*aos*) or deficient in JA/JA-Ile but accumulating OPDA (*opr2opr3*). The transcriptomic results show very little or no differences at all between these mutants in their response to wounding.

Secondly, the authors show that exogenous treatment with OPDA, rather than the endogenous molecule is responsible for the OPDA responses described previously.

Finally, to clarify if OPDA or 4,5-ddh-JA is the mobile signal between cellular compartments, they took a trans-organellar complementation approach of OPR isoenzymes in *opr2opr3* mutant background. This part is very interesting and elegant, and authors conclude that the mobile molecule is 4,5-ddh-JA.

Despite addressing an important question and the elegance of the approach, the conclusions are not fully supported by the results and there are a few issues that need further clarification.

Major issues:

1-Despite authors assumption, the double *opr2opr3* mutant is not a complete depletion of JA since OPR1 is still active in the cytosolic pathway. Similarly, *aos* can also produce some OPDA by the non-enzymatic pathway and JA. I agree with the authors that in their measurements the levels of JA in the mutants are very low. However, if they want to test a putative role of OPDA independent of JA, the use of “leaky” mutants is not the cleanest option. This is particularly relevant, as the authors introduce, when the precedent of using the “leaky” *opr3* has caused a wealth of wrong conclusions in the field.

Thanks to the reviewer for raising this point. Both, *aos* and *opr2opr3* double mutant, are the best non-JA-producing mutants known to date. A non-enzymatic production of JA has never been shown, even our measurements for both mutants showed JA/JA-Ile levels at detection limit of the methods (see below). Indeed, if a leakiness of OPDA and JA/JA-Ile production would exist in the *aos* and *opr2opr3* mutants, respectively, we expect it to be reflected in our hormone measurements as well as in the transcriptional data. In this study, we presented three sets of hormonal data from the mutants under control and wounding conditions, which, in our opinion, do not hint at a leakiness of the used mutants in OPDA or JA production under these

conditions. To alleviate such concerns, the actual values from these measurements are now shown in the Supplemental data (Tables S1-S3, see below). It becomes obvious that no increase in JA and JA-Ile levels occurred due to wounding in *aos*. For the *opr2opr3* mutant, we recorded a very slight induction of JA (0.002 nmol/g of fresh weight), which we reported in the text (lines 119-123) and compared it to the reported levels published for the *opr3* mutant (0.2 nmol/g of fresh weight; (Stintzi and Browse, 2000)). Here, the *opr2opr3* mutant is a hundred times more deficient than the *opr3* mutant, hence it cannot be considered leaky like *opr3*, which was also validated by (Chehab et al., 2011) and (Chini et al., 2018). Moreover, even if there would be a “leakiness” in JA production in these mutants after wounding, this would be also visible as JA-inducible gene expression (similarly to the results visible after exogenous application, where some of the excess of OPDA in the cytosol seems to be converted to JA-Ile). This was not the case (see Fig. S5). Nevertheless, both mutants differed dramatically in the level of OPDA after wounding – about 29 nmol/g FW in *opr2opr3* and 0.2 nmol/g FW in *aos*. This almost 150-fold difference between both mutants was the most important argument to search for OPDA-induced genes.

Tables S1-S3:

Table S1: OPDA and JA levels shown in Fig. S1c-d: Seedlings grown without MeJA pretreatment under control and wounding conditions do not show accumulation of OPDA or JA levels in the *aos* mutant.

	cis -OPDA (nmol/g fresh weight)		JA (nmol/g fresh weight)	
	Control	Wounding	Control	Wounding
Col-0	0.733 (\pm 0.119)	21.127 (\pm 0.166)	0.016 (\pm 0.0046)	2.879 (\pm 0.400)
opr2opr3	0.277 (\pm 0.038)	4.251 (\pm 0.718)	0 (\pm 0)	0.00243 (\pm 0.0029)
aos	0.060 (\pm 0.012)	0.045 (\pm 0.008)	0.00018 (\pm 0.00032)	0.00015 (\pm 0.00027)

Table S2: OPDA and JA levels shown in Fig S1e: Measurements were conducted using MeJA-pretreated and wounded seedlings (set-up shown in Fig. S2a). Note the increased JA levels in both mutants only following MeJA pretreatment (w/o wounding), being expected from residual levels sustaining in the tissue due to the pretreatment. Seedlings of *aos* again did not accumulate OPDA.

	cis -OPDA (nmol/g fresh weight)			JA (nmol/g fresh weight)		
	Mock -Wounding	MeJA -Wounding	MeJA +Wounding	Mock -Wounding	MeJA -Wounding	MeJA +Wounding
Col-0	9.585 (± 4.215)	12.343 (± 2.897)	26.857 (± 1.675)	0.0484 (± 0.016)	0.169 (± 0.051)	5.253 (± 0.678)
opr2opr3	2.2107 (± 0.540)	13.522 (± 3.048)	29.056 (± 6.894)	0.00106 (± 0.00098)	0.059 (± 0.0141)	0.073 (± 0.009)
aos	0.135 (± 0.1098)	0.208 (±0.218)	0.241 (±0.071)	0.0013 (±0.0011)	0.0536 (±0.0418)	0.0655 (±0.0145)

Table S3: OPDA and JA levels shown in Fig S5a: Measurements using rosette leaves from adult plants, which were pretreated with MeJA and wounded as a final treatment before harvest (set-up shown in Fig. S2c). The levels of both OPDA and JA mirror that obtained with seedlings, presented in both tables above. They clearly show very low levels of OPDA and JA in the mutants being not enhanced by wounding and further validate their non-leakiness.

	cis -OPDA (nmol/g fresh weight)			
	- MeJA - Wounding	- MeJA + Wounding	+ MeJA - Wounding	+ MeJA + Wounding
Col-0	3.545 (± 0.3406)	25.998 (±8.359)	9.406892 (± 4.245)	29.243 (± 13.116)
opr2opr3	4.570 (± 2.895)	8.731 (± 2.029)	6.423 (± 0.2401)	15.633 (± 7.648)
aos	0.0277 (± 0.0242)	0 (± 0)	0.0140 (± 0.0244)	0.0290 (± 0.0503)
	JA (nmol/g fresh weight)			
	- MeJA - Wounding	- MeJA + Wounding	+ MeJA - Wounding	+ MeJA + Wounding
Col-0	0.0506 (± 0.0331)	6.849 (± 2.423)	0.0873 (± 0.0527)	2.406 (± 2.226)
opr2opr3	0.0151 (± 0.0144)	0.00286 (± 0.0025)	0.00535 (± 0.0048)	0.0026 (± 0.0046)
aos	0.01088 (± 0.0079)	0.0024 (± 0.0042)	0.0118 (± 0.0090)	0.0110 (± 0.00534)

Instead of leaky mutants, if authors want to test a putative JA-independent (=COI1-independent) role of OPDA, they should use the *coil* mutant. Does *coil* accumulates OPDA after wounding? The use of *coil* would also avoid over-manipulation of the plants with the MeJA treatment. At least, the *coil* results would complement/confirm those presented here.

Of course, the use of the *coil*-mutant would exclude JA-Ile-related gene expression. While this mutant can be used to characterize COI1-independent gene expression, it is not possible to determine whether such a COI1-independent response is OPDA-dependent or not without using an OPDA-deficient mutant, such as the *aos* mutant. The reviewer's critique overlooks the fact that the wound response involves multiple signaling pathways and incorrectly assumes that a COI1-independent response is necessarily equivalent to an OPDA-dependent response. Nevertheless, we followed this advice by the reviewer and checked (i) the levels of OPDA, JA and JA-Ile, and (ii) the expression of selected genes in this mutant after wounding (shown in Figure S4). The disadvantage in using the *coil*-mutant is, however, that the plants contain just 25 % of the OPDA level of WT and a “rescue” by treatment with JA is not possible. Therefore, a putative OPDA-specific induction of gene expression in this mutant is to expect much smaller than in *opr2opr3* due to the differences in the OPDA levels. Here too, the comparison to the wound response of *aos* is the most important one to characterize OPDA-induced processes. The results using the *coil* mutants support our narrative, as the known OPDA markers, mainly *DREB2A*, *GST6*, and *ERF5*, showed a similar induction by wounding in the *coil* mutants and wild type, despite these mutants accumulating lower OPDA levels upon wounding. *ZAT10* and *FADOXR* showed a diminished but not abolished induction in the *coil* mutants, but their induction in the *aos* mutant, which does not produce OPDA (Figure 2), indicates their OPDA independence. Hence, the transcriptional induction of all major OPDA-markers that we tested does not correlate with OPDA levels, indicating that it is OPDA-independent (Figure S4).

2- Transcriptomic analyses: Although pairwise comparisons show scarce differences between *opr3opr2* and *aos* mutants, the Venn diagrams in Figs 2A and Fig S5B show an important difference (108+173 and 211+406 genes). Since pairwise comparisons do not take into account basal levels, a better analysis such as WGCNA could arise significant differences.

The Venn diagrams in Fig. 2a and former Fig. S5b (data presented now in Fig. S6b, c) are prone to misinterpretation since the basis of the comparisons performed was not explained carefully. These Venn diagrams show the comparison between genotypes based on the

differentially expressed genes (DEGs) by wounding, selected individually within each genotype with a fold-change (FC) cutoff of 2. Considering this FC cutoff is used within the three individual datasets, it is important to note that if a gene A has an FC=2 in *opr2opr3* and an FC=1.99 in *aos*, it will appear in the *opr2opr3*-specific area of the Venn diagram, but not in *aos*. This could then be interpreted as a significant difference in gene A expression between *opr2opr3* and *aos*, although this is not true in the direct comparison. Therefore, we changed the Venn diagrams into bar graphs showing the number of up—and down-regulated genes in each genotype upon wounding (Fig. 2a, Fig. S6 b-c). To provide a more accurate interpretation of the whole data, we have complemented this analysis with direct comparisons between the genotypes, along with heatmaps and PCA plots (Fig. S3, Fig. 4a) offering an overview of the whole transcriptomic datasets of the genotypes.

In contrast, the pairwise comparison between the *opr2opr3* and *aos* mutants considered the differences in basal transcript levels and upon wounding (Fig. 3). In this context, the control condition accounts for differences in the basal levels of transcripts between the mutants and showed 7 DEGs only.

Additionally, patterns shown in Fig S3a would suggest that JA/JA-Ile have a very little role in wound-induced gene expression, since only a few genes differ between WT and mutants. However, the likely answer to this issue is that the mutants are leaky and, therefore, the use of *coil* is important. This explanation also applies to Fig S4 for the induction of JA-dependent genes in the mutants.

The reviewer is correct – one could assume from the heatmap shown in Fig. S3a that the minority of genes is JA/JA-Ile regulated upon wounding. The heatmap displays a clustering of the 2000 most variable genes at 1 h post wounding across the entire dataset. Thereby, the greatest variability in the transcriptome data comparing the conditions (control and wounding) and the genotypes (wild type and mutants), is due to the wounding, appearing quite similar across the tested genotypes. This could be interpreted as the mutants being still synthesizing JA/JA-Ile. However, our hormone data and previous work on the *opr2opr3* and *aos* mutants show that they are indeed JA-deficient. Taking both hormone and transcriptome data together, the results show that JA participates only partially in the wound transcriptional response. Accordingly, the analysis of the wound-induced DEGs within each genotype indicates that there are 10.84% and 7.48% more DEGs in Col-0 than in *opr2opr3* and *aos*, respectively (Fig. 2a, using a cutoff of FC=2 and FDR=0.05). These genes/DEGS are, therefore, JA-dependent.

It is well-established that JA/JA-Ile can trigger essential specialized metabolism pathways indispensable for plant immunity but without necessarily inducing the highest number of DEGs in the **early** wound response. There are only few studies that have isolated the exact portion of DEGs that are JA-regulated in the wound response. Initially, studies using microarrays revealed that application of jasmonate to cell cultures and Col-0 leaves induced 589 DEGs out of 15,426 genes and 147 DEGs out of 8000 genes, respectively (Lorenzo et al., 2003; Pauwels et al., 2008). Insect feeding led to only 111 DEGs out of 7,200 genes, from which 74% were COI1-dependent (Reymond et al., 2004). Furthermore, a microarray study comparing the wound responses in wild type and the *coi1-16* mutant showed 237 DEGs, from which 131 and 106 were COI1-independent and COI1-dependent, respectively (Devoto et al., 2005). Even a meta-analysis of RNA-seq data obtained from MeJA and coronatine application for various time periods showed that only a minority of genes is specifically regulated by JA/COR (Zhang et al., 2020). This was also highlighted by a global analysis of Arabidopsis early wound response showing that only 26% of the wounding-regulated genes are induced by JA treatment (Moore et al., 2021). These rather low number of JA-related genes is sufficient to confer resistance to pathogens and herbivores, since similar to the *coi1* mutant plants, *opr2opr3* and *aos* mutants have been previously demonstrated to be susceptible to these pests (Goodspeed et al., 2012; Zhurov et al., 2013; Chini et al., 2018). Hence, their defense mechanisms remain defective due to their JA deficiency.

Additionally, Fig. S5b depicts several genes that are reliably used in the literature as JA markers and JA-inducible defense markers. These markers clustered as JA-dependent in our data and were not significantly induced in the mutants. Their JA-dependence confirms both the reliability of our transcriptomic data and the JA/JA-Ile deficiency of the mutants used.

3- The major issue in this work, however, is the assumption that “in vivo” the specificity of OPR1 and OPR2 for 4,5-ddh-JA and OPDA would be similar to their specificity “in vitro”. It is clear that the double *opr2opr3* still accumulates some JA, which supports that “in vivo” OPR1 either reduce OPDA or 4,5-ddh-JA. The fact that OPDA can be in the cytoplasm is obvious by its “travel” from the chloroplasts to the peroxisomes, and also by previous works demonstrating that OPDA is conjugated with aminoacids, or oxidized by the recently identified JID1.

Thank you for raising this point. Indeed, there is always the problem to conclude from *in vitro* data on the *in vivo* activity of enzymes. Nevertheless, as shown by (Chini et al., 2018), OPR1

is neither *in vitro* nor *in vivo* able to reduce 4,5-ddhJA. On the one hand, no reduction of 4,5-ddh-JA was observed for OPR1 in the enzymatic assay. On the other hand, the *in vivo* (non)activity has been concluded from the facts that (i) wound-induced levels of 4,5-ddh-JA in *opr1opr3* double mutant were as high as those in the single mutant *opr3-3*, and (ii) wound-induced JA accumulation was only slightly reduced in *opr1opr3* compared to *opr3*, whereas it was undetectable in *opr2opr3*. The authors concluded that OPR2 is the main enzyme responsible for JA reduction from 4,5-ddh-JA but OPR1 is not.

Regarding the occurrence of OPDA within the cytosol, we refer to the arguments given in response to Reviewer 1:

Using the transorganellar complementation approach, we showed that OPDA might be restricted to the organelles of its biosynthesis (plastids) and metabolism (peroxisomes). An occurrence of OPDA in the cytosol can, however, not be excluded, since there has to be a transfer from the plastids to the peroxisomes via the cytosol. While the use of the term “OPDA sequestration” in our phrasing may have been interpreted as implying the absence of OPDA in the cytosol, we actually state that OPDA is present in the cytosol but in a limited amount (see lines 469-471, 484-487, 1075-1076). The amounts of OPDA within the cytosol might be, however, rather minor due to the following facts: (i) OPR1 located in the cytosol was not able to rescue fully the mutant, while its peroxisomal counterpart led to a full complementation (Fig. 7b-c). This partial complementation suggests a limited presence of OPDA in the cytosol rather than its absence. (ii) OPDA-amino acid conjugates are occurring late in development of plants (Floková et al., 2016) and were detected at very low levels (1-10 pmol/g FW of conjugates in comparison to 25 nmol/g FW of OPDA, see (Mik et al., 2023)). (iii) As a precursor of the bioactive JA-Ile, the transport of OPDA through the cytosol is likely tightly regulated, as indicated by its export from chloroplasts via JASSY (Guan et al., 2019) and import to the peroxisome via the ABC transporter COMATOSE (CTS) (Theodoulou et al., 2005). Moreover, as a reactive electrophile species, OPDA is relatively cytotoxic (Alm eras et al., 2003; Mueller et al., 2008), and its detoxification through conjugation with GSH has been demonstrated in Arabidopsis and tobacco (Mueller et al., 2008; Ohkama-Ohtsu et al., 2011), indicating the necessity for limited/regulatory presence in the cytosol. Moreover, the conversion of exogenously applied OPDA to 4,5-ddh-JA within the peroxisomes is evidenced by Ueda et al. in their accompanying manuscript. These facts have been now discussed in more detail in lines 490-504.

Minor points:

-Intro: “Lower plants” is not a correct term. It is better to use “non-vascular plants” or other more appropriate term

Changed accordingly.

-Intro: The “leakiness of the opr3” was first demonstrated by Chehab et al., 2011.

Chehab et al. (2011) demonstrated that the *opr3-1* mutant is a conditional mutant and is not a null-mutant allele under fungal infection, as it produces *OPR3* transcripts and consequently JA (Figure 3C in this paper). Under control conditions or upon wounding, the *opr3-1* mutant was not considered “leaky” since *OPR3* transcripts did not accumulate, and JA was not measurable. Chini et al. (2018) used the null mutant *opr3-3*, which is “OPR3-transcript-free” under several conditions, and identified a JA formation in that mutant through OPR2. Here, we refer to the leakiness in JA biosynthesis due to the cytosolic OPR3-bypass pathway, not to the leakiness of the mutant allele. The text has been changed accordingly (lines 88-94).

-Results: Page 6: Have the authors tested the levels of AOC after wounding?

The levels of AOC protein have not been tested after wounding of seedlings/leaves, since it has been shown previously that – at least in short-time periods of treatment with JA or wounding – AOC protein levels do not increase and a clear increase in AOC protein levels was detectable only after 24 h (see Fig. 8 in (Stenzel et al., 2003)). Given that we are using an 1-hour time point after wounding, significant changes in protein abundance are unlikely to be observed. We used JA-treatments on three consecutive days and finished them four days before onset of the wounding (see Fig. S2). This set-up was chosen to obtain the long-term effect due to the positive feedback to increase the protein levels over plant development on the one hand, and to allow jasmonate levels to get back to basal levels before onset of wounding. This has been tested at time point 0, i.e. directly before wounding.

-Supp Fig 5a: Differences are not statistically significant in non-wounded plants. This argues against the MeJA treatment.

The reviewer is correct, the OPDA levels did not show significant differences in the non-wounded wild type and *opr2opr3* rosette leaves without MeJA pretreatment. It seems, however, that the high levels in *opr2opr3* are due to an outlier, as this value shows a high standard deviation (see Table S3 above, shown as Supplementary Tab. S3). Nevertheless, wounding enhanced the OPDA levels in Col-0 and *opr2opr3*, both showing high levels in comparison to *aos* making the follow-up transcriptomics analysis reliable.

-Supp Fig 6: The result is not so surprising since, as shown in Fig 4B, such a high concentration of OPDA induces the accumulation of JA-Ile in the mutants. Again, the use of *coil* would prevent this type of results.

Figure S6 (now presented as Fig. S7a) depicts the complete transcriptional response and is clustered based on gene ontology enrichment analysis. Next to the main clusters 2 and 3 showing enrichment in sulfur metabolism and common response to cellular stress, respectively, cluster 1 shows significant enrichment of a part of the JA/defense pathway in the *opr2opr3* mutant fed with OPDA. This result complements and correlates with the slightly enhanced JA-Ile levels shown in previous Fig. 4c (now Fig. S7c). This finding is worth highlighting, given that such conversion of exogenously fed OPDA to JA-Ile in the *opr3* single mutant or wild type was not taken into consideration by any previous study or just evaluated as a minor factor (Arnold et al., 2016). These results, however, differ from those shown by Ueda et al. (accompanying manuscript), who did not detect any conversion of applied OPDA to JA/JA-Ile. This might be caused by the cultivation of seedlings and the application mode. Most importantly, however, the measurement method used might cause the difference, since the levels are close to the detection limit of the method and might be dependent on the sensitivity of the LC-MS. These very low levels are also reflected by the low expression levels of JA-induced genes, e.g., *CHL1* shows only 25 % of the transcript accumulation after OPDA treatment of *opr2opr3* in comparison to wounded wild type seedlings (Supplementary Figs. 5 and 7).

-Fig 4c: It is surprising the low amount of cis-OPDA after OPDA treatment. Is this result correct? Why is the plant not accumulating the OPDA from the treatment?

The levels of OPDA within the OPDA-treated seedlings reached levels of about 45 nmol/g FW, which is almost 6-fold increase in comparison to water treated seedlings within 30 min of

treatment. This level of OPDA accumulation aligns with the data shown by (Arnold et al., 2016) demonstrating that in wild type *Arabidopsis* seedlings accumulated 40-45 nmol/g FW of *cis*-OPDA upon treatment with 50 μ M OPDA after 30 min (Figure 1A, (Arnold et al., 2016)). These OPDA levels remained high in the following incubation period and did not increase further up to 4 h of treatment.

-Fig 5c-g: I couldn't find these panels

We apologize for this mistake and have corrected it.

-Author's data suggest that the JA-independent response after wounding is related to general stress. Have authors considered that other stresses different than wounding may induce higher levels of OPDA, which may induce an OPDA-specific response?

We thank the reviewer for this comment. We tried, however, to be very careful in the interpretation of our data and stated that our results – the limited signaling capacity of wound-induced OPDA – is specifically shown for wound-response of seedlings and rosette leaves. Other stresses were not tested and might lead to OPDA-specific responses. This is now explicitly stated (again) in discussion, lines 490-504.

-Are expression levels of transgenics in the complementation assays comparable?

We apologize for missing to present the respective data. They are now included as Fig. S10.

- In figures 6a-h and 7c, is there any possibility to quantify the phenotype/recovery of the fertility?

For most of the complementation assays we got a complete rescue of the *opr2opr3* phenotype in terms of full fertility, i.e. 100 % of flowers developed into proper siliques containing seeds. There were only two approaches, which did not lead to rescue at all: the empty vector control and localization of OPR1 in mitochondria. In these cases, all siliques were small, and no seeds could be obtained from the transgenic plants (= 0 %). The “partial rescue” upon expression of OPR3 in ER and OPR1 in cytosol refers to development of few seed-containing and slightly

smaller siliques. In the case of ER-OPR3 this has been quantified as shown below for three independent lines. Since these data are not relevant for the manuscript, we do not include them.

Phenotyping of the stem length and fertility-related parameters of *opr2opr3* lines complemented with OPR3 located either in the peroxisomes (full rescue) or in the ER-OPR3 (partial rescue).

Data show the partial rescue of *opr2opr3* by expression of ER-OPR3 and are represented in box plots of five biological replicates with standard deviation and median lines (A-D), scatter plots of five biological replicates with means (E) and violin plots with median and quartiles (F). Different letters denote statistically significant differences among genotypes as determined by Two-way ANOVA followed by Tukey HSD ($p < 0.05$).

A. stem length, B. flowering time, C. flower number, D. silique yield, E. first silique position, F. silique size (taken from Mekkaoui, PhD thesis, Martin-Luther-University Halle, 2023)

- In the discussion, dn-OPDA is described as "a beta-oxidation product of OPDA". Actually, the major source of dn-OPDA is 16:3, and the beta-oxidation of OPDA contributes very little.

Thanks to the reviewer for raising this point. According to Chini et al. (2018), dn-OPDA can be produced from OPDA, at least if the OPR3 is not present. Since this might be a special case, we followed the advice of the reviewer and changed the text accordingly (lines 361-364).

References:

- Alm ras E, Stolz S, Vollenweider S, Reymond P, M ne-Saffran  L, Farmer E** (2003) Reactive electrophile species activate defense gene expression in *Arabidopsis*. *Plant J* **34**: 205-216
- Arnold MD, Gruber C, Flokov  K, Miersch O, Strnad M, Nov k O, Wasternack C, Hause B** (2016) The recently identified isoleucine conjugate of *cis*-12-oxo-phytodienoic acid is partially active in *cis*-12-oxo-phytodienoic acid-specific gene expression of *Arabidopsis thaliana*. *PLoS ONE* **11**: e0162829
- Chehab EW, Kim S, Savchenko T, Kliebenstein D, Dehesh K, Braam J** (2011) Intronic T-DNA insertion renders *Arabidopsis opr3* a conditional jasmonic acid-producing mutant. *Plant Physiology* **156**: 770-778
- Chini A, Monte I, Zamarre o AM, Hamberg M, Lassueur S, Reymond P, Weiss S, Stintzi A, et al.** (2018) An OPR3-independent pathway uses 4,5-didehydrojasmonate for jasmonate synthesis. *Nature Chemical Biology* **14**: 171
- Devoto A, Ellis C, Magusin A, Chang H-S, Chilcott C, Zhu T, Turner JG** (2005) Expression profiling reveals *COII* to be a key regulator of genes involved in wound- and methyl jasmonate-induced secondary metabolism, defence, and hormone interactions. *Plant Mol. Biol.* **58**: 497-513
- Flokov  K, Feussner K, Herrfurth C, Miersch O, Mik V, Tarkowsk  D, Strnad M, Feussner I, et al.** (2016) A previously undescribed jasmonate compound in flowering *Arabidopsis thaliana* – The identification of *cis*-(+)-OPDA-Ile. *Phytochemistry* **122**: 230-237
- Goodspeed D, Chehab EW, Min-Venditti A, Braam J, Covington MF** (2012) *Arabidopsis* synchronizes jasmonate-mediated defense with insect circadian behavior. *Proceedings of the National Academy of Sciences of the USA* **109**: 4674-4677
- Guan L, Denkert N, Eisa A, Lehmann M, Sjuts I, Weiberg A, Soll J, Meinecke M, et al.** (2019) JASSY, a chloroplast outer membrane protein required for jasmonate biosynthesis. *Proceedings of the National Academy of Sciences* **116**: 10568-10575
- Lorenzo O, Piqueras r, S nchez-Serrano J, Solano R** (2003) ETHYLENE RESPONSE FACTOR1 integrates signals from ethylene and jasmonate pathways in plant defense. *The Plant Cell* **15**: 165-178
- Mik V, Posp sil T, Brunoni F, Gr z J, No zkov  V, Wasternack C, Miersch O, Strnad M, et al.** (2023) Synthetic and analytical routes to the L-amino acid conjugates of *cis*-OPDA and their identification and quantification in plants. *Phytochemistry* **215**: 113855
- Moore BM, Lee YS, Wang P, Azodi C, Grotewold E, Shiu S-H** (2021) Modeling temporal and hormonal regulation of plant transcriptional response to wounding. *The Plant Cell* **34**: 867-888
- Mueller S, Hilbert B, Dueckershoff K, Roitsch T, Krischke M, Mueller MJ, Berger S** (2008) General detoxification and stress responses are mediated by oxidized lipids through TGA transcription factors in *Arabidopsis*. *Plant Cell* **20**: 768-785

- Ohkama-Ohtsu N, Sasaki-Sekimoto Y, Oikawa A, Jikumaru Y, Shinoda S, Inoue E, Kamide Y, Yokoyama T, et al.** (2011) 12-Oxo-phytodienoic acid-glutathione conjugate is transported into the vacuole in *Arabidopsis*. *Plant and Cell Physiology* **52**: 205-209
- Park S-W, Li W, Viehhauser A, He B, Kim S, Nilsson AK, Andersson MX, Kittle JD, et al.** (2013) Cyclophilin 20-3 relays a 12-oxo-phytodienoic acid signal during stress responsive regulation of cellular redox homeostasis. *Proceedings of the National Academy of Sciences* **110**: 9559-9564
- Pauwels L, Morreel K, De Witte E, Lammertyn F, Van Montagu M, Boerjan W, Inze D, Goossens A** (2008) Mapping methyl jasmonate-mediated transcriptional reprogramming of metabolism and cell cycle progression in cultured *Arabidopsis* cells. *Proc Nat Acad Sci USA* **105**: 1380-1385
- Reymond P, Bodenhausen N, Van Poecke RMP, Krishnamurthy V, Dicke M, Farmer EE** (2004) A conserved transcript pattern in response to a specialist and a generalist herbivore. *Plant Cell* **16**: 3132-3147
- Scholz SS, Reichelt M, Boland W, Mithöfer A** (2015) Additional evidence against jasmonate-induced jasmonate induction hypothesis. *Plant Science* **239**: 9-14
- Stenzel I, Hause B, Feussner I, Wasternack C** (2003) Transcriptional activation of jasmonate biosynthesis enzymes is not reflected at protein level. *In* N Murata, M Yamada, I Nishida, H Okuyama, J Sekija, W Hajime, eds, *Advanced Research on Plant Lipids*. Kluwer Academic Publishers, pp 267-270
- Stenzel I, Hause B, Miersch O, Kurz T, Maucher H, Weichert H, Ziegler J, Feussner I, et al.** (2003) Jasmonate biosynthesis and the allene oxide cyclase family of *Arabidopsis thaliana*. *Plant Mol. Biol.* **51**: 895-911
- Stintzi A, Browse J** (2000) The *Arabidopsis* male-sterile mutant, *opr3*, lacks the 12-oxophytodienoic acid reductase required for jasmonate synthesis. *Proc. Natl. Acad. Sci. USA* **97**: 10625-10630
- Taki N, Sasaki-Sekimoto Y, Obayashi T, Kikuta A, Kobayashi K, Ainai T, Yagi K, Sakurai N, et al.** (2005) 12-oxo-phytodienoic acid triggers expression of a distinct set of genes and plays a role in wound-induced gene expression in *Arabidopsis*. *Plant Physiol.* **139**: 1268-1283
- Theodoulou FL, Job K, Slocombe SP, Footitt S, Holdsworth M, Baker A, Larson TR, Graham IA** (2005) Jasmonic acid levels are reduced in COMATOSE ATP-binding cassette transporter mutants. Implications for transport of jasmonate precursors into peroxisomes. *Plant Physiology* **137**: 835-840
- Wasternack C, Song S** (2017) Jasmonates: biosynthesis, metabolism, and signaling by proteins activating and repressing transcription. *Journal of Experimental Botany* **68**: 1303-1321
- Yi R, Du R, Wang J, Yan J, Chu J, Yan J, Shan X, Xie D** (2023) Dioxygenase JID1 mediates the modification of OPDA to regulate jasmonate homeostasis. *Cell Discovery* **9**: 39
- Zhang N, Zhou S, Yang D, Fan Z** (2020) Revealing shared and distinct genes responding to JA and SA signaling in *Arabidopsis* by meta-analysis. *Frontiers in Plant Science* **11**
- Zhurov V, Navarro M, Bruinsma KA, Arbona V, Santamaria ME, Cazaux M, Wybouw N, Osborne EJ, et al.** (2013) Reciprocal responses in the interaction between *Arabidopsis* and the cell-content-feeding chelicerate herbivore spider mite. *Plant Physiology* **164**: 384-399

Answers to the concerns expressed by the reviewers regarding our manuscript “Transcriptomics and trans-organellar complementation reveal a limited signaling capacity of 12-*cis*-oxo-phytodienoic acid in wounded Arabidopsis” (NCOMMS-24-18845A-Z)

Dear editor, dear reviewers,

first, we want to thank the reviewers for evaluating (again) our manuscript. We are very grateful for their positive overall assessment of our work and appreciate the critical points they have raised. In the following text, we address all the concerns all reviewers had – either by including new data, changing the text or directly replying to their points. Changes that were made to fulfil the editorial requests are listed in the “Cover letter”. All reviewer comments are underlined, our answers appear in normal font.

Reviewer #2 (Remarks to the Author):

In this revised version authors have addressed some of my concerns, but still important issues remain unsolved. The most important one is again the assumption of OPRs substrate specificity based on previous published results. The authors refer to a previous work (Chini et al., 2018) to support that:

- 1- 4,5-ddh-JA is not a substrate of OPR1
- 2- Wound induced JA accumulation was only slightly reduced in opr3opr1 compared with opr3
- 3- Chini et al., conclude that OPR1 is not involved in 4,5-ddh-JA reduction.

I have carefully revised this work (Chini et al., 2018) and found that these conclusions cannot be drawn from this paper for the following reasons:

- 1- Fig 6D in Chini et al., only shows that OPR1 does not work in this particular assay. However, there is no proof that this enzyme works at all. Since this is a purified enzyme in vitro, it may well be that the protein is just not functional. Therefore, if the authors want to support their conclusion, they should demonstrate that OPR1 is functional and can reduce OPDA but not 4,5-ddh-JA
- 2- If “wound induced JA accumulation was slightly reduced in opr3opr1” this mean that OPR1 uses 4,5-ddh-JA as a substrate. The small (statistically significant) reduction is likely a consequence of OPR2 being more active
- 3- “Chini et al., conclude that OPR1...” This sentence is not true. I have looked very carefully through Chini et al. and they never concluded that OPR1 is not involved in 4,5-ddh-JA. Actually, they conclude the opposite; that both OPR1 and OPR2 use 4,5-ddh-JA as a substrate, although OPR2 is much more active than OPR1. There are several places in the manuscript where Chini et al., explain that OPR1 and OPR2 are both involved in this reduction, with OPR2 being the main player, but a sentence concluding that OPR1 has no role on 4,5-ddh-JA reduction cannot be found in this paper.

Therefore, the experiments of transorganellar complementation are based on wrong premises and would require a formal demonstration of the enzymatic specificity of OPR1.

We are very grateful to the reviewer for this valuable comment and apologize for over-interpreting or even misinterpreting the data published by Chini et al. (2018). The reviewer is correct that these authors discussed their results much more carefully and did not conclude that OPR1 is not

involved in the conversion of 4,5-ddh-JA. To clarify this point, we now performed enzymatic assays with all three OPR enzymes recombinantly produced in *E. coli* to compare their activity on OPDA and 4,5-ddh-JA.

Figure: *In vitro* activity of OPR3, OPR2 and OPR1 in the reduction of OPDA and 4,5-ddh-JA.

As shown above (and in Supplemental Fig. 11), their activities differ greatly: OPR3 consumed OPDA completely and converted 4,5-ddh-JA to JA only partially, whereas OPR2 consumed 4,5-ddh-JA completely and converted OPDA to OPC-8 only partially. Most surprisingly, OPR1 converted both substrates, but with very low activity as indicated by the small product peaks, and the low amount of substrate consumed being almost neglectable as compared to the inactive enzyme control. These data correlate with the enzymatic characteristics of the three OPRs as published by Schaller et al. (2000). Using also recombinantly produced enzymes these authors showed specific activities for the conversion of (9*S*,13*S*)-OPDA of 117 pkat (mg protein)⁻¹ for OPR1 and 50 pkat (mg protein)⁻¹ for OPR2, whereas OPR3 has a specific activity of 17.8 nkat (mg protein)⁻¹. These activities were reflected also by the respective enzymes expressed *in planta* (Schaller et al. 2000). Regarding the conversion of 4,5-ddh-JA, only OPR2 has been tested by Chini et al. (2018) and showed an apparent K_M value in the range of that of OPR3 for OPDA.

The reviewer is right that we cannot conclude from the trans-organellar complementation assay that 4,5-ddh-JA is the mobile compound because OPR1 converts both, OPDA and 4,5-ddh-JA. However, taking into account the different activities of the three OPR enzymes used for the trans-organellar complementation assay, we assume that the amount of OPDA and/or 4,5-ddh-JA able to access the mitochondria and staying within the cytosol is sufficient to result in a detectable conversion to OPC-8 and/or JA by OPR2 or OPR3, but is not sufficient for the activity of OPR1. Therefore, these results suggest a limited occurrence of endogenously formed OPDA in the cytosol and mitochondria, whereas it might be preferentially compartmentalized in plastids and peroxisomes. We have now changed the respective paragraphs of the “Results” and “Discussion” and the summarizing scheme shown in Fig. 8: The former hypothesis that 4,5-ddh-JA is the mobile compound is herewith rejected and only discussed as hypothesis. Nevertheless, we think that the compartmentation of OPDA might contribute to its missing signaling capacity in the early wound response of seedlings.

Reviewer #3 (Remarks to the Author):

The manuscript by Mekkaoui et al. addressed the long-debated and important question on whether OPDA functions as an independent signaling molecule. Comparative transcriptomics studies on rapid (1h) wound response of mutants deficient in OPDA and JA/JA-Ile(aos) or deficient in JA/JA-Ile but accumulating OPDA (opr2opr3) show that very few differential expressed genes between these mutants. Furthermore, the authors show that exogenously applied, but not endogenous OPDA is responsible for the expression of OPDA-marker genes described previously. Interesting and elegant transorganellar complementation experiments of OPR isoenzymes in opr2opr3 mutant background were carried out to test if OPDA or 4,5-ddh-JA is mobile between cellular compartments. The revised manuscript has largely addressed the issues raised by the reviewers. The accompanying manuscript by Ueda et al demonstrated that OPDA signaling is independent of the canonical COI1-JAZ-MYC signaling pathway using coi-1, myc2myc3myc4 and opr2-1opr3-3 Arabidopsis mutant, and that OPDA is not genuine bioactive forms using cts mutant. The common mutants such as opr2-1opr3-3 and coi-1, and very similar figure (Fig. 1 vs Fig 1A) were used in both manuscripts, incorporation of some of their results into this manuscript would consolidate these conclusions.

Thanks to the reviewer for this comment. Regarding the incorporation of results provided by Ueda et al. (accompanying manuscript): We refer to their data within our discussion to support our hypothesis about the missing signaling function of endogenously synthesized OPDA, e.g. uptake and conversion of applied OPDA into peroxisomes, and 4,5-ddh-JA as genuine signaling compound after OPDA application. Ueda et al. used in their approaches, however, application of OPDA only, which does not allow a direct comparison of both data sets or even an integration of their data into our manuscript.

Major issues:

1. The studies are focused on rapid (1 hour) wound responses. Is there the possibility that OPDA signaling activated during late phases of wounding?

The reviewer is correct – we cannot exclude a function in later phases of wounding. To be more concise, we changed the title and text of our manuscript to emphasize that only the early wound-response has been analyzed. Nevertheless, as OPDA levels increase in the cell and peak at 60 minutes following wounding, its signaling is expected to occur or be initiated within this time period. The rise of the levels of a signaling molecule within the cell usually correlates with its perception followed by its signal transduction (e.g., JA-Ile, auxin signaling), yet a later signaling for OPDA cannot be excluded unless investigated. It is important also to take into consideration that OPDA at basal levels did not show signaling, contrasting to signaling molecules that have developmental functions. Here, 10-day-old seedlings and mature rosettes did not show transcriptional differences in dependence of endogenously occurring OPDA – unlike endogenous JA/JA-Ile, where such transcriptional differences occurred as expected from its known growth-related functions.

2. I wonder whether expression of OPRs in the nucleus would rescue the male sterility of opr2opr3 mutant.

Overexpression of nuclear located OPR3 in *opr2opr3* was done and showed rescue of the fertility (see Figs. 5 and 6). To test the effect of overexpression of the other OPRs, we decided to direct them into peroxisomes, cytosol and mitochondria to compare the “native” site of OPR3 location

with the artificial location in the cytosol, where OPDA has to be transferred through, and one compartment, which is an organelle not involved in JA biosynthesis. We missed to direct them also into the nucleus – to perform this transformation during the time of manuscript revision was not feasible.

Minor points

1. L246, there is no Fig. 4d.

It is Fig. 4c. The mistake has been corrected.

2. Fig. 7d, WT and *opr2opr3* control should be added.

The reviewer is right that both controls have to be added. In the former version, we indicated levels from *opr2opr3* transformed with empty vector by a dashed red line, since these levels were already shown in Fig. 6. We included now another line (dashed blue) to indicate the wild-type levels (Col-0 transformed with empty vector) presented already in Fig. 6.

3. In Fig. S6, the average content of cis-OPDA and dn-OPDA is significantly different between *opr2opr3* and *aos* mutant, but statistical analyses showed no significant (ns), please check.

According to the request of reviewer #5, we applied now another statistical test to show significant differences between genotypes and treatment. Also here, differences – although clearly visible by eye – do not appear statistically significant due to the test used. We checked and changed all graphs accordingly. Indeed, statistically significant differences between *opr2opr3* and *aos* were visible in pairwise comparisons but were not obtained when applying the One-Way ANOVA or Two-way-ANOVA with Tukey-HSD. This might be due to the sample size, as when increasing the number of biological replicates such differences are detected. We were not able, however, to perform new experiments with higher sample numbers for all the data sets due to the re-submission timeframe.

4. Fig. S9, why the subcellular localization patterns at the ER (a, b) and cytosol (c, d) are very similar?

To make the difference between ER and cytosolic localization clearer, we repeated the protoplast transformation using ER-located and cytosolic YFP:OPR3 and OPR3:YFP, in co-transformation with the respective organelle-markers. New micrographs are now provided in Supplemental Fig. 9a-d and examples shown below. Note that the ER-localized fluorescence is much more restricted to faint net-like structures while the cytosolic label surrounding the plastids and vacuole appears coarser and more diffuse.

Reviewer #4 (Remarks to the Author):

The manuscript “Transcriptomics and trans-organellar complementation reveal a limited signaling capacity of 12-cis-oxo-phytodienoic acid in wounded Arabidopsis” by Khansa Mekkaoui et al., presents an interesting hypothesis on OPDA compartmentalization and limited signaling properties, contradicting previous reports on (exogenous) OPDA ability to induce gene expression independently on the COI1 receptor.

The authors used an elegant strategy to assess the role of endogenous OPDA using knock-out mutants treated with MeJA and wounding to induce comparable jasmonate levels, as well as a trans-organellar complementation to determine OPDA subcellular localization. Although I appreciate the authors’ efforts to investigate the role of OPDA endogenously produced after wounding, I think that including additional controls in their transcriptomic analyses would strengthen their conclusions and clarify potential contradictions with previously published datasets. Unless these controls are included, it is difficult to evaluate whether the experimental conditions are comparable to previous reports, which hinders the interpretation of the results. The presented results do not support the authors’ claim that OPDA does not have signaling functions. Similar to the previous reviewers’ comments, I strongly recommend the use of *coi1* mutants and OPDA exogenous treatment to get a comprehensive picture. The newly added results on the expression of some marker genes in *coi1-16* and *coi1-30* mutants is not enough to assess the contribution of JA/JA-Ile and OPDA genes to the massive transcriptional changes observed in wounded Col-0, *opr2opr3* and *aos* plants. I agree with the other reviewers’ comments that the number of genes classified as JA/JA-Ile-dependent are strikingly low, even though the JA/JA-Ile-dependent genes constitute a fraction of the wounding response. I found surprising that the JA-dependent wound response (Figure S5) seems to be already activated in Col-0 control plants and only a few of these genes are up-regulated upon wounding in Col-0.

We do not agree with the argument of the reviewer regarding the number of JA/JA-Ile dependent genes upregulated in leaves of Col-0 (shown in Supplementary Fig. 5). Under control conditions (without wounding), there are 88 DEGs in Col-0 in comparison to *opr2opr3* and *aos*. These genes are JA/JA-Ile dependent and might be expressed in Col-0 due to the positive JA-feedback mechanism occurring during development. JA has widely characterized growth-related functions, so signaling during development is necessarily expected when comparing JA-producing and JA-deficient plants. This contrasts to the wound response of Col-0 leaves, where additional 352 DEG were recorded. As explained in the former “rebuttal letter”, these numbers are in the range of commonly detected JA/JA-Ile regulated genes (see “rebuttal-letter_NCOMMS-24-18845”, pages 14-15).

The inclusion of *coi1-30* in the RNAseq experiment would be instrumental to differentiate the JA/JA-Ile/COI1-dependent genes during wounding and differentiate them from the OPDA- or wound-induced genes.

We clearly see the point the reviewer is rising regarding the use of the *coi1* mutant. Obtaining the global transcriptomic changes upon wounding would unequivocally discriminate between COI1-dependent and COI1-independent gene expression. But this was not in the scope of our study: We selected consciously a mutant, which can produce high levels of OPDA (which is not achieved in *coi1*), but not JA/JA-Ile to get insights into the OPDA-mediated transcriptional changes. The other mutant used (*aos*), which is deficient in both compounds, helps to clearly identify the OPDA and JA/JA-Ile independent transcriptional response. As reviewer #5 also mentioned, the *coi1*

transcriptional changes would be expectedly similar to those of *opr2opr3* and of *aos*, and would therefore not deliver new insights. Moreover, the interpretation would be very difficult, since OPDA is at least partially converted to JA/JA-Ile in *coi1*. This might lead to another answer – an OPDA and JA/JA-Ile-dependent, but COI1-independent response. This is by far not the aim of this study. Nevertheless, to show the COI1-independent induction of the previously defined OPDA marker genes, we used even two alleles of *coi1* and measured their transcript accumulation upon wounding in these plants (Supplementary Fig. 4).

Along this line, I also found surprising that exogenous OPDA induced JA genes in *opr2opr3* (Figure S7), which might indicate that *opr2opr3* is not the right tool to assess the role of OPDA independent on JA/JA-Ile/COI1. Is this induction due to OPR1 reducing OPDA in the cytosol in planta?

We think that the induction of JA-regulated genes in the *opr2opr3* mutant by OPDA treatment does not compromise its utility as a tool for studying OPDA function: At control and under wounding conditions, JA signaling in *opr2opr3* remained largely uninduced (Supplementary Fig. 5, see above). Notably, Chini et al. (2018) demonstrated that the *opr2opr3* mutant exhibits a loss of defense responses and is susceptible to pathogen attack and insect feeding due to its JA deficiency. Taking into consideration these data, this mutant is currently the most suitable genetic background available for uncoupling OPDA and JA-Ile functions.

Results from OPDA application rather emphasize the importance of exercising caution when using OPDA application as the conversion of exogenous OPDA to JA/JA-Ile *in planta* should be documented and carefully considered when interpreting results (in our opinion this also applies to the use of OPDA conjugates in feeding experiments). Unlike previous studies, we explicitly addressed this limitation in OPDA application experiments and interpreted the results while taking into consideration possible JA-Ile signaling which can be mistakenly attributed to OPDA.

Based on our data, we can only speculate that the synthesis of JA in *opr2opr3* upon OPDA application is mediated by OPR1. It is important to note that it occurred exclusively under the artificial condition of external OPDA supply. Here, two scenarios are possible: (i) OPDA is directly converted to OPC-8, which is then transported into peroxisomes to undergo β -oxidation, and (ii) OPDA is directly transported into peroxisomes giving rise to relatively high levels of 4,5-ddh-JA as shown by Ueda et al. (accompanying manuscript). As OPR1 can inefficiently convert OPDA and 4,5-ddh-JA to JA (shown by our new data shown in Supplementary Fig. 11 and above in the answer to Reviewer #2), its activity leading to some JA production – either by the one or the other way – cannot be neglected. However, at this stage it is not possible to conclude that OPR1 significantly contributes to JA-Ile production. In contrast, it seems that it rather contributes to it in a very minor way. This holds also true for the conversion of endogenously formed (wound-induced) OPDA. Despite generating high levels of OPDA in *opr2opr3* seedlings upon wounding, this did not result in the putative OPR1-mediated formation of JA-Ile.

Was wounding not strong enough to induce OPDA? Is it a matter of timing? Why were the wounded plants collected after 1 h and the seedlings treated with OPDA only for 30 min?

Our dataset demonstrates that OPDA is produced in significantly high amounts using our wounding method, as a high induction of OPDA levels was consistently observed across multiple independent biological replicates and independent experiments, shown in Fig. 1e, Fig. 6i, Supplementary Fig. 1b, Supplementary Fig. 4c, and Supplementary Fig. 6a. For instance, our wounding of seedlings resulted in OPDA levels rising from 0.7 nmol/gFW to 21.12 nmol/gFW in

Col-0 (see Supplementary Fig. 1) and such levels are found across the experiments. This data is in accordance with literature, e.g., Koo et al. (2009) recorded an OPDA content of around 15-20 nmol/gFW in Arabidopsis leaves of Col-0 at 1 hour after wounding.

The sampling at 60 min after wounding relies on preliminary time-course experiments where we found that accumulation of JA/JA-Ile together with OPDA peaks at 30 min and 1 h following wounding, with 1 hour time point showing slightly, but not significantly higher accumulation of both compounds compared to 30 min. Hence, these time-points are expected to correlate with an increased signaling, which is the case for JA-Ile. Moreover, these early wounding time points are also the most studied time points for JA-Ile/OPDA signaling. For OPDA treatment, we relied on a time-course experiment from Arnold et al. (2016). They showed that the endogenous OPDA content peaked at 30 min after application and was even lower at 60 min. Moreover, seedlings at 30 min after addition of OPDA showed endogenous OPDA levels, which were almost in the same range of that measured in wounded seedlings (see Supplementary Fig. 7e). In our opinion, this parameter is methodologically more relevant for comparing these two different experimental setups than the timing itself.

Again, including *coi1-30* in this experiment (wounded and OPDA-treated plants) would be helpful to clarify which genes are regulated by JA/JA-Ile/COI1 and which genes are regulated by OPDA and/or wounding.

We have included wounding experiments for *coi1-16* and *coi1-30* mutants (Supplementary Fig. 4), where we tested OPDA markers. These mutants are helpful for studying JA-Ile-mediated responses that depend on or are independent of COI1. However, exploring the details of JA signaling is not the focus of this manuscript, as adding an RNAseq dataset on JA-Ile/COI1 signaling would divert attention from our main topic. While we acknowledge that JA signaling is an interesting aspect that warrants further clarification, our study focuses on OPDA signaling. In this context, JA-Ile signaling was characterized solely to distinguish it from OPDA and wound-related signaling responses. Using OPDA application to *coi1* mutant, which retains active biosynthesis enzymes (including the three OPR enzymes, β -oxidation, and JAR1), would still result in JA-Ile production. In our view, this does not provide additional information for studying OPDA function. Consequently, this approach would not allow us to draw definitive conclusions about OPDA signaling.

Regarding the OPDA levels, in Figure S4d a wound-induced OPDA increase in *coi1-30* could be expected. Comparison between the wound-induced OPDA accumulation in Col-0 plants as well as the control in Figure S4c and S4d showed remarkable differences between the experiments. I am aware that inflicting mechanical wounding to Arabidopsis leaves might vary depending on the person and on the lab. I was wondering whether more intense wounding would result in a more pronounced accumulation of OPDA that would result in regulation of the previously reported OPDA-induced genes.

We agree on the variation in OPDA levels under control conditions and on the fact that the differences upon wounding are not that high *coi1-30* plants. Whereas *coi1-16* plants were four weeks old and grown under optimal conditions, *coi1-30* plants were already 6 weeks old and suffered from some water stress and overcrowding in the trays leading to touching while handling plants. This was expected to lead to such fluctuations and might be the reason for the higher OPDA levels under control conditions and the missing increase in OPDA levels upon wounding.

Due to time limitations, we were not able to circumvent this issue, however, even though OPDA was not increasing in this mutant, the transcript accumulation of the OPDA-marker genes was induced similarly to *coi1-16*, *opr2opr3* and *aos* (see Fig. 2) pointing again to their independence from OPDA.

As one of the reviewers pointed out, exogenous treatment with OPDA usually induces a hyperaccumulation of OPDA, whereas in Figure S7d the OPDA accumulation looks relatively similar after wounding or exogenous OPDA treatment. For the sake of comparison, this is a very good point (similar accumulation using two different triggers), but I am not convinced of these conditions being optimal for the proposed analysis. Again, an appropriate control would be OPDA-treated and wounded Col-0 and coi1-30.

In response to a previous reviewer's comment, we discussed that most OPDA application experiments – except for Arnold et al. (2016) – do not report the level of OPDA uptake following exogenous treatment. However, the values we report here are consistent with those of Arnold et al. (2016). Furthermore, the accompanying manuscript by Ueda et al. employed a more precise experimental setup using labeled OPDA and determined that the uptake of *cis*-OPDA-d5 is approximately 30 nmol/gFW at 30 minutes after feeding with 30 μ M OPDA (Fig. 4a, accompanying manuscript). These findings are fully consistent with our OPDA application results.

In Figure S6, the induction with MeJA did not seem to work as in seedlings based on the hormone measurements. Moreover, the PCA plot shows that the samples do not cluster so well. I agree that PC1 shows the variance depending on the treatment, but PC2 does not indicate that the samples group depending on the genotype. This variability might affect the interpretation of the results.

The reviewer is correct that the pre-treatment of adult plant was less efficient compared to seedlings. This was due to the experimental set-up: seedlings were grown in liquid media, which was then supplied with MeJA, whereas the adult plants had to be sprayed. The latter treatment was not completely optimized before sequencing and some of the leaves might not be pre-treated sufficiently. Also the control treatment (spraying with water) might have induced touch-responses in varying degrees as even water spraying induces touch responses, which are JA-mediated (Van Moerkercke et al. 2019). This might have resulted in the high variation also visible in the PCA plot. Nevertheless, the analysis of DEGs in *opr2opr3* and *aos* rosette leaves – done as an additional control only and therefore presented in the Supplementary data – resulted in the same genes as the analysis of seedlings showed.

Regarding the complementation of *opr2opr3* by any OPR3 variant (localized in different subcellular compartments), I would not necessarily interpret such results as compartmentalization of OPDA in the peroxisome. Alternatively, is it because during the translation of the OPR3 variants they could potentially encounter OPDA in the cytosol? In such case, the trans-organelle complementation might not be feasible with the current approach.

The reviewer is correct in noting that OPR enzymes might encounter OPDA in the cytosol. We carefully investigated this using microscopy, as all our experiments employed YFP-tagged versions of OPRs. Utilizing highly sensitive LSMs, we never detected YFP in the cytosol when OPRs were targeted to organelles. However, we were aware that the presence of these enzymes in the cytosol might be below the detection limit of our imaging. To address this, we leveraged the fact that OPR3 functions as a dimer and conducted split-YFP assays, where one OPR3 variant

was directed to an organelle (fused to one half of YFP) and the other was localized in the cytosol (fused to the other half of YFP). However, these experiments did not yield conclusive results. At the very least, we can state that the majority of the overexpressed protein was localized to the targeted organelle.

I am very intrigued by the complementation of opr2opr3 by mitochondrial OPRs. What do you think that the OPDA might be doing in the mitochondria? Does OPDA regulate mitochondrial genes?

I really like the idea of the trans-organellar complementation using OPRs with different substrate specificity, but there are a few aspects that should be discussed in more depth, such as the differences between enzymatic in vitro assays and the activity in planta, or the potential role of OPDA in mitochondria.

Although we excluded in the former version of our manuscript that OPDA might go into mitochondria, we cannot exclude this anymore. However, the rescue obtained with mitochondrial-located OPR3 does not indicate that OPDA occurs naturally/has a function in the mitochondria, it rather reflects the ability of the compound (or of its down-stream product 4,5-ddh-JA) to translocate to this compartment. We do not conclude from this result that OPDA has any potential role in mitochondria as none of the biosynthetic enzymes occur naturally in this organelle. The complementation with cytosolic and mitochondrial located OPR1, which converts OPDA and 4,5-ddh-JA although only to very limited degree, shows that there might be only limited amounts of OPDA or 4,5-ddh-JA in these compartments (see answer to Reviewer #2). We rewrote parts of the “Results” and “Discussion” accordingly and changed the summarizing Fig. 8. As pointed out in the answer to Reviewer #2, it has been demonstrated by Schaller et al. (2000) that the activity of recombinantly produced OPR enzymes reflects also the activity of enzymes produced *in planta*.

Reviewer #5 (Remarks to the Author):

Studying OPDA-specific functions in Angiosperm plants is difficult and full of pitfalls. The revised (v2) manuscript of Mekkaoui et al is an advanced study that reads very well. While many relevant comments have been made previously by two reviewers and consistent changes have been introduced, I will not come back to all solved questions but rather complement with a few additional points.

This is a nice piece of work where a long standing and ever-evolving question is shined with new light, owing to the combination of a smart set of genetic tools along with an elegant complementation strategy. By doing so, the authors delineate precisely the sectors of the wound response that are under the control of JA/JA-Ile, and spectacularly find out that there is no OPDA signature upon wounding. A main message is that some established 'OPDA markers' are not regulated by endogenous OPDA upon wounding. In a second part, they investigate convincingly some compartmentation aspects of the substrates of the different OPR isoforms and finally propose possible reasons for distinct responses to external OPDA supply.

The zig-zag historic route for knowledge acquisition on the topic of OPDA signaling is well introduced in the ms, taking into account most of the past studies and highlighting their limitations.

Authors treated properly queries on v1 by reviewer#1

#1 : OK with the arguments provided, but maybe reviewer #1 meant analyzing transcriptome in coi1. The coi1 transcriptome would be expectedly similar to those of opr2opr3 and of aos, but to my opinion the interpretation would be blurred by the fact that part of the OPDA produced would be converted to JA/JA-Ile, thus the homeostasis of OPDA is different from opr2opr3. Also, as argued, coi1 has less OPDA and cannot be complemented.

#2: OK

#3: OK

#4: the amounts of OPDA before and after OPDA feeding do not correspond to the cited Fig. 4c. Should be Fig. S7e ?

corrected

#5: OK

#7: OK

8: OK

My own comments:

Abstract

L26: slightly accumulate JA-Ile ?

L28: endogenous OPDA.

L29: were 'found' wound-induced

We had to rewrite the abstract drastically to keep the editorial requirements (150 words). Nevertheless, we corrected the respective words where they are still used.

Results

1. L116: The developmental stage/experimental setup used should be precised at start of results (Fig. S1), as wounding 10 d-old seedlings is unusual. The in vitro-grown material is therefore composed of whole seedlings (including roots), versus leaf-only for adult plants. Methods should

also precise i) number of seeds placed in which type of wells to achieve 120 seedlings; ii) how standardized wounding is applied to 120 liquid-grown seedlings.

Thanks to the reviewer for this comment. For all experiments with seedlings, ten to twelve seeds were sown into 2 mL of MS liquid medium per well in 24-well tissue culture plates. Seedlings from 12 wells (half a tray) were pooled to form one biological replicate for the experiments done with 120 seedlings per biological replicate, while seedlings from three wells were pooled in the experiments done with 30 seedlings per biological replicate. When grown in liquid MS, seedlings tend to cluster together; therefore, wounding was performed by squeezing the clustered seedlings (not individual ones) eight times using forceps with serrated teeth.

We added the following text to the “Material and Methods”: “After surface sterilization with 4% bleach, ten to twelve seeds were sown into 2 mL liquid Murashige and Skoog (MS) medium (pH 5.7, 1% sucrose) per well in a 24-well tissue culture plate (TPP, 92424), stratified at 4°C for three days and grown for ten days. Seedlings from several wells were pooled to form one biological replicate. Adult plants were grown individually in pots containing steam-sterilized clay, coir fiber, and vermiculite for four weeks. Three rosettes were pooled to form one biological replicate. Plant growth was conducted in Phytocabinets (Percival Scientific, www.percival-scientific.com/) at a light intensity of 120 $\mu\text{E m}^{-2} \text{s}^{-1}$ under short day conditions (10/14 hours light/dark cycle), at 21/19 °C and 65% relative humidity. For wounding treatments, clustered seedlings were squeezed eight times with moderate pressure using forceps with serrated teeth, while a single wound was inflicted in the midrib of all leaves of the adult rosettes.” (lines 514-524)

What about metabolite leaking/dilution from crushed material to medium?

We did not quantify such a leakage as our experimental design did not result in variation/inconsistencies across biological replicates or across independent experiments that would indicate possible impact of a metabolite leakage on our hormone measurements. Additionally, OPDA quantification results from seedlings do not hint towards such a leakage as the average amount of OPDA in wounded seedlings (grown in liquid MS) resembles that of wounded adult leaves which were not submerged in any liquid pre- or post-wounding (Supplementary Figure 6a).

2. There is an issue with statistical analysis in some instances (eg Fig. 1d, 1e, Sup Fig 6a.): authors discuss in text effects of treatment (for exemple gene inducibility), but statistical significance is shown only between genotypes, not treatments. A two-way ANOVA would be better suited to pinpoint what is significant.

We applied a Two-way-ANOVA where appropriate and changed the figures accordingly.

3. L212: It is noteworthy that CYP81D11 is the top upregulated gene by phytoprostane and OPDA in Muller et al (<https://doi.org/10.1105/tpc.107.054809>). Therefore, this would be a candidate OPDA marker.

The reviewer is correct that *CYP81D11* is upregulated in our data set and has previously reported as induced by phytoprostanes. We apologize for overlooking this. In the revised manuscript, we included the following sentence (lines 209-210): “Among them, *CYP81D11* has been shown to be highly induced by lipophilic xenobiotics such as phytoprostanes, OPDA, and *cis*-jasmonone^{30,31}.” However, in our opinion it is questionable whether a single gene is sufficient to serve as a marker for the signaling capacity of a compound.

4. L124: AOC is encoded in Arabidopsis by 4 genes (AOC1-4). The authors should precise if all 4 AOC are impacted by this positive feedback loop, and if the antibody used recognizes all 4 isoforms.

The text has been changed to clarify that the AOC is indeed encoded by four genes and that we detect all four isoforms by immunoblot simultaneously (see lines 126-132). Using an antibody that binds to all four isoforms (see Stenzel et al. 2003), we are unable to determine whether all AOCs proteins or only a subset is impacted by the JA feedback loop.

5. L170: I am not sure how these values are obtained considering the numbers in Fig. 2a. The number of JA-Ile-dependent DEGs appears substantially lower in 10d seedlings (barely visible in Fig. 2a, compared to a reduction by half in mutant rosettes Fig S6b) than in adult rosettes. Therefore, this sector of the wound-response seems to be age-dependent. Also, can the JA-dependent sector be limited compared to other studies because an early time point (1 h) is analyzed, and early-responding (signaling) transcripts are less JA-dependent, whereas JA-dependent, (defense) related transcripts are regulated at later timepoints ?

The reviewer is correct that the numbers of DEGs in seedlings of the three genotypes do not appear highly different but those of rosette leaves do. Nevertheless, the percentage change in seedlings between the DEGs of the genotypes was calculated (according to the mathematical formula: $(A - B)/(B) \times 100\%$) and revealed that an increase of 10.84% (= 206 DEGs) and 7.48% (128 DEGs) was found for WT/*opr2opr3* and WT/*aos*, respectively. These values are in the range of typically found for JA/JA-Ile dependent genes. The higher difference in DEGs between wild type and mutant rosette leaves points to a development-dependent response as proposed by the reviewer. This is now mentioned: "Here, the bigger difference in the number of DEGs between wild type and mutants in comparison to seedlings is likely caused by the developmental stage." (line 219-221)

It is correct that we analyzed only an early time point after wounding. It is well documented that JA/JA-Ile is produced after wounding within minutes (Glauser et al. 2008) and transcripts of the "early-induced" genes accumulate within 30 -60 min. As both JA and OPDA levels peak at 1 hour following wounding it is expected that their perception and the activation of their signaling cascade would correlate with their levels within the cell. This is the case for JA accumulation leading to its signaling starting already few minutes after wounding. Early JA responsive genes like the JAZ transcription factors, which are markers of JA signaling, have a pattern of transcript accumulation tightly correlating with JA accumulation as shown by Chung et al. (2008). The presence of a later JA transcriptional response does not exclude the presence of a detectable/significant earlier response at the transcriptional level. As OPDA accumulation in the cell rises and is peaking around 1 hour after wounding, its signaling is expected also to occur (or at least to start) within the same timeframe, unless its perception/signaling is independent from its pattern of accumulation in the cell. Even if a "hypothetically stronger" later signaling function is occurring, an earlier response "starting" would be expected if we take into consideration the rise of OPDA content in the cell. Even if there were a (low) number of OPDA-dependent genes similarly to the JA-dependent genes, they should be detectable within the data set.

6. Fig. 5b shows a nuclear export signal picture to the cytoplasm (NES), not a nuclear localization signal (NLS). Please explain/correct.

We apologize for the mistakes in the respective figure legend. We corrected the legend, but also the captions in Fig. 5.

In Fig. 5d, we show the nuclear-localized OPR3 Δ SRL, which was directed to the nucleus by adding a nuclear localization signal (NLS) at the N-terminus of OPR3, resulting in NLS-OPR3 Δ SRL-YFP. OPR3 Δ SRL, which is defective in peroxisomal import, showed a nuclear localization in addition to the cytosol in transient experiments in *N. benthamiana* (Fig. 5b). To ensure an exclusive cytosolic localization of the OPR3 cytosolic variant, a nuclear export signal (NES) was added at the C-terminus of OPR3 Δ SRL, creating OPR3 Δ SRL-YFP-NES (Fig. 5e). The construct with OPR3 Δ SRL-YFP-NES is therefore the cytosolic variant of OPR3 that was used in this study both for protoplast co-localization experiment (Supplementary Fig. 9) and for generating the transgenic Arabidopsis lines (Fig. 5e).

7. Fig. S12: how is it explained that OPDA levels are quite fluctuating across the two equivalent lines whereas JA/JA-Ile levels are strikingly stable in the same sibling lines ?

The reviewer is correct that the OPDA levels appear to be fluctuating. Statistical analysis using Two-way-ANOVA revealed, however, that there are no significant differences among most of the lines. Only one line (line #2 of px-OPR2) deviates and showed higher OPDA levels. We do not have an explanation for why the OPDA levels were fluctuating but the corresponding JA levels were stable. It is tempting to speculate that JA levels are highly regulated and higher OPDA levels do not necessarily lead to higher JA levels.

Minor points:

L142: this statement is not fully valid in absence of data for wounding without MJ treatment. Correct, this statement has been removed.

All Figures with bar charts : individual datapoints are barely visible in black on dark grey histograms. As a general comment, colors could be used for bar filling, or lighter tones of background used.

The gray scales have been modified in all figures to ensure better reading.

L120: It is not clear if authors mean detection or quantification. As discrete values are given for opr2opr3 wounded, 0.002 nmol/g FW seems like LOQ.

The reviewer is right. The text has been modified accordingly (line 121).

L275: this conclusion applies to a wounded seedling context, but cannot be generalized yet, as signaling properties have been associated with thermotolerance.

Thanks to the reviewer for this comment. According to this we specified throughout the manuscript that our results rely on the early wound-response of seedlings. Our results include also data of mature rosettes showing similar outcomes as seedlings, in addition to the control condition, which we think should not be disregarded, as in the case of JA we detect transcriptional differences between JA-producing and deficient-plants not only at wounding but also at control condition. Hence, to be faithful to the whole dataset that we present, we think that our conclusions

encompass the investigation of OPDA signaling at basal conditions, and early wounding of seedlings and mature rosettes.

The thermotolerance function of OPDA was only demonstrated in *Marchantia polymorpha* and *Klebsormidium nitens*, as heat led to an accumulation of OPDA which in turn indeed led to a thermotolerance phenotype in these species (Monte et al. 2020). In *Arabidopsis*, however, such a phenotype was not demonstrated, as OPDA was not shown to accumulate following heat and no thermoresistance phenotype was shown. The authors rather investigated Gene Ontology enrichment in *opr3-3 coi1-1* double mutant plants treated with OPDA and found an enrichment in the response to heat and thermotolerance without characterizing specific genes related to this putative heat “signaling” function. We do not exclude such a function, however, also such a signaling function in *Arabidopsis* is only suggested through a GO analysis and is not yet demonstrated.

L281: the phenotypic rescue can happen only if the OPR3 product, OPC:8 is produced in a location from where it can access β -oxidation in peroxisomes, possibly by comatose-facilitated import. One could imagine a scenario where a re-located version of OPR3 catalyzes the reduction of occurring OPDA, but that OPC:8 cannot reach peroxisomes and prevent subsequent formation of JA-Ile (for ex can OPDA freely enter and 4,5ddh-JA exit mitochondria ?). In other words, the restoration of fertility requires more than just OPDA metabolization.

The reviewer is correct – if OPR3, when localized in any organelle, converts OPDA to OPC-8, this compound must then be transported into peroxisomes for β -oxidation. According to the rescue of the mutant by all OPR3 variants, there seems to be no limitation, since targeting OPR3 in any organelle rescues the JA/JA-Ile production. However, as demonstrated by Ueda et al. (accompanying manuscript), exogenously applied OPDA is preferentially taken up by peroxisomes in a CTS-dependent manner and subsequently converted to 4,5-ddh-JA. Given the differing hydrophilicity of these compounds, the movement of 4,5-ddh-JA into other organelles is more likely than that of OPDA, as is the transport of OPC-8 back into peroxisomes. We attempted to clarify this by analyzing the mitochondrial localization of OPR1; however, since OPR1 converts both substrates, we can no longer draw definitive conclusions about the identity of the “mobile” substrate.

L287: cytosol and nucleus?

Corrected to “Removal of this signal resulted in cytosolic and nuclear localization of a fusion of OPR3 Δ SRL with YFP when transiently expressed in *Nicotiana benthamiana* leaves, indicating that import into peroxisomes was abolished (Fig. 5b).” (lines 285-287)

L295: ER-OPR3 and cytosol-OPR3 also label the nucleus

We think that the reviewer based this conclusion solely on the protoplast subcellular localization assay (Supplementary Fig. 9). Due to the comment from Reviewer #3, we repeated the protoplast transformation and provided new micrographs (see answer to Reviewer #3). These show unequivocally that neither ER-OPR3 nor cytOPR3 is located within the nucleus. This is especially obvious for the cytosol, where the organelle marker (free mCherry) labels the nucleus but cytOPR3 not. Since OPR3 Δ SRL (defective in peroxisomal import) expressed in *N. benthamiana* labelled the nucleus in addition to the cytosol (Fig. 5b; title of Fig. 5 changed accordingly), we decided to include an NES for testing the subcellular localization in *N. benthamiana* protoplasts and for

creating stable transgenic Arabidopsis lines. The OPR3 Δ SRL:NES does not show any nuclear localization neither in protoplasts (Supplementary Fig. 9) nor in the Arabidopsis stable transgenic lines (Fig. 5e).

L316: why only these 3 compartments?

To test the effect of overexpression of the OPR3 on wound-induced JA/JA-Ile production, we decided to use these three compartments to compare the “native” site of OPR3 location (peroxisomes) with the artificial location in the cytosol, where OPDA has to be transferred through, and one compartment, which is an organelle not involved in JA biosynthesis. We included these reasons in the text (lines 313-316).

Discussion

L411: At first sight, this contradicts markers used by Ueda et al. This means that they can be induced by exogenous OPDA, but are wound-induced independently of OPDA.

The reviewer is correct – this might be a contradiction. In the following text, we explain that either the electrophilic property of OPDA and 4,5-ddh-JA or a common stress response might be responsible for the induction of these genes. (lines 402-403 and 421-424)

L427: ‘conclusion’ rather than ‘fact’. Here I suggest to be more cautious: OPDA could mediate responses by chemical reactivity, not requiring a unique, specific receptor. Also, the conclusion should be restricted to wounding as other biological situations may lead to different interpretations.

We replaced the word “fact” by “conclusion” (line 422). Our transcriptomic results dismiss the possibility of a transcriptional mediated signaling of OPDA through a specific receptor or through its electrophilic property. If such electrophilic property mediates transcriptional changes (e.g., via TGA transcription factors) as it was previously reported (Park et al. 2013; Müller et al. 2017), we would still expect differential transcriptional change in *opr2opr3* compared to *aos* at wounding and/or control conditions. An aspect that cannot be excluded is whether the signaling of OPDA through its electrophilic properties mediates changes at the protein level – we did not address in this work as our focus is the signaling through transcriptional regulation, an aspect that was widely investigated when studying OPDA signaling functions in previous studies.

It is important to note that our data-set provides a transcript profiling not only at early wounding but also at control condition. At control condition, OPDA is not absent as significant basal levels of OPDA are detectable (Fig. 1E, Supplementary Fig. 6a). Therefore, we think that restricting our conclusions to wounding only would not reflect the results from the whole dataset that we present here. We agree that other biological conditions or treatments or developmental cues not yet uncovered might result in a signaling of OPDA in Arabidopsis and we mention this in the discussion (lines 504-506).

L449: it would be worth adjusting concentration of exogenous OPDA such as 4,5-ddh-JA also reaches 500 pmol/g FW and compared gene inducing activity.

Thanks to the reviewer for raising this point. It is a great idea to “adjust” the concentration of applied OPDA to get 4,5-ddh-JA levels that are in the range of the endogenously formed levels upon wounding. Since the goal of our study was to clarify the putative signaling of OPDA, the application experiment was used to reach the same levels of OPDA as seedlings produce upon wounding and to compare the alterations in the transcriptome at similar levels of OPDA either

produced endogenously or applied exogenously. The separation of putative signaling by 4,5-ddh-JA is beyond the current study but might be highly interesting to be addressed in future.

L476: maybe a mistake on OPR2 and OPDA. This contradicts line L336 and ref 21.

Thanks to the reviewer – the mistake has been corrected.

L491: in wounded Arabidopsis

Here we are reporting complementation of male fertility and complementation of JA formation upon wounding. The complementation of fertility reflects JA/OPDA dynamics at control condition in the complemented lines; therefore, we consider that the results from this experiment are not limited to wounded Arabidopsis.

What about treating aos with OPDA? This would show the responses to exogenous OPDA without the context of endogenous OPDA. opr2opr3 also produces arabidopsides, aos not..

Treating the *aos* mutant with OPDA will lead to JA/JA-Ile formation, as OPRs and other JA biosynthesis enzymes downstream of AOS remain active. Consequently, this experimental condition will trigger JA-Ile signaling, preventing the distinction between JA and OPDA signaling. Therefore, this approach does not allow for drawing conclusions specifically about OPDA signaling.

It is unexpected to see that all plant lines expressing an OPR active on 4,5-ddh-JA do complement regardless of the compartment. Does this mean that there is some 4,5-ddh-JA in any cellular compartment?

The conclusion that we drew from this result was that 4,5-ddh-JA possesses the ability to translocate to all different cell compartments. This does not mean that in natural conditions 4,5-ddh-JA literally occurs in these cell compartments, since its levels are in wild-type plants rather low. However, since targeting of OPR1 into mitochondria does not show unequivocally that only 4,5-ddh-JA might appear there (OPR1 is active on both substrates, although to very limited extent), we have withdrawn this conclusion (see response to Reviewer #2).

References:

- Arnold MD, Gruber C, Floková K, Miersch O, Strnad M, Novák O, Wasternack C, Hause B (2016) The recently identified isoleucine conjugate of *cis*-12-oxo-phytodienoic acid is partially active in *cis*-12-oxo-phytodienoic acid-specific gene expression of *Arabidopsis thaliana*. PLoS ONE 11 (9):e0162829. doi:10.1371/journal.pone.0162829
- Chini A, Monte I, Zamarreño AM, Hamberg M, Lassueur S, Reymond P, Weiss S, Stintzi A, Schaller A, Porzel A, García-Mina JM, Solano R (2018) An OPR3-independent pathway uses 4,5-didehydrojasmonate for jasmonate synthesis. Nature Chemical Biology 14:171. doi:10.1038/nchembio.2540
- Chung HS, Koo AJK, Gao X, Jayanty S, Thines B, Jones AD, Howe GA (2008) Regulation and function of Arabidopsis JASMONATE ZIM-domain genes in response to wounding and herbivory. Plant Physiology 146 (3):952-964
- Glauser G, Grata E, Dubugnon L, Rudaz S, Farmer EE, Wolfender J-L (2008) Spatial and temporal dynamics of jasmonate synthesis and accumulation in Arabidopsis in response to wounding. J Biol Chem 283 (24):16400-16407
- Koo AJK, Gao X, Jones AD, Howe GA (2009) A rapid wound signal activates the systemic synthesis of bioactive jasmonates in Arabidopsis. Plant J 59 (6):974-986
- Monte I, Kneeshaw S, Franco-Zorrilla JM, Chini A, Zamarreño AM, García-Mina JM, Solano R (2020) An ancient COI1-independent function for reactive electrophilic oxylipins in thermotolerance. Current Biology 30 (6):962-971.e963. doi:10.1016/j.cub.2020.01.023
- Müller SM, Wang S, Telman W, Liebthal M, Schnitzer H, Viehhauser A, Sticht C, Delatorre C, Wirtz M, Hell R, Dietz K-J (2017) The redox-sensitive module of cyclophilin 20-3, 2-cysteine peroxiredoxin and cysteine synthase integrates sulfur metabolism and oxylipin signaling in the high light acclimation response. The Plant Journal 91 (6):995-1014. doi:10.1111/tpj.13622
- Park S-W, Li W, Viehhauser A, He B, Kim S, Nilsson AK, Andersson MX, Kittle JD, Ambavaram MMR, Luan S, Esker AR, Tholl D, Cimini D, Ellerström M, Coaker G, Mitchell TK, Pereira A, Dietz K-J, Lawrence CB (2013) Cyclophilin 20-3 relays a 12-oxo-phytodienoic acid signal during stress responsive regulation of cellular redox homeostasis. Proceedings of the National Academy of Sciences 110 (23):9559-9564. doi:10.1073/pnas.1218872110
- Schaller F, Biesgen C, Müssig C, Altmann T, Weiler EW (2000) 12-Oxophytodienoate reductase 3 (OPR3) is the isoenzyme involved in jasmonate biosynthesis. Planta 210 (6):979-984
- Stenzel I, Hause B, Miersch O, Kurz T, Maucher H, Weichert H, Ziegler J, Feussner I, Wasternack C (2003) Jasmonate biosynthesis and the allene oxide cyclase family of *Arabidopsis thaliana*. Plant Mol Biol 51:895-911
- Van Moerkercke A, Duncan O, Zander M, Šimura J, Broda M, Vanden Bossche R, Lewsey MG, Lama S, Singh KB, Ljung K, Ecker JR, Goossens A, Millar AH, Van Aken O (2019) A MYC2/MYC3/MYC4-dependent transcription factor network regulates water spray-responsive gene expression and jasmonate levels. Proceedings of the National Academy of Sciences 116 (46):23345-23356. doi:10.1073/pnas.1911758116

Response to reviewers' comments to NCOMMS-24-18845B:

We are grateful to the reviewers for evaluating the revised version of our manuscript. We are happy that we have improved the manuscript and that it is now almost ready for publication.

The reviewers' comments and our answers are listed in the following paragraphs.

Reviewer #3:

No comments

Reviewer #4 (Remarks to the Author):

In the revised version of the manuscript the authors have convincingly addressed the reviewers' concerns. Regarding the induction of JA-regulated genes by wounding in *opr2opr3*, the authors should correct the following statement:

Line 253-255: "Validation of transcript accumulation of JAZ2, JAZ7, JAZ13 and CHL1 by RT-qPCR confirmed their exclusive induction in *opr2opr3* seedlings upon OPDA application but not upon endogenous OPDA accumulation by wounding (Supplemental Fig. 7d)."

This is not correct. The expression of JAZ2 and JAZ7 is induced by wounding in *opr2opr3* albeit to a lesser extent than in *opr2opr3* treated with OPDA. How do you explain the induction of JA-regulated genes in *opr2opr3* after wounding if you consider that the OPDA conversion to JA-Ile is negligible?

Answer:

The reviewer is right that *JAZ2* and *JAZ7* are slightly induced by wounding in the *opr2opr3* mutant. A JA-independent induction of some of the so-called JA-specific genes was previously observed by Chung et al. ("Regulation and function of Arabidopsis *JASMONATE ZIM*-domain genes in response to wounding and herbivory", Chung HS, Koo AJK, Gao X, Jayanty S, Thines B, Jones AD, Howe GA, Plant Physiology, 2008, 146: 952-964, Fig2). Here, several JAZ genes are induced by wounding in the *coi1* mutant despite the absence of JA-Ile perception. The induction is weaker in the *coi1* mutant compared to wild type - similarly to what we observed in OPDA application vs. wounding of *opr2opr3* seedlings. Hence, their weak induction in the wounded *opr2opr3* mutant does not necessarily indicate JA production. A wound-dependent, JA-independent low induction in this case can be the explanation.

We changed the text in lines 253-255 accordingly: "Validation of transcript accumulation of *JAZ2*, *JAZ7*, *JAZ13* and *CHL1* by RT-qPCR confirmed their induction in *opr2opr3* seedlings upon OPDA application (Supplemental Fig. 7d). While the induction of *JAZ13* and *CHL1* was exclusive to OPDA treatment, *JAZ2* and *JAZ7* showed a weak induction by wounding in the *opr2opr3* seedlings, however, to a lower extent compared to OPDA treatment."

Minor comments:

Suppl. Fig. 1B: the y axis lacks part of the labels

Corrected

Figure 7: The figure will be easier to read if the graphs in panel d showed the same order as the images in a-c: cytosol, peroxisome, mitochondria.

Corrected accordingly

Additional comments regarding the response to **Reviewer #2's** concerns (paraphrased):

The authors have addressed the concerns raised by Reviewer #2 on the misinterpretation of the results in the Chini et al., 2018 paper. I would recommend that OPR1 is also included in the final model in Figure 8 together with OPR2 in the cytosol of the WT cell.

done

Reviewer #5 (Remarks to the Author):

The authors responded convincingly to my comments.

L32 abstract: 'located' may be replaced with 'targeted'

done